# A small climate-amplifying effect of climate-carbon cycle feedback

Xuanze Zhang [1,2✉], Ying-Ping Wang [3,4✉], Peter J. Rayner [5], Philippe Ciais [6], Kun Huang[2], Yiqi Luo [7], Shilong Piao [8], Zhonglei Wang[9], Jianyang Xia [2], Wei Zhao[10], Xiaogu Zheng[11], Jing Tian[1] & Yongqiang Zhang [1✉]

The climate-carbon cycle feedback is one of the most important climate-amplifying feedbacks of the Earth system, and is quantified as a function of carbon-concentration feedback parameter ($\beta$) and carbon-climate feedback parameter ($\gamma$). However, the global climate-amplifying effect from this feedback loop (determined by the gain factor, $g$) has not been quantified from observations. Here we apply a Fourier analysis-based carbon cycle feedback framework to the reconstructed records from 1850 to 2017 and 1000 to 1850 to estimate $\beta$ and $\gamma$. We show that the $\beta$-feedback varies by less than 10% with an average of $3.22 \pm 0.32$ GtC ppm$^{-1}$ for 1880–2017, whereas the $\gamma$-feedback increases from $-33 \pm 14$ GtC K$^{-1}$ on a decadal scale to $-122 \pm 60$ GtC K$^{-1}$ on a centennial scale for 1000–1850. Feedback analysis further reveals that the current amplification effect from the carbon cycle feedback is small ($g$ is $0.01 \pm 0.05$), which is much lower than the estimates by the advanced Earth system models ($g$ is $0.09 \pm 0.04$ for the historical period and is $0.15 \pm 0.08$ for the RCP8.5 scenario), implying that the future allowable $CO_2$ emissions could be $9 \pm 7\%$ more. Therefore, our findings provide new insights about the strength of climate-carbon cycle feedback and about observational constraints on models for projecting future climate.

[1] Key Laboratory of Water Cycle and Related Land Surface Processes, Institute of Geographic Sciences and Natural Resources Research, Chinese Academy of Sciences, Beijing, China. [2] Research Center for Global Change and Ecological Forecasting, School of Ecological and Environmental Science, East China Normal University, Shanghai, China. [3] Terrestrial Biogeochemistry Group, South China Botanical Garden, Chinese Academy of Sciences, Guangzhou, China. [4] CSIRO Oceans and Atmosphere, Private Bag 1, Aspendale, Victoria, Australia. [5] School of Earth Sciences, Climate and Energy College, University of Melbourne, Parkville, Victoria, Australia. [6] Laboratoire des Sciences du Climat et de l'Environnement, LSCE/IPSL, CEA-CNRS-UVSQ, Université Paris-Saclay, Gif-sur-Yvette, France. [7] Center for Ecosystem Science and Society, Northern Arizona University, Flagstaff, AZ, USA. [8] Sino-French Institute for Earth System Science, College of Urban and Environmental Sciences, Peking University, Beijing, China. [9] Wang Yanan Institute for Studies in Economics (WISE) and School of Economics, Xiamen University, Xiamen, China. [10] National Meteorological Center, China Meteorological Administration, Beijing, China. [11] Key Laboratory of Regional Climate-Environment Research for East Asia, Institute of Atmospheric Physics, Chinese Academy of Sciences, Beijing, China. ✉email: xuanzezhang@igsnrr.ac.cn; yingping.wang@csiro.au; zhangyq@igsnrr.ac.cn

                    1

With increasing atmospheric $CO_2$ and a warming climate during the industrial era, land and ocean reservoirs have together absorbed >50% of anthropogenic $CO_2$ emissions[1], playing a significant role in reducing anthropogenic warming[2]. Whether the ocean and land sinks will continue to take up a similar fraction in the future remains uncertain, and one of the major causes for that uncertainty is the feedback between the carbon cycle and the physical climate system[3–5]. Global climate change affects carbon uptake by land and oceans, which impacts the rate of increase in atmospheric $CO_2$ and, in turn, climate change. This feedback loop between the physical climate system and the global carbon cycle of the Earth system was quantified using a modeling approach[6–9].

Previous studies quantified the climate–carbon cycle feedback as a function of the carbon-concentration feedback response parameter ($\beta$) and the carbon–climate feedback response parameter ($\gamma$)[7,10]. The $\beta$ (GtC ppm$^{-1}$) and $\gamma$ (GtC K$^{-1}$) are also defined as the rates of change in land and ocean carbon storages relative to a fixed reference time to atmospheric $CO_2$ concentration increase and to global climate change that is often quantified by the global-mean surface temperature change, respectively. From the perspective of the land and ocean reservoirs, $\beta$ is positive, and $\gamma$ is negative. Therefore, $\beta$-feedback reduces the impact of $CO_2$ emissions on atmospheric $CO_2$ concentrations and then global warming, while $\gamma$-feedback amplifies global warming. The combined effects of the $\beta$-feedback and the $\gamma$-feedback and the nonlinear interaction between them determine the strength of the climate–carbon cycle feedback loop which is known as the feedback gain factor ($g$).

Both $\beta$ and $\gamma$ were previously quantified using physical climate models coupled with the global carbon cycle under idealized experiments[7,10,11]. Based on the experiments by the advanced Earth system models, the last two successive assessments by the International Panel on Climate Change (IPCC) found that the uncertainty of the climate–carbon cycle feedback was dominated by $\beta$[12,13]. These Earth system models under idealized experiments did not account for the biophysical effect of land-use change on climate and ecological effects on the residence time of carbon in the land biosphere. Modeling experiments showed that in a high $CO_2$-induced warming climate system, the nonlinear carbon–climate feedback can reduce the ocean carbon uptake by 3.6–10.6% based on the simulations of seven Earth system models[14].

On the other hand, observations can be used to constrain the estimated climate–carbon cycle feedbacks. Applying an emerging constraint-based approach to instrumental records, Cox et al.[15] estimated $\gamma$ for tropical land to be $-53 \pm 17$ GtC K$^{-1}$ by 2100. Using data from three ice cores and multiple temperature reconstructions from tree ring data over 1050–1800, Frank et al.[16] reported a $\Delta C_A/\Delta T_A$ (defined as $\eta$ in this study) of 7.7 ppm K$^{-1}$ with a likely range of 1.7–21.4 ppm K$^{-1}$, which was much lower than previous estimates (e.g., ~40 ppm K$^{-1}$)[17,18]. However, the $\eta$ is not truly comparable with $\gamma$, because their relationship also depends on $\beta$ (see Eq. (3) below). In quantifying the response of climate to an increase in atmospheric $CO_2$ concentration or anthropogenic emissions, two other quantities are frequently used in the studies of carbon–climate interactions: the sensitivity of climate to atmospheric $CO_2$ ($\alpha \equiv \Delta T_A/\Delta C_A$, in K GtC$^{-1}$, which is equivalent to $1/(2.12 \eta)$ in this study)[7,19], and the transient climate response to cumulative $CO_2$ emission (TCRE $\equiv \Delta T_A/\Delta C_E$, in K GtC$^{-1}$)[20]. The values of $\beta$, $\gamma$, $\alpha$, and TCRE are related to each other through one equation (see Eq. (7))[10,21], and that equation is applicable to any individual frequency (or timescale) (see "Methods").

## Results

**Estimates of $\beta$ and $\gamma$ across timescales.** Based on the previous studies[7,10,14,19], the climate–carbon cycle feedback framework with a nonlinear feedback term as a function of $\beta$ and $\gamma$ parameters, i.e., $f(\beta,\gamma)$ in a unit of GtC ppm$^{-1}$ K$^{-1}$ or GtC GtC$^{-1}$ K$^{-1}$, in a $CO_2$ emission-driven coupled climate–carbon cycle system, can be expressed as:

$$\Delta C_E = \Delta C_A + \beta \Delta C_A + \gamma \Delta T_A + f(\beta, \gamma) \Delta C_A \Delta T_A \quad (1)$$

where $\Delta C_E$, $\Delta C_A$, and $\Delta T_A$ are changes in cumulative $CO_2$ emissions, atmospheric $CO_2$, and global surface temperature, respectively, during a time interval ($\Delta t$). Here, we applied a Fourier analysis-based spectral decomposition approach to observations or reconstructed records of global surface temperature and atmospheric $CO_2$ to quantify how the two feedback parameters vary over different periods of time, or across different timescales (1/frequencies) over the same time period. As shown in the "Methods" section, the relationship among the four parameters at any given timescale ($k$) is given by

$$(\text{TCRE}^{-1})_k = (1 + \beta_k)(\alpha^{-1})_k + \gamma_k^* \quad (2)$$

where $\gamma^* = \gamma + f(\beta, \gamma) \Delta C_A$ represents the combined carbon–climate feedback from the linear $\gamma$-feedback and the nonlinear impact of atmospheric $CO_2$ on the carbon–climate feedback (see "Methods"). Equation (2) also shows that variations of $\text{TCRE}^{-1}$ and $\alpha^{-1}$ for a given timescale are linearly related, with a slope of $1 + \beta_k$ and an intercept of $\gamma_k^*$. Our result from modeling experiments (see Supplementary Table 4) showed that the nonlinear feedback term $f(\beta, \gamma)$ is relatively small, and its contribution ($f(\beta, \gamma)\Delta C_A$) to the $\gamma^*$ is $15 \pm 23\%$, while its contribution ($f(\beta, \gamma)\Delta T_A$) to the $\beta$-feedback is negligible ($3 \pm 3\%$). Thus, we indicate that the nonlinear feedback term has negligible effect on the estimate of the slope of $1 + \beta_k$.

We estimated $(\text{TCRE}^{-1})_k$ from the ratio of the variability in $C_E$ to the variability in $T_A$, and $(\alpha^{-1})_k$ from the ratio of the variability in $C_A$ to the variability in $T_A$ on a given timescale (see "Methods"). In Fourier analysis, the variabilities of $C_E$, $C_A$, and $T_A$ were represented by the amplitudes of their harmonics at a given frequency or period (Supplementary Fig. 1), which were then used to calculate $\text{TCRE}^{-1}$ and $\alpha^{-1}$ across different timescales (see "Methods"). Different from the previous approaches based on a Taylor series expansion and modeling experiments (hereafter referred to as the FEA approach, see Eqs. (22–25) in "Methods")[7,10,14], the estimated $\beta$ and $\gamma$ using Fourier analysis-based approach do not depend on a reference time that was used to compute changes in $\Delta C_E$, $\Delta C_A$, and $\Delta T_A$ as in the FEA approach.

In the following, we apply the Fourier analysis-based approach to observed time series of $C_E$, $C_A$, and $T_A$ to estimate $\beta_k$ and $\gamma_k$ for the industrial period (1850–2017). By ignoring the timescale dependence of $\beta$, and using the value of $\beta$ for the industrial period, we estimated $\gamma_k$ at different timescales using large ensembles based on combinations of ice-core atmospheric $CO_2$ records ($C_A$) and reconstructed surface air temperature ($T_A$) datasets ("Methods") over the period 1000–1850. Finally, we compare the observation-based estimates of $\beta$ and $\gamma$, and corresponding feedback gain factor ($g$) with those estimated from the simulations from multiple Earth system models.

**Estimates of $\beta$ and $\gamma^*$ for the industrial period (1850–2017).** Figure 1 shows the observed increase in atmospheric $CO_2$ ($\Delta C_A$) from ~285 ppm in 1850 to 405 ppm in 2017 in response to the increase in cumulative $CO_2$ emissions ($\Delta C_E$), including fossil fuel combustion[22] and land-use change[23,24], which amounted to ~$630 \pm 42$ GtC by 2017 since 1850, with respect to a concurrent

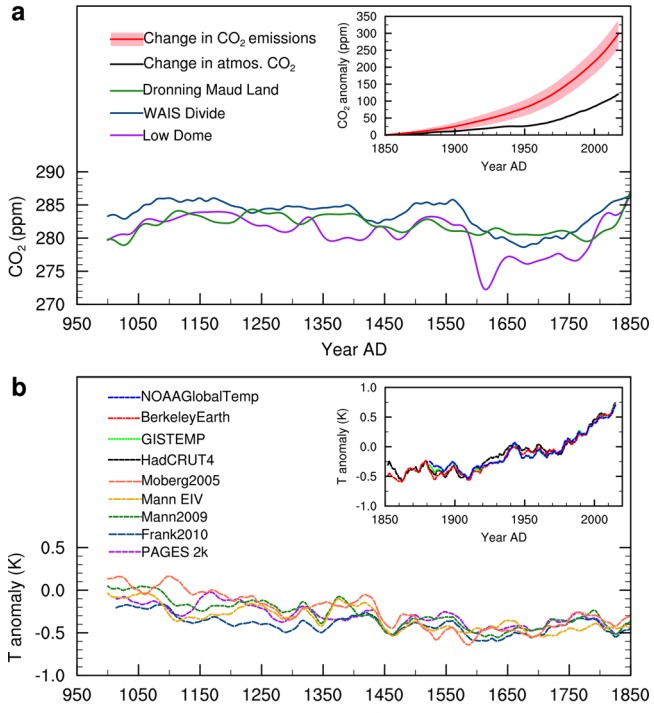

**Fig. 1 Variability in atmospheric CO₂ concentration and temperature over the past millennium. a** Variations in atmospheric CO₂ concentrations over 1000–1850 from three Antarctic ice-core records at Law Dome, WAIS Divide, and Dronning Maud Land. The subplot in **a** shows changes in atmospheric CO₂ concentrations and cumulative anthropogenic CO₂ emissions over 1850–2017 with respect to values in 1850. The cumulative anthropogenic CO₂ emissions with an uncertainty of ±1σ confidence interval (red-shaded) include CO₂ release from both fossil fuel combustion and land-use change (unit was converted to ppm from GtC). **b** Variations in northern hemispheric-mean temperature anomalies from five reconstructions (PAGES2k, Frank2010, Mann2009, MannEIV, and Moberg2005). The subplot in **b** shows global-mean temperature anomalies calculated from land surface air temperature and sea surface temperature with respect to the average of 1961–1990 from datasets of HadCRUT4 and Berkeley Earth for 1850–2017, and GISTEMP and NOAA GlobalTemp for 1880–2017. The reconstructions of 1000–1850 in (**a**) and (**b**) were smoothed with 30-year splines. Temperature anomalies in 1850 (1880)–2017 in (**b**) were smoothed with 5-year splines.

increase in global-mean annual temperature ($\Delta T_A$) of ~1.1 K over the same period based on the averages of the four global temperature datasets. Using Fourier analysis, we found that the amplitudes of $C_E$, $C_A$, and $T_A$ for the industrial period increased nonlinearly with timescale (Fig. 2a and b), but the variations in $(TCRE^{-1})_k$ and $(\alpha^{-1})_k$ with timescale were linearly related to each other ($R^2 = 0.99$, $P < 0.001$) with a slope (mean ± 1 standard deviation) of $2.52 \pm 0.15$ (see Fig. 2c and Supplementary Table 1 for individual estimates). Across different timescales from inter-annual to multi-decadal (e.g., $k$ from 2 to 90 years) for the 1880–2017 period, $(\alpha^{-1})_k$ varied from 50 to 250 GtC K⁻¹ and $(TCRE^{-1})_k$ from 125 to 600 GtC K⁻¹ (Fig. 2c).

The slope of the linear regression between $(TCRE^{-1})_k$ and $(\alpha^{-1})_k$ across different timescales can be used to estimate $(1 + \beta_k)$ based on Eq. (2). The result in Fig. 2c shows that the value of $\beta_k$ is approximately constant across different timescales, with an average value of $3.22 \pm 0.32$ GtC ppm⁻¹ (or $1.52 \pm 0.15$ GtC GtC⁻¹) throughout the industrial period in 1880–2017 (Supplementary Table 1). The uncertainty in $\beta$ results from the uncertainties in $C_E$ from fossil fuel combustion and land-use change estimates and

errors in the temperature datasets and the regression (see "Methods"). Using the estimated $\beta$ for the industrial period, we estimated the corresponding $\gamma^*$ to be $-10.9 \pm 3.6$ GtC K⁻¹ for the period 1880–2017 based on Eq. (6).

To diagnose the constancy of $\beta$ estimated from the industrial period and the reliability of our Fourier analysis-based approach, we used a box model to predict changes in temperature and CO₂ concentration with CO₂ emissions as forcing input (see "Methods", Eqs. (29)–(30)). We ran the box model using the estimate of two parameters based on observations (i.e., $\beta = 3.22$ GtC ppm⁻¹ and $\gamma^* = -10.9$ GtC K⁻¹) and the annual cumulative emissions over 1850–2017. The model predictions fitted very well to the observed values of atmospheric CO₂ concentration and surface temperature ($R^2 = 0.99$, RMSE = 3.5 ppm for CO₂, and $R^2 = 0.96$, RMSE = 0.17 K for temperature) (see Supplementary Fig. 2). The greater trend of the predicted global surface temperature relative to the observed after 1980s (Supplementary Fig. 2) is likely associated with $\gamma^*$ being treated as a constant, while in reality $\gamma^*$ could vary with internal climate variability and across different timescales. Using the box model predicted annual CO₂ and temperature, we in return, estimated a nearly constant $\beta$ (3.42 GtC ppm⁻¹) and a $\gamma^*$ ($-12.2$ GtC K⁻¹) for the period 1880–2017 based on our approach of Eqs. (1) and (2). Both values of $\beta$ and $\gamma^*$ estimated from the predicted CO₂ and temperature time series by the box model fall within the uncertainties of observation-based $\beta$ and $\gamma^*$, suggesting that the $\beta$ for 1880–2017 being nearly constant is robust.

As the $\gamma^*$ for the industrial period consists of the $\gamma$-feedback and the nonlinear feedback effect of $f(\beta, \gamma)\Delta C_A$, which results from the multi-decadal climate variability and anthropogenic CO₂ emissions. Because of the relatively short records over the industrial period, we could not separate the $\gamma$-feedback parameter directly from the $\gamma^*$ using our analysis framework. The $\eta$ (or $\alpha^{-1}$) for the industrial period includes a possibly significant contribution from an emissions-driven increase in atmospheric CO₂ concentration to the climate–carbon feedback. As a result, the mean value of $\eta$ for 1850–2017 is 109 ppm K⁻¹, much higher than the estimates of 7.7–40 ppm K⁻¹ during the preindustrial period before 1850s[16,17] that is considered to be at quasi-equilibrium with small variation in greenhouse gas forcing. The reason for this large discrepancy in $\eta$ between the preindustrial and industrial periods is likely a consequence of the nonlinear dependence of radiative forcing on atmospheric CO₂ concentrations[25], and the temperature change in response to the increase in atmospheric CO₂ during the industrial era has not reached steady state, as large part of atmospheric CO₂ increase during industrial period was driven by emissions not due to warming-induced CO₂ release from land and ocean reservoirs. It is also well known that equilibrium climate sensitivity is often considerably larger than the transient climate sensitivity[26]. In the following, we used reconstructed records of atmospheric CO₂ and surface temperature during the preindustrial last millennium from 1000 to 1850[16–18,27,28] to estimate the $\gamma$.

**Timescale dependence of $\gamma$ over the preindustrial last millennium (1000–1850).** The preindustrial last millennium (1000–1850) climate could be considered as a quasi-equilibrium state with very little change in CO₂ emissions, therefore the $\Delta C_A$ during 1000–1850 was largely driven by carbon–climate feedback without the complication of concurrent increase in $C_E$ as the industrial period. We ignored the possible influences of CO₂ emissions from the early land-use[29] on global surface temperature during 1000–1850. As the atmospheric CO₂ was relatively stable (280 ± 8 ppm) during the 1000–1850 period, i.e., $\Delta C_A$ was only ~3% of atmospheric CO₂ concentration (Fig. 1a), we assumed

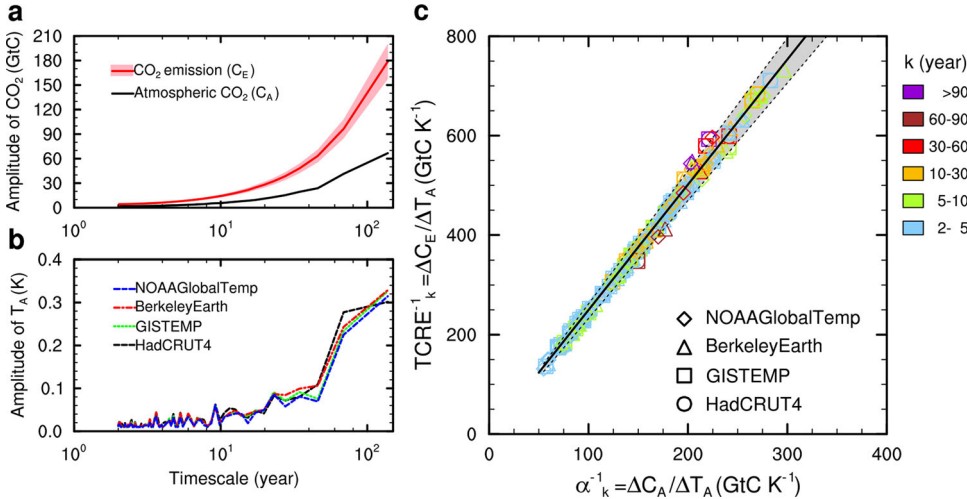

**Fig. 2 Linear relationship between $\frac{1}{\alpha}$ and $\frac{1}{TCRE}$ across timescales. a** Amplitude spectrum from Fourier analysis for annual atmospheric $CO_2$ ($\Delta C_A$) and cumulative anthropogenic $CO_2$ emissions ($\Delta C_E$) for 1880–2017. **b** Same as (**a**) but for annual global-mean temperature from four observational datasets (HadCRUT4, GISTEMP, Berkeley Earth, and NOAA GlobalTemp) for 1880–2017. **c** Estimates of $\frac{1}{\alpha}$ ($=\Delta C_A/\Delta T_A$) across timescales against $\frac{1}{TCRE}$ ($=\Delta C_E/\Delta T_E$) for four observational global-mean temperature datasets with atmospheric $CO_2$ records (or anthropogenic emissions) for 1880–2017. The solid line with the shaded area between the dashed lines is the linear regression of all datasets with a slope ($p$) of 2.52 ± 0.15 ppm ppm$^{-1}$, which indicates that the carbon-concentration feedback parameter $\beta$ is 3.22 ± 0.32 GtC ppm$^{-1}$.

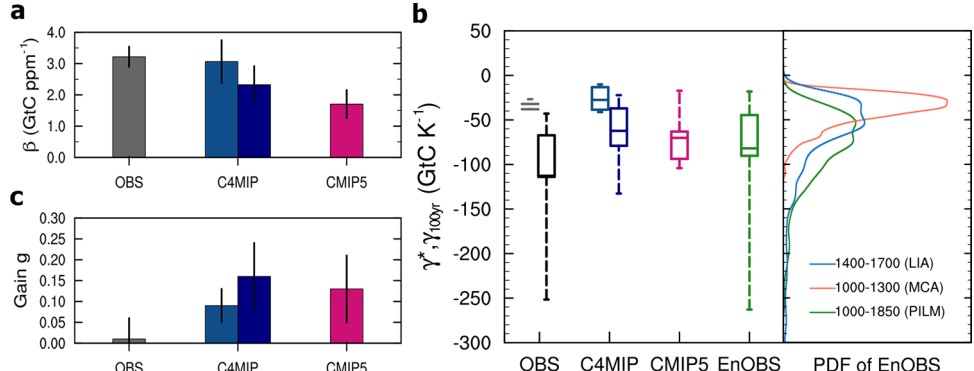

**Fig. 3 Ensemble estimates of climate–carbon cycle feedback parameters from observations and models. a** Histogram of the $\beta$ estimates (error bars for mean ± 1$\sigma$) derived from four instrumental temperature and $CO_2$ datasets for 1880–2017 (in gray), $\beta^{BGC}$ derived from 11 C$^4$MIP models[7] for 1880–2017 (in dodger blue) and for 2018–2100 (in navy blue), and $\beta^{BGC}$ derived from nine CMIP5 models[11] for the 140-year 1pctCO2 experiment (in deep pink). **b** Boxplot of the $\gamma^*$ derived from four instrumental temperature and $CO_2$ for 1880–2017 (in gray), and $\gamma_{100yr}$ estimates derived from an ensemble of 15 members (3 ice-core $CO_2$ records × 5 reconstructed temperature, in black) and from the EnOBS for 1000–1850 (in green), and $\gamma^{COU-BGC}$ derived from 11 C$^4$MIP models for 1880–2017 (in dodger blue) and for 2018–2100 (in navy blue), and $\gamma^{COU-BGC}$ derived from nine CMIP5 models (in deep pink). In **b**, the large ensemble estimates of the $\gamma_{100yr}$ from the EnOBS also provides probability distributions (right panel) of the cooler period of Little Ice Age (LIA, 1400–1700) and the warmer period of Medieval Climate Anomaly (MCA, 1000–1300), compared to those of the full preindustrial last millennium (PILM, 1000–1850). **c** Same as (**a**), but for the feedback gain factor ($g$) derived from the $\beta$ and AF estimates.

that its effects on $\beta$-feedback and $\gamma$-feedback were small (e.g., $f(\beta, \gamma)\Delta C_A \Delta T_A \approx 0$). For the 1000–1850 period with $\Delta C_E \approx 0$, Eq. (2) can be further simplified as

$$\gamma_k = -m(1+\beta)\eta_k - f(\beta, \gamma)\Delta C_A \approx -m(1+\beta)\eta_k \quad (3)$$

where $m = 2.12$ GtC ppm$^{-1}$ is a factor for converting atmospheric $CO_2$ in ppm to GtC. As the $\beta$ is found to be nearly constant during the industrial period, we assumed that the constant value is applicable to the 1000–1850 period. This is reasonable, as the error sensitivity analysis showed that the uncertainty in the estimated $\gamma$ for 1000–1850 induced from the uncertainty in $\beta$ is still smaller than those from the difference in the estimated $\gamma$ from ice-core $CO_2$ records and reconstructed temperatures (Supplementary Fig. 5). Thus, using 15 combinations of 3 ice-core $CO_2$ records[28,30,31] and 5 temperature reconstructions[16,32–35] as shown in Fig. 1 and Eq. (3) with $\beta$ of

3.22 ± 0.32 GtC ppm$^{-1}$, we estimated that the $\gamma$ at a 100-year timescale ($\gamma_{100yr}$) for 1000–1850 was −122.8 ± 60.2 GtC K$^{-1}$ (Fig. 3b and Supplementary Table 2). Furthermore, $\gamma_k$ increases with timescale, varying from −33 ± 14 GtC K$^{-1}$ on timescales of 10–70 years to −110 ± 40 GtC K$^{-1}$ over timescales of 200–800 years (Supplementary Fig. 6).

Using another set of >1500 combinations of 521 reconstructed temperature records from 1000 to 1850[16] and 3 ice-core $CO_2$ records (see "Methods", Supplementary Fig. 4), we calculated much larger ensemble (>1500) estimates (EnOBS) of $\eta_k$ and $\gamma_k$ using the Fourier analysis-based approach. Our results confirmed that $\eta_k$ systematically increased with timescale, varying from 6 ± 6 ppm K$^{-1}$ on timescales of 10–70 years to 20 ± 8 ppm K$^{-1}$ over timescales of 200–800 years due to the increased $\gamma_k$ on magnitude with timescale (Supplementary Fig. 6). The range of the EnOBS-based $\eta_{100yr}$ for 1000–1850 was 7–23 ppm K$^{-1}$ with a median of

9 ppm K$^{-1}$, consistent with the estimate of $\eta$ (1.7–21.4 ppm K$^{-1}$) by Frank et al.[16]. The resulting $\gamma_{100yr}$ of EnOBS for 1000–1850 was $-82 \pm 42$ GtC K$^{-1}$, varying from a minimum of $-260$ GtC K$^{-1}$ to a maximum of $-15$ GtC K$^{-1}$ (Fig. 3b). The EnOBS-based $\gamma_{100yr}$ over timescales of 200–800 years was $-80 \pm 50$ GtC K$^{-1}$ (Supplementary Fig. 6), which is 30% smaller than the estimate of $\gamma_{100yr}$ from 3 ice-core $CO_2$ records × 5 temperature reconstructions. These results suggest that the timescale or temporal dependance of $\eta$ over the 1000–1850[16] is largely driven by the positive feedback of terrestrial and oceanic carbon pools to climate (i.e., the $\gamma$ feedback), implying that on longer timescales, warming of the climate would cause more release of $CO_2$ into the atmosphere and in return, amplify warming.

The estimate of $\gamma$ also depends on the mean climate state (quasi-equilibrium or transient). Here we compared estimates of $\gamma$ between the warmer period of the Medieval Climate Anomaly (1000–1300) with a mean temperature anomaly of $-0.18 \pm 0.09$ K and the cooler period of the Little Ice Age (1400–1700) with a mean temperature anomaly of $-0.41 \pm 0.03$ K, and both anomalies were calculated relative to the mean global surface temperature from 1961 to 1990. We further analyzed probability distributions of the EnOBS-based $\gamma_{100yr}$ for three different periods: 1000–1850, 1000–1300, and 1400–1700. The estimates of $\gamma_{100yr}$ for the cooler period of 1400–1700 had a comparable probability distribution with the 1000–1850 period (Fig. 3b). However, the $\gamma_{100yr}$ for the warmer period of 1000–1300 showed a much narrower distribution with a less negative mean ($-38 \pm 17$ GtC K$^{-1}$, or 50% smaller) than that for 1400–1700 ($-72 \pm 56$ GtC K$^{-1}$), suggesting that $\gamma_{100yr}$ feedback is sensitive to the state of the mean climate. The higher $\gamma_{100yr}$ for 1400–1700 resulted from a colder climate and some drastic fluctuations in $CO_2$ (e.g., larger $\Delta C_A$ leads to more nonlinear feedback contribution from $f(\beta, \gamma)\Delta C_A$), especially the strong dip at ~1600 AD, while the lower $\gamma_{100yr}$ for 1000–1300 is associated with smaller variations in $CO_2$ and a warmer climate (Fig. 1). The $CO_2$ drop in the Little Ice Age (1400–1700) may have been driven not only by natural disturbances (e.g., volcanic eruptions) but also probably by human-induced land-use effects[36], which could have led to a more negative estimate of $\gamma_{100yr}$ for 1400–1700.

**Comparisons of observation-based and model-based estimates of feedback parameters.** Previous studies have noted the timescale dependence of $\eta$ (i.e., $\Delta C_A / \Delta T_A$) and the relationship between $\eta$ and the climate–carbon cycle feedback parameters ($\beta$ and $\gamma$)[10,16,37]. Here we further quantified the different timescale dependence of $\beta$ and $\gamma$. Using an observation-based constraint[18], Cox and Jones estimated that $\beta$ was between 3 and 5 GtC ppm$^{-1}$ and $\gamma$ fell between $-250$ and $-50$ GtC K$^{-1}$. For the $\beta$-feedback of land, recent studies estimated that the $CO_2$ fertilization effect on global plant biomass carbon only was $25 \pm 4$ GtC year$^{-1}$ for a 100 ppm $\Delta CO_2$ over 1980–2010[38], and for the global terrestrial C sink was $3.5 \pm 1.9$ GtC year$^{-1}$ per 100 ppm $\Delta CO_2$ over 1959 to 2010[39], which is equivalent to a land $\beta$-feedback of $1.75 \pm 0.95$ GtC ppm$^{-1}$ over the same period. Our findings indicate that observation-based (land + ocean) $\beta$ is approximately constant across different timescales at $3.22 \pm 0.32$ GtC ppm$^{-1}$ for 1880–2017. While magnitudes for $\gamma$ increased with timescale from decadal to multi-centennial, and our estimate of $\gamma_{100yr}$ had a narrower uncertainty than previous estimates, with an average of $-122.82 \pm 60.16$ GtC K$^{-1}$ from the 3 ice-core $CO_2$ records × 5 temperature reconstructions and of $-81.91 \pm 41.90$ GtC K$^{-1}$ from EnOBS for 1000–1850. Uncertainties related to the estimated $\gamma_{100yr}$ from the two datasets (15 members versus >1500 members of EnOBS) did not overlap, which may suggest the uncertainty in $\gamma_{100yr}$ was from $-180$ to $-40$ GtC K$^{-1}$.

We then compared our observation-based estimates of $\beta$ and $\gamma$ using Fourier analysis-based approach with those from Earth system models. Estimates of $\beta$ and $\gamma$ from Earth system models were calculated from three sets of model simulations: the biogeochemically-coupled (BGC), radiatively coupled (RAD), and fully coupled (COU) simulations based on the FEA approach[7,14] (see "Methods"). Eleven first-generation coupled climate–carbon cycle models (C$^4$MIP) were driven with the prescribed $CO_2$ emissions from the historical period (1860–2005) and the future period (2006–2100) under the IPCC SRES A2 scenario without land-use change[7]. Only COU and BGC simulations were conducted for the C$^4$MIP models. The subsequent analysis used nine models from phase 5 of the Coupled Model Inter-comparison Project (CMIP5) used 1% year$^{-1}$ increasing $CO_2$ for 140 years, no land-use change (or the 1pctCO$_2$ experiment)[11]. $\beta$ and $\gamma$ were estimated from three simulations (COU, BGC, and RAD) by each participating model. Because of the nonlinear feedback, the changes in size of carbon pools (land, ocean, and atmosphere) in the COU simulation were not equal to the sum of the simulated changes of those pool sizes in the RAD and BGC simulations[14,40,41]. According to "Methods", we calculated the direct $\beta$-feedback from the COU-BGC simulations ($\beta^{GC} \approx \Delta C_B^{BGC} / \Delta C_A^{BGC}$) and the direct $\gamma$-feedback from the COU-RAD simulations ($\gamma^{RAD} = \Delta C_B^{RAD} / \Delta T_A^{RAD}$) and the total $\gamma$-feedback (direct plus indirect) from the COU-BGC simulations ($\gamma^{COU-BGC} \approx (\Delta C_B^{COU} - \Delta C_B^{BGC}) / \Delta T_A^{COU}$, in theory this is the $\gamma^*$) using the FEA approach, for the observation-overlapped period of 1880–2017 and the future emission scenario of 1880–2100 for the C$^4$MIP models, and the 1pctCO$_2$ 140-year period for the CMIP5 models, respectively. We also estimated the nonlinear feedback term from the difference between COU simulations and the BGC and RAD simulations ($f(\beta, \gamma) \approx [\Delta C_B^{COU} - (\Delta C_B^{BGC} + \Delta C_B^{RAD})] / \Delta C_A^{COU} \Delta T_A^{COU}$) and its contribution to $\gamma$-feedback ($f(\beta, \gamma) \Delta C_A^{COU}$) for the CMIP5 models ("Methods").

The estimated $\beta^{BGC}$ from 11 C$^4$MIP models for 1880–2017 using the FEA approach was $3.07 \pm 0.68$ GtC ppm$^{-1}$ which is close to the observation-based $\beta$ ($3.22 \pm 0.32$ GtC ppm$^{-1}$) for the same period using historical $CO_2$ emissions as forcing. Previous studies demonstrated that the carbon-concentration feedback was strongly dependent on the growth rate of atmospheric $CO_2$ and hence on emission scenarios[10]. When extending the calculation to the future high emission scenario of the IPCC SRES A2 (close to the RCP8.5 pathway, the mean $CO_2$ growth rate was about 0.72% year$^{-1}$), we found that $\beta^{BGC}$ from C$^4$MIP models was largely reduced by 19.5% to $2.47 \pm 0.60$ GtC ppm$^{-1}$ for the period of 1880–2100 (Fig. 3a, Supplementary Table 3). As the 1pcCO$_2$ experiments of CMIP5 models were configured under a higher emission scenario (1% year$^{-1}$ of the $CO_2$ growth rate), we found that the estimated $\beta^{BGC}$ from nine CMIP5 models using the FEA approach was $1.71 \pm 0.44$ GtC ppm$^{-1}$, which is 45% smaller than the observation-based or C$^4$MIP-based $\beta$ for the historical period 1880–2017 (Fig. 3a), indicating that carbon-concentration feedbacks became smaller under higher growth rates of $CO_2$ in the climate–carbon cycle system.

The $\gamma^{COU-BGC}$ is theoretically the $\gamma^*$ in this study, as the $\gamma^{COU-BGC}$ was calculated from all feedback effects in COU simulations minus the direct $\beta$-feedback in BGC simulations. We found that the estimated $\gamma^{COU-BGC}$ from 11 C$^4$MIP models were on magnitude increased from $-27.52 \pm 11.93$ GtC K$^{-1}$ for 1880–2017 to $-52.18 \pm 26.54$ GtC K$^{-1}$ for 1880–2100, when the $\beta^{BGC}$ was decreased for 1880–2100 (Fig. 3a, b, Supplementary Table 3). The estimated $\gamma^{COU-BGC}$ from nine CMIP5 models was $-70.14 \pm 32.43$ GtC K$^{-1}$, which theoretically came from direct feedback $\gamma^{RAD}$ ($-65.08 \pm 30.74$ GtC K$^{-1}$) and the nonlinear

feedback contribution $f(\beta, \gamma)\Delta C_A$ ($-9.6 \pm 10.03$ GtC K$^{-1}$, ~15 ± 17% of the $\gamma$-feedback) under the high emission scenario of the 1pctCO$_2$ experiments (Fig. 3b, Supplementary Table 4). Without CO$_2$ impact, the estimated nonlinear feedback parameter $f(\beta, \gamma)$ was $-11.22 \pm 11.72 \times 10^{-3}$ GtC ppm$^{-1}$ K$^{-1}$ (Supplementary Table 4). On the other hand, as the observation-based $\gamma_k$ for 1000–1850 and the model-based $\gamma^{\text{COU-BGC}}$ refer to different periods and timescales which exaggerates their differences owing to the timescale dependency of $\gamma$-feedback, it should be noted that there is limited comparability between them.

To further assess whether $\beta$ varies with timescale, we estimated $\beta_k^{\text{BGC}}$ and $\gamma_k^{\text{RAD}}$ by applying the FEA approach to the simulations from the nine CMIP5 models from inter-annual to centennial timescales (see "Methods" and Supplementary Texts 1, 2). The results supported that $\beta$-feedback was approximately constant across different timescales while $\gamma$-feedback increased significantly with timescale for most models (Supplementary Fig. 7a, b). Similar timescale dependences were also found for $\beta_k^{\text{BGC}}$ and $\gamma_k^{\text{RAD}}$ by applying the FEA approach to the simulations from C$^4$MIP models (Supplementary Fig. 8).

**Estimates of feedback gain ($g$) for 1880–2017 and high CO$_2$ emissions scenarios.** For further comparisons, we estimated the observational and model-based airborne fraction (AF) of cumulative CO$_2$ emissions (AF $= \Delta C_A / \Delta C_E$) and climate–carbon cycle feedback gain factor ($g$) across timescales ($g = 1 - 1/[\text{AF}(1 + \beta)]$, see "Methods"). We estimated AF$_k$ over different timescales from amplitudes of atmospheric CO$_2$ concentration and cumulative CO$_2$ emissions using Fourier analysis (Fig. 2a), then calculated the average and standard deviation of AF from AF$_k$ over timescales (Supplementary Fig. 7). We showed that the observation-based AF for the period 1880–2017 was nearly constant across timescales with an average of 0.40 ± 0.05 (Supplementary Fig. 7c), suggesting that about 60% of CO$_2$ emission was taken up by land and ocean. Previous studies found the relationship between AF and the two feedback parameters: AF $= 1/(1 + \beta + \alpha\gamma^*)$[10]. As for the period 1880–2017, we found a relatively smaller contribution of $\alpha\gamma^*$ (~0.05 GtC GtC$^{-1}$) from $\alpha$ (0.005 K GtC$^{-1}$) and $\gamma^*$ ($-10.9$ GtC K$^{-1}$) compared to $\beta$ (~1.52 GtC GtC$^{-1}$) from the observational estimates, i.e., $\beta \gg \alpha\gamma^*$, suggesting that the carbon-concentration feedback ($\beta$) dominated the relatively stable cumulative airborne fraction over the industrial period.

Further analysis showed that the observation-based feedback gain ($g$) was very small (0.01 ± 0.05) for 1880–2017 (Fig. 3c). We showed that C$^4$MIP-based AF for 1880–2017 was also nearly constant (0.45 ± 0.06) across different timescales (Supplementary Fig. 7c), slightly larger than the observed estimate. However, the C$^4$MIP-based $g$ for the same period 1880–2017 was 0.09 ± 0.04, larger than the observational value by about an order of magnitude (Fig. 3c). As a result, the observation-based and C$^4$MIP-based feedback amplification $G$ ($G = 1/(1-g)$, see "Methods") are 1.01 ± 0.05 and 1.10 ± 0.04, respectively, suggesting the modeled amplification effect is about 9 ± 7% larger. Under high emission scenarios, the C$^4$MIP-based $g$ increased to 0.15 ± 0.08 with increased AF (0.56 ± 0.09) for 1880–2100, and the CMIP5-based $g$ increased to 0.13 ± 0.08 for the 1pctCO$_2$ (Fig. 3c), which are much higher than the observation-based $g$.

## Discussion

This study expanded the traditional climate–carbon cycle feedback framework[7,10,14,19] by including a nonlinear feedback term (see Eq. (1)). Using CMIP5 modeling experiments under a scenario of high CO$_2$ growth rate (1% yr$^{-1}$), we found that the contribution of the nonlinear term to land and ocean $\beta$-feedback is relatively small (3 ± 3%), while its contribution to land and

ocean $\gamma$-feedback is 15 ± 23%. These estimates are noticeably smaller than those simulated from previous modeling studies on land and ocean (20% for $\beta$ and 45% for $\gamma$), or on ocean only (6% for $\beta$ and 60% for $\gamma$)[10,14] (see also Supplementary Text 3 for detailed discussion). However, the estimated nonlinear contributions vary significantly across CMIP5 models (0.2–9.6% for $\beta$ and 0.8–45% for $\gamma$, see Supplementary Table 4), which suggests a large uncertainty in the modeled nonlinear feedback among the advanced Earth system models.

This study also stated the relationship among the four commonly used quantities ($\beta$, $\gamma$, $\alpha$, and TCRE) which is consistent to some similar relationships as stated by previous carbon–climate feedback studies[10,21]. Using historical temperature and CO$_2$ records over 1880–2017, we estimated that the carbon-concentration feedback parameter $\beta$ is 3.22 ± 0.32 GtC ppm$^{-1}$ and the climate–carbon cycle feedback gain factor $g$ is 0.01 ± 0.05. These estimates were nearly constant across inter-annual to decadal timescales. On the other hand, we also found that the $\gamma$-feedback parameter increased with timescale from $-33 \pm 14$ GtC K$^{-1}$ on a decadal scale to $-122 \pm 60$ GtC K$^{-1}$ on a centennial scale based on reconstructions over 1000–1850. Furthermore, the estimated climate amplification from carbon–climate feedback based on observations in this study is much smaller than the previous estimates by IPCC reports based on model simulations under high emission scenarios[12,13]. Our results based on observations have significant implications for understanding the strength of the climate–carbon cycle feedback and the allowable CO$_2$ emissions to mitigate future climate change. For example, the allowable emissions based on Earth system models for a 68% probability of limiting warming to 1.5 °C above the preindustrial level by ~2050 as stipulated in the Paris Agreement[42], are ~115 GtC (or 420 Gt CO$_2$)[43], using the much smaller feedback gain of 0.01 ± 0.05 than the C$^4$MIP-based estimates during the same period, we estimate that the allowable emissions would be 9 ± 7% more, or 125 ± 8 GtC.

In this study, the observation-based $\beta$ and AF were found to be nearly constant (with <10% change), which together determined the nearly constant feedback gain ($g$). This can be explained by the linear system in response to exponential increase of forcing (LinExp) theory[44,45], in which the carbon–climate system over the industrial period can be approximated as a linear system of the carbon cycle forced by exponentially growing CO$_2$ emissions with $y = 0.27e^{0.018t}$ for the 1850–2017 (Supplementary Fig. 9a, b), then all ratios of responses to forcings are constant[44]. Using the simulations over the historical period (1901–2010) from 14 terrestrial ecosystem models forced by observational climate, land-use change, and atmospheric CO$_2$[46], we showed that when excluding the effects of climate change and land use change, both annual global GPP and cumulative land carbon sink increased exponentially in response to exponentially growing atmospheric CO$_2$ over 1901–2010 (Supplementary Fig. 9c, d). Therefore, the ratio of the exponential increase in carbon uptake and the exponential increase in CO$_2$ sustained a nearly constant value of $\beta$ over the historical period.

The timescale dependency of the $\gamma$-feedback can be used to reconcile the diverging estimates of $\gamma$ parameters in the previous studies. The estimated $\gamma$ at centennial timescales for the pre-industrial period ($-180$ to $-40$ GtC K$^{-1}$) was found to be much more negative than the value of $\gamma^*$ at multi-decadal timescales ($-14$ to $-7$ GtC K$^{-1}$) for the industrial period, implying that the climate sensitivity of carbon cycle depends on the base climate state. If we allow the present transient climate system to reach an equilibrium state, the estimated climate sensitivity of the carbon cycle would be more negative. Furthermore, the $\gamma$ (or $\gamma^*$)-feedback also depends on CO$_2$ emissions scenario. By comparing observation-based $\gamma^*$ and model-based $\gamma^{\text{COU-BGC}}$ from C$^4$MIP

models for the industrial period (1880–2017) with the model-based $\gamma^{COU-BGC}$ estimate for the future period with higher $CO_2$ emissions, we showed that under the conditions of the higher $CO_2$ growth and warmer climate states, the $\gamma$-feedback and the nonlinear feedback contribution to the $\gamma^*$ became more negative, while the $\beta$-feedback decreased (Fig. 3, Supplementary Table 3). The contribution of the nonlinear carbon–climate feedback to total carbon–climate feedback was estimated to be $15 \pm 17\%$ for the 1pctCO2 high emission scenario as used in the CMIP5 study. For the ocean, the nonlinear feedback may become greater due to the decreased downward carbon transport to the deep ocean owing to reduced overturning in a warmer climate[14]. The $\gamma$ encapsulates both the direct effect of warming on plant and soil respiration[47,48] and sea water solubility ($\gamma$ gets more negative), but also the indirect effect of warming on biological productivity and phenology[49] and on stratification/reduction of the overturning[14,41] ($\gamma$ gets less negative). How the $\gamma$-feedback will change in a warmer climate with high $CO_2$ emissions depends on the contributions by these competing processes, and the uncertainties in the earth system models used for the analysis[4,50]. Overall, the $\gamma$-feedback at equilibrium under a high emissions scenario would become much more negative than the $\gamma$-feedback for the industrial or preindustrial periods, and likely play the dominant role in the positive net climate–carbon cycle feedback.

Uncertainties in the estimates of $\beta$, $\gamma$, and corresponding $g$ arise from the errors in observations, model uncertainties, methodology, and study periods chosen. First, we did not consider separately the contributions of non-$CO_2$ greenhouse gases (e.g., $CH_4$, $N_2O$, $O_3$) from those of $CO_2$ on climate variation, which may have led to some biases in the estimates of $\beta$ and $\gamma$. Second, the overestimation of $C^4MIP$-based $g$ compared to the observation-based $g$ for the same period 1880–2017, might be due to several reasons in model descriptions, e.g., poor descriptions of terrestrial ecosystem processes (carbon pools, carbon–water coupling, soil respiration, vegetation phenology, and ecosystem climate adaptation, etc.), omission of land-use change, and biases in model initial climate/carbon pool base-states, etc. Third, errors could arise from the two approaches (the Fourier analysis approach and the FEA approach) for estimating observation-based $\beta$ and $\gamma^*$ or model-based $\beta^{BGC}$ and $\gamma^{COU-BGC}$ and corresponding $g$. Comparative analysis showed that the ensemble means of $C^4MIP$-based $\beta^{BGC}$ and $\gamma^{COU-BGC}$ using the FEA approach are consistent with those of $C^4MIP$-based $\beta$ and $\gamma^*$ using the Fourier analysis approach for the 1880–2017 period, indicating a relatively small difference of $C^4MIP$-based $g$ between using the two approaches (Supplementary Fig. 11).

In addition, although the $\beta$-feedback was found to be stable over the industrial period with the approximately exponential growth of $CO_2$, applying the same $\beta$ value to the preindustrial period may result in bias in estimated $\gamma$ for the preindustrial period. Sensitivity analysis found that an overestimation (or underestimation) of 50% in $\beta$ would result in an underestimation (or overestimation) of about 30% in $\gamma_k$ at timescales from 10 to 1000 years for the 1000–1850 period (Supplementary Fig. 5). This change of 30% ($\sim$25 GtC $K^{-1}$) in $\gamma_{100yr}$ is still smaller than the uncertainty of $\gamma_{100yr}$ (a standard deviation of 41.90 GtC $K^{-1}$) that was mainly caused by the large divergences in the three ice-core $CO_2$ records and reconstructed temperatures. Furthermore, this study ignored the contribution of early land-use change to climate change ($\zeta_{100yr} \approx 0$) during 1000–1850, therefore may have overestimated the $\gamma_{100yr}$, e.g., for the Little Ice Age (1400–1700). The observed atmospheric $CO_2$ drop in the 1400–1700 period occurred because of the cooling-induced increase in terrestrial C uptake[27], but it can also be partly explained by enhanced forest restoration from reduced land-use change as a result of aban-

donment of agricultural land from the collapse of native population in the Americas during 1500–1650[51]. Despite these uncertainties, we demonstrated the nature of the timescale dependency of $\gamma$ on climate–carbon cycle feedback and important implications for observational constraints on earth system models for projecting future climate changes.

## Methods

**Anthropogenic $CO_2$ emission data.** We calculated the global annual total anthropogenic $CO_2$ emission flux ($F_E$) as the sum of annual emission fluxes from fossil fuel combustion and industrial processes ($F_{FF}$) and from land use and land cover changes ($F_{LUC}$) by human activity covering the period 1850–2017, both of which were obtained from the Global Carbon Project's annual global carbon budget report[1]. The $F_{FF}$ (uncertainty of $\pm 5\%$ for a $\pm 1\sigma$ (68%) confidence interval) was estimated by ref. [22] and the $F_{LUC}$ (uncertainty of $\pm 0.7$ GtC year$^{-1}$ representing a $\pm 1\sigma$ confidence interval) was averaged from emission estimates based on two bookkeeping models by ref. [23] and ref. [24]. We then calculated the cumulative annual anthropogenic $CO_2$ emission: $C_E(t) = \int_0^t F_E dt = \int_0^t (F_{FF} + F_{LUC})dt$. Units were converted from GtC to ppm by dividing $m$ (=2.12 GtC ppm$^{-1}$) for comparison with atmospheric $CO_2$ concentration[52]. The conversion factor $m$ was also used for unit conversions of $\beta$, $\gamma$, $\alpha$, and $\eta$ parameters.

**Instrumental and ice-core reconstructed atmospheric $CO_2$ records.** Global annual atmospheric $CO_2$ concentration over 1850–2017 was reconstructed from a combination of ice-core $CO_2$ records and instrumental $CO_2$ measurements since 1950s[53,54]. Information about the inter-annual variability in the $CO_2$ data covering 1850–1940s from ice-core $CO_2$ would be lost, as ice cores smooth atmospheric $CO_2$ records by firn diffusion, which may lead to some biases in the timescale analysis.

The highly resolved $CO_2$ covering the preindustrial period of 1000–1850 was compiled from ice-core records in Antarctica including three datasets at Law Dome[30,55], WAIS Divide[28], and Dronning Maud Land[31], respectively. As the three records have different resolutions, $CO_2$ values were resampled to the nearest calendar year and then smoothed using 30-year splines (Fig. 1a). However, large variations remain among the three records of $CO_2$ from ice-core (see Fig. 1a and Supplementary Fig. 3a).

**Instrumental and reconstructed temperature datasets.** All available instrumental temperature datasets including HadCRUT4[56] and Berkeley Earth[57] over 1850–2017, and GISTEMP[58] and NOAA GlobalTemp[59] over 1880–2017 were used. Global annual temperature anomalies were calculated by area-weighted averaging over both land (2 m air temperature) and ocean (sea surface temperature) minus the mean surface air temperature of land and ocean for the period from 1961 to 1990.

Five reconstructions of Northern Hemispheric temperature were obtained from four groups which were abbreviated as PAGES2k[32], Frank2010[16], Mann2009[34], MannEIV[33], and Moberg2005[35] in this study (Fig. 1b). All temperature reconstructions were smoothed with 5-year splines, and adjusted to removed-means with respect to 1961–1990. Northern Hemispheric-mean temperature had been shown to be highly representative for global-mean temperature variations[16]. The ensemble estimates of 521 calibrated temperature reconstructions were obtained from the Frank2010[16], in which all amplitude and variability on >30-year timescales are well preserved for uncertainty analysis. The ensemble estimates were recalibrated by nine available datasets of Northern Hemispheric-mean temperature reconstructions using a reconstructing technique based on a state-space time series and Kalman filter algorithm[60]. The ensemble means (Fig. 1b) and individual temperature reconstructions (Supplementary Fig. 4a) by Frank2010 are used for a large ensemble of >1500 estimates (EnOBS) of the $\eta$ and $\gamma$ over the preindustrial period. Our estimate based on Fourier analysis shows that uncertainty in the $\eta$ on centennial timescales (Supplementary Fig. 4b) was consistent with the estimate by Frank2010 which was based on a lag-regression method (searching highest correlations) of $CO_2$ and temperature reconstructions which were smoothed with spline using a range of cutoff time (50, 75,…, 200 years)[16].

**Theoretical analysis for the climate–carbon cycle feedback.** Following previous analysis[7,10], the increase from anthropogenic $CO_2$ emissions ($\Delta C_E$) is the sum of changes in the three stores,

$$\Delta C_E = \Delta C_A + \Delta C_L + \Delta C_O \qquad (4)$$

where $\Delta C_A$, $\Delta C_L$, and $\Delta C_O$ are the changes in carbon storage on atmosphere, land, and ocean, respectively, over a time period ($\Delta t$) evolved from the reference climate state. For the analysis for industrial period, we calculated $\Delta C_E$, $\Delta C_A$, $\Delta C_L$, and $\Delta C_O$ (in a unit of GtC) during $\Delta t$ since 1880. Note that change in atmospheric $CO_2$ concentration in a unit of ppm can be converted to GtC by a factor of $m$ ($\sim$2.12 GtC ppm$^{-1}$)[52]. Within the climate–carbon cycle feedback system, following the linear carbon cycle feedback framework by refs. [7,10,19] and the nonlinearity of carbon cycle feedback discovered by ref. [14], we consider the changes in land and

ocean carbon are the combined effects of carbon-concentration feedback ($\beta$) and carbon–climate feedback ($\gamma$) and the nonlinear feedback between climate and atmospheric $CO_2$,

$$\begin{cases} \Delta C_L = \beta_L \Delta C_A + \gamma_L \Delta T_A + f_L(\beta_L, \gamma_L)\Delta C_A \Delta T_A \\ \Delta C_O = \beta_O \Delta C_A + \gamma_O \Delta T_A + f_O(\beta_O, \gamma_O)\Delta C_A \Delta T_A \end{cases} \quad (5)$$

where $\beta_L$ and $\beta_O$ are the $\beta$-feedback on land and ocean, respectively, $\gamma_L$ and $\gamma_O$ being the $\gamma$-feedback on land and ocean, respectively. $\beta_L$ and $\beta_O$ are in a unit of $GtC\,GtC^{-1}$ and can be converted to a commonly used unit of $GtC\,ppm^{-1}$ by multiplying $m$ ($=2.12$ $GtC\,ppm^{-1}$). $\gamma_L$ and $\gamma_O$ are in a unit of $GtC\,K^{-1}$. $f_L(\beta_L, \gamma_L)$ and $f_O(\beta_O, \gamma_O)$ are the nonlinear feedback ($GtC\,ppm^{-1}\,K^{-1}$) as a function of $\beta$ and $\gamma$ feedback parameters on land and ocean, respectively. Combining Eqs. (4) and (5), we have

$$\Delta C_E = \Delta C_A + \beta \Delta C_A + \gamma \Delta T_A + f(\beta, \gamma)\Delta C_A \Delta T_A = (1+\beta)\Delta C_A + \gamma^* \Delta T_A \quad (6)$$

where the total $\beta$-feedback is $\beta_L + \beta_O$, $\gamma$-feedback is $\gamma_L + \gamma_O$, and $f(\beta, \gamma)$-nonlinear feedback is $f_L(\beta_L, \gamma_L) + f_O(\beta_O, \gamma_O)$. As defined by ref. [14], when assuming the carbon stock in biosphere ($C_B = C_L + C_O$) at the reference climate state as a function of climate and $CO_2$: $C_B = F(C_A, T_A)$, then $\beta$, $\gamma$, and $f(\beta, \gamma)$ can be expressed as the 1st order and 2nd order coefficients of the Taylor series of $C_B$ since the initial time ($t=0$): $\beta = \frac{\partial F}{\partial C_A}\big|_0$, $\gamma = \frac{\partial F}{\partial T_A}\big|_0$, and $f(\beta, \gamma) = \frac{\partial^2 F}{\partial C_A \partial T_A}\big|_0 + \frac{1}{2}\frac{\partial^2 F}{\partial C_A^2}\big|_0 \frac{\Delta C_A}{\Delta T_A} + \frac{1}{2}\frac{\partial^2 F}{\partial T_A^2}\big|_0 \frac{\Delta T_A}{\Delta C_A} + R_3$. The nonlinear feedback $f(\beta, \gamma)$ in this study represents the 2nd and high-order terms of the Taylor expansion. As previous studies mainly focused on the nonlinearity of the carbon–climate ($\gamma$-) feedback[10,14,41], in this study, we combined the $\gamma$-feedback and the atmospheric $CO_2$ change's impacts on the nonlinear feedback as $\gamma^* = \gamma + f(\beta, \gamma)\Delta C_A$. From this definition, the $\gamma^* \approx \gamma$, when $\Delta C_A \approx 0$.

The Eq. (6) can be rewritten as by dividing both sides by $\Delta T_A$,

$$\frac{1}{TCRE} = (1+\beta)\frac{1}{\alpha} + \gamma^* \quad (7)$$

where $\alpha \equiv \Delta T_A / \Delta C_A$ is the sensitivity of climate to atmospheric $CO_2$ (in a unit of $K\,GtC^{-1}$), which represents the change in temperature in response to a change in $CO_2$ concentration[7,19]. The $\alpha$ is a useful measure to quantify the feedbacks between climate and carbon cycle over both preindustrial and industrial periods. The $TCRE \equiv \Delta T_A / \Delta C_E$ is transient climate response to cumulative $CO_2$ emission (in a unit of $K\,GtC^{-1}$)[20], which quantifies the ratio of change in temperature to cumulative carbon emissions, providing another useful measure to estimate the total allowable emissions for a given temperature change, as there is a near-linear relationship between cumulative $CO_2$ emissions and global temperature change[20].

In this study, we notated $\frac{1}{\alpha}$ as $\eta$, i.e., $\eta \equiv \frac{1}{\alpha}$. In some previous studies, the $\eta$ was also defined as the sensitivity of atmospheric $CO_2$ to climate for cases of no anthropogenic $CO_2$ emission involved in the climate–carbon cycle system, e.g., over the preindustrial last millennium[16–18,37,61]. But for the industrial period, this definition expressing the sensitivity of $CO_2$ to climate could be physically meaningless, as the $CO_2$ increase over this period is not only due to the climatic impact on carbon stores, but also is primarily driven by the increasing anthropogenic $CO_2$ emissions. During the preindustrial period (1000–1850), the atmospheric $CO_2$ remained very stable ($280 \pm 8$ ppm), with only small anthropogenic $CO_2$ emissions from the land-use change and negligible anthropogenic emissions from fossil fuels, and those anthropogenic emissions had little induced global warming, thus[37]

$$\gamma_k = -m(1+\beta)\eta_k - f(\beta, \gamma)\Delta C_A \quad (8)$$

where $m = 2.12$ $GtC\,ppm^{-1}$ is a factor for converting units in ppm to GtC.

**Feedback gain factor g.** The gain factor ($g$) of the climate–carbon cycle feedback[7,10,19] is expressed as

$$g = \frac{-\gamma^*}{(1+\beta)}\frac{\Delta T_A}{\Delta C_A} = \frac{-\gamma^* \alpha}{(1+\beta)} \quad (9)$$

Substituting Eq. (7) into Eq. (9) gives

$$g = 1 - \frac{\alpha}{TCRE}\frac{1}{(1+\beta)} = 1 - \frac{1}{AF(1+\beta)} \quad (10)$$

where $AF = \Delta C_A / \Delta C_E$ is the airborne fraction of cumulative $CO_2$ emissions. Thus, we can further estimate the amplification factor $G = 1/(1-g) = AF(1+\beta)$. For a net positive climate–carbon cycle feedback, $g > 0$ and $G > 1$.

**Estimating $\beta$ and $\gamma$ across different timescales from observations.** Over the industrial $CO_2$ emission forcing period (1850–2017), as both changes in anthropogenic $CO_2$ emission ($\Delta C_E$) and atmospheric $CO_2$ concentration ($\Delta C_A$) are accumulated over time intervals ($\Delta t$). From Eq. (6) with the three unknows $\beta$, $\gamma$, and $f(\beta, \gamma)$, we can have

$$\frac{\Delta C_E}{\Delta t} = \frac{\Delta C_A}{\Delta t} + \beta \frac{\Delta C_A}{\Delta t} + \gamma \frac{\Delta T_A}{\Delta t} + f(\beta, \gamma)\Delta C_A \frac{\Delta T_A}{\Delta t} \quad (11)$$

when $\Delta t \to 0$, then $\Delta C_A \to 0$, the nonlinear feedback contribution is close to zero, i.e., $f(\beta, \gamma)\Delta C_A \to 0$, hence $\gamma^* \approx \gamma$. Therefore, rewriting Eq. (11) into partial

differential form ($\Delta t \to 0$) is

$$\frac{\partial C_E}{\partial t} = (1+\beta)\frac{\partial C_A}{\partial t} + \gamma^* \frac{\partial T_A}{\partial t} \quad (12)$$

As the year-to-year or decade-to-decade variations of $C_E$, $C_A$, and $T_A$ in the climate system are mostly driven by more than one factor (e.g., the El Niño-Southern Oscillation) at different timescales[62,63]. Based on the theory of Fourier analysis, fluctuations of these variables with time (including variability and trend) thus can be seen as the wave superposition over different frequencies[64].

In this study, we then apply the Fourier analysis to the nonlinear climate–carbon cycle feedback framework to estimate $\beta$ and $\gamma$ across different timescales from observations. Expressing the $C_E$, $C_A$, and $T_A$ as functions of time $t$, writing them as the sum of periodic basis functions at different frequencies (Supplementary Fig. 1); this can be written as

$$\begin{cases} C_E(t) = \sum_k a_k \sin(\omega_k t + \varphi_{1,k}) \\ C_A(t) = \sum_k b_k \sin(\omega_k t + \varphi_{2,k}) \\ T_A(t) = \sum_k c_k \sin(\omega_k t + \varphi_{3,k}) \end{cases} \quad (13)$$

where $k$ is wavenumber (or with respect to timescale), $\omega_k = 2\pi \frac{k}{N}$ is the angular frequency, and $a_k$, $b_k$, and $a_k$ are amplitudes of $C_E$, $C_A$, and $T_A$, respectively, and $N$ is the time period in years. The time variable $t$ varies from 0 to $N$. Then,

$$\begin{cases} \frac{\partial C_E}{\partial t} = \sum_k a_k \omega_k \cos(\omega_k t + \varphi_{1,k}) \\ \frac{\partial C_A}{\partial t} = \sum_k b_k \omega_k \cos(\omega_k t + \varphi_{2,k}) \\ \frac{\partial T_A}{\partial t} = \sum_k c_k \omega_k \cos(\omega_k t + \varphi_{3,k}) \end{cases} \quad (14)$$

We can substitute Eq. (14) into Eq. (12), which yields,

$$\sum_k h_k (a_k - (1+\beta)b_k - \gamma^* c_k)\omega_k \sin(\omega_k t + \xi_k) = 0 \quad (15)$$

where $\xi_k = \arctan\left(-\frac{\cos\varphi_{1,k} + \cos\varphi_{2,k} + \cos\varphi_{3,k}}{\sin\varphi_{1,k} + \sin\varphi_{2,k} + \sin\varphi_{3,k}}\right)$, and $h_k = \sqrt{(\sin\varphi_{1,k} + \sin\varphi_{2,k} + \sin\varphi_{3,k})^2 + (\cos\varphi_{1,k} + \cos\varphi_{2,k} + \cos\varphi_{3,k})^2} \neq 0$. We find that Eq. (15) only holds when

$$a_k - (1+\beta)b_k - \gamma^* c_k = 0 \quad (16)$$

Comparing Eq. (16) with Eq. (7), we notate $\frac{1}{TCRE}$ as $\zeta$, i.e., $\zeta \equiv \frac{1}{TCRE}$, and $\frac{1}{\alpha}$ as $\eta$, then have the solutions of $\zeta$ and $\eta$ over timescales,

$$\zeta_k = \frac{a_k}{c_k} \quad (17)$$

$$\eta_k = \frac{b_k}{c_k} \quad (18)$$

The $\eta$ can be estimated from ratios of the amplitudes in $C_E$ to the amplitudes in $T_A$ at any given timescale. Similarly, the $\zeta$ can be estimated from the amplitudes in $C_E$ and $T_A$. Fourier analysis based on fast Fourier transform (FFT)[65] is used for estimating amplitudes for each time series of $C_E$, $C_A$, and $T_A$ over industrial or preindustrial periods (see Supplementary Fig. 1). From Eqs. (16)–(18), we have

$$\zeta_k = (1+\beta)\eta_k + \gamma^* \quad (19)$$

From Eq. (19) or Eq. (2) and Fig. 2c, we find that the $\beta$ can be estimated from a linear regression of $\zeta_k$ and $\eta_k$ with residual errors ($\in$)

$$\zeta_k = p\eta_k + \epsilon \quad (20)$$

where $p = \frac{cov(\zeta_k, \eta_k)}{var(\eta_k)}$, then $\beta = p - 1$. The $\gamma^*$ on different timescales ($\gamma_k^*$) for the industrial period can be calculated by Eq. (20) with input from estimates of $\zeta_k$ (or $TCRE_k^{-1}$) and $\eta_k$ (or $\alpha_k^{-1}$) for a given timescale. If considering that the $\beta$-feedback of the climate–carbon cycle system is the same over both the industrial and preindustrial periods on long-term timescales, e.g., of 100 year, assuming $\zeta_{100yr} \approx 0$, from Eq. (8) we have

$$\gamma_{100yr} = -m(1+\beta)\eta_{100yr} \quad (21)$$

where the $\eta_{100yr}$ is estimated on the 100-year timescale from temperature reconstructions and $CO_2$ ice-core records over 1000–1850.

**Estimating $\beta$ and $\gamma$ and $f(\beta, \gamma)$ from C$^4$MIP and CMIP5 simulations.** We followed the FEA approach based on a Taylor series expansion as defined in Friedlingstein et al.[7], Arora et al.[11], and Schwinger et al.[14] of estimating $\beta$ and $\gamma$ and over a time period of $N$ years from the biogeochemically-coupled (BGC) and radiatively coupled (RAD) CMIP5 simulations, respectively, using the COU-BGC, COU-RAD experiment pairs for $\Delta t = N$. In this study, we used the COU-BGC approach:

$$\beta^{BGC} = \frac{\Delta C_B^{BGC}\Delta T_A^{COU} - \Delta C_B^{COU}\Delta T_A^{BGC}}{\Delta C_A^{BGC}(\Delta T_A^{COU} - \Delta T_A^{BGC})} \approx \frac{\Delta C_B^{BGC}}{\Delta C_A^{BGC}} \quad (22)$$

$$\gamma^{COU-BGC} = \frac{\Delta C_B^{COU} - \Delta C_B^{BGC}}{\Delta T_A^{COU} - \Delta T_A^{BGC}} \approx \frac{\Delta C_B^{COU} - \Delta C_B^{BGC}}{\Delta T_A^{COU}} \quad (23)$$

and the COU-RAD approach:

$$\beta^{COU-RAD} = \frac{\Delta C_B^{RAD}\Delta T_A^{COU} - \Delta C_B^{COU}\Delta T_A^{RAD}}{\Delta C_A^{COU}\Delta T_A^{RAD}} \approx \frac{\Delta C_B^{COU} - \Delta C_B^{RAD}}{\Delta C_A^{COU}} \quad (24)$$

$$\gamma^{RAD} = \frac{\Delta C_B^{RAD}}{\Delta T_A^{RAD}} \quad (25)$$

where $\Delta C_A^{COU} = \Delta C_A^{BGC}$ for CMIP5 or C[4]MIP simulations. To calculate the non-linear feedback parameter $f(\beta, \gamma)$, we also defined:

$$f(\beta,\gamma) = \frac{\Delta C_B^{COU} - (\Delta C_B^{BGC} + \Delta C_B^{RAD})}{\Delta C_A^{COU}(\Delta T_A^{COU} - \Delta T_A^{BGC})} \approx \frac{\Delta C_B^{COU} - (\Delta C_B^{BGC} + \Delta C_B^{RAD})}{\Delta C_A^{COU}\Delta T_A^{COU}} \quad (26)$$

We set the reference time at 1880 and then calculated the $\beta^{BGC}$, $\gamma^{RAD}$, $\gamma^{COU-BGC}$, and $f(\beta, \gamma)$ for the observation-overlapped period of 1880–2017 and the future emission scenario of 1880–2100 for the C[4]MIP models, and the 1pctCO$_2$ 140-year period for the CMIP5 models, respectively.

**A box model for diagnosing the climate–carbon cycle feedback parameters**. The coupled climate–carbon cycle system can be simplified as the combination of variations in temperature ($T_A = T_0 + \Delta T_A$) and CO$_2$ ($C_A = C_0 + \Delta C_A$) over time intervals ($\Delta t$). The change in $T_A$ is assumed to increase logarithmically with CO$_2$[17,66],

$$\Delta T_A = \frac{s}{\ln(2)} * \ln\left(\frac{C_A}{C_0}\right) + \varepsilon \quad (27)$$

where $s$ is the impact of CO$_2$ on the temperature that is suggested to be 1.5–4.5 K by the IPCC reports[67], and $\varepsilon$ is the residual term from climate internal variability. From Eq. (6),

$$\Delta C_A = \frac{\Delta C_E - \gamma^*\Delta T_A}{1 + \beta} \quad (28)$$

We then have the three parameters box model with input from CO$_2$ emissions,

$$C_A = C_0 + \frac{\Delta C_E - \gamma^*(T_A - T_0)}{1 + \beta} \quad (29)$$

$$T_A = T_0 + \frac{s}{\ln(2)}\ln\left(\frac{C_A}{C_0}\right) + \varepsilon \quad (30)$$

where $C_0$ and $T_0$ are initial values at the first year, here referenced to 1850. To validate the estimated $\beta$ and $\gamma^*$ for the industrial period, we applied this box model to predict temperature and CO$_2$ over 1850–2017 using annual cumulated CO$_2$ emissions, by setting $\beta = 3.22$ GtC ppm$^{-1} = 1.52$ GtC GtC$^{-1}$, $\gamma^* = -10.9$ GtC K$^{-1}$, $s = 3 \pm 1.5$ K and $\varepsilon$ being the detrended $T_A$ anomaly time series from HadCRUT4 (Supplementary Fig. 2). Result shows that the increasing trend in predicted annual CO$_2$ is very close to the observation ($R^2 = 0.99$, RMSE = 3.5 ppm), while predicted temperature has a larger trend since 1980s ($R^2 = 0.96$, RMSE = 0.17 K), compared to observed records (Supplementary Fig. 2).

**Uncertainty**. We quantified the uncertainty by a $\pm 1\sigma$ (standard deviation) that represents a 68% confidence interval. The uncertainty in $C_E$ (Fig. 1) resulted from the uncertainty in $F_E$ ($\sigma_{F_E}$) that was calculated by $\sigma_{F_E} = \sqrt{\sigma_{F_{FF}}^2 + \sigma_{F_{LUC}}^2}$. The uncertainty in $F_{FF}$ ($\sigma_{F_{FF}}$) was $\pm 5\%$ of $F_{FF}$[22]. This is consistent with a more detailed recent analysis of uncertainty for $F_{FF}$[68]. The uncertainty in $F_{LUC}$ ($\sigma_{F_{LUC}}$) was $\pm 0.7$ GtC year$^{-1}$ at each year[1]. To estimate the $\beta$ and its uncertainty ($\sigma_\beta$), we first estimated the $\beta_i$ and $\sigma_{\beta,i}$ ($i = 1, \cdots, 4$) for each combination of 1 CO$_2 \times 1$ CO$_2$ emission $\times$ 4 temperature datasets (HadCRUT4, GISTEMP, Berkeley Earth, and NOAA GlobalTemp) that included uncertainty on $C_E$ and the regression. We then estimated the mean value of $\beta$ by $\overline{\beta} = \frac{1}{4}\sum\beta_i$ and the $\sigma_\beta$ by $\sigma_\beta = \frac{1}{4}\sqrt{\sum\sigma_{\beta_i}^2}$ (Supplementary Table 1). Similarly, we estimated uncertainty in $\gamma$ ($\sigma_\gamma$) for the industrial period at a given timescale by $\sigma_\gamma = \frac{1}{4}\sqrt{\sum\sigma_{\gamma_i}^2}$. As the uncertainty in $\gamma$ ($\sigma_\gamma$) for the preindustrial period arises from uncertainties in ice-core CO$_2$ records and temperature reconstructions, we estimated the $\sigma_\gamma$ over 1000–1850 or 1400–1700, or 1000–1300 periods at a given timescale by calculating the standard deviation of the ensemble of $\gamma$ from 3 ice-core CO$_2$ records $\times$ 5 temperature reconstructions (or the EnOBS): $\sigma_\gamma = \sqrt{\frac{1}{n-1}\sum_{i=1}^{n-1}(\gamma_i - \overline{\gamma})^2}$ (Supplementary Table 2, Fig. 3c, d). The estimated uncertainties in $\beta$ and $\gamma$ for C[4]MIP or CMIP5 models were calculated as the standard deviations of their ensembles of $\beta$ and $\gamma$ for each model (Supplementary Tables 3, 4). As $G = \frac{1}{1-g} \approx 1 + g$, uncertainty in $G$ ($\sigma_G$) was $\sigma_g$, and uncertainty in

relative change of $G$ from C[4]MIP models compared to OBS (($G_{MOD} - G_{OBS}$)/$G_{OBS}$) was $\frac{G_{MOD}}{G_{OBS}} * \sqrt{\left(\frac{\sigma_{G_{MOD}}}{G_{MOD}}\right)^2 + \left(\frac{\sigma_{G_{OBS}}}{G_{OBS}}\right)^2}$.

## Data availability

Resulting data that support the findings in this study are available in the Supplementary Tables, and more related datasets can be downloaded at https://github.com/xuanzezhang/climate–carbon-cycle-feedback. Global Carbon Project report 2018 dataset (including anthropogenic CO$_2$ emission): https://www.icos-cp.eu/GCP/2018. Observed atmospheric CO$_2$ is available at http://wwww.esrl.noaa.gov/gmd/ccgg/trends/global.html. Ice-core CO$_2$ and temperature reconstruction datasets are retrieved from their published papers listed in References. The HadCRUT4 is available at https://www.metoffice.gov.uk/hadobs/hadcrut4/. The GISTEMP is retrieved from https://data.giss.nasa.gov/gistemp/. NOAA GlobalTemp is retrieved from https://climatedataguide.ucar.edu/climate-data/global-surface-temperature-data-mlost-noaa-merged-land-ocean-surface-temperature/. Berkeley Earth temperature is available at http://berkeleyearth.org/data/. The coupled/uncoupled simulations of C[4]MIP and CMIP5 models are available at https://www.c4mip.net.

## Code availability

The NCL processing codes are available via GitHub at https://github.com/xuanzezhang/climate–carbon-cycle-feedback (https://doi.org/10.5281/zenodo.4575812) or from the corresponding author upon request.

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

## Acknowledgements

This study was funded by the National Key Research and Development Program of China (2017YFA0604603), the CAS Pioneer Talent Program, and the National Key Research and Development Program of China (2016YFA0602501). X.Z.Z. was also sponsored by the National Natural Science Foundation of China (42001019) and the Shanghai Sailing Program (19YF1413100). Y.P.W. was supported by the National Environmental Science Program (climate change and earth system science). P.C. acknowledges support from the European Research Council Synergy grant ERC-2013-SyG-610028 IMBALANCE-P. We gratefully thank Dr. Pierre Friedlingstein for providing constructive comments. We acknowledge the following data providers and model developers: the Global Carbon Project, the University of East Anglia Climatic Research Unit (CRU), the NASA Goddard Institute for Space Studies, the NOAA, and Berkeley Earth. We also gratefully thank Drs Chris Jones and Vivek Arora for providing the outputs of 11 C⁴MIP and nine CMIP5 models.

## Author contributions

X.Z.Z. and Y.P.W. designed the study. X.Z.Z. performed the analysis. X.Z.Z., Y.P.W., Y.Q.Z., and P.C. drafted the paper. P.J.R. proposed the conception of box model. K.H., Y.Q.L., S.L.P., Z.L.W., J.Y.X., W.Z., X.G.Z., and J.T. contributed to the interpretation of the results and to the text.

## Competing interests

The authors declare no competing interests.
