## [Peer Review File · Nature Communications]

Reviewers' comments:

Reviewer #1 (Remarks to the Author):

This study uses the well-known linear carbon cycle feedback framework with the novelty that it applies a Fourier analysis to reveal the variability of the carbon cycle feedbacks across timescales. This analysis is applied to reconstructions based on observations and Earth system model outputs. The study finds that while the carbon-concentration feedback is relatively constant, the carbon climate feedback varies becoming more negative on centennial timescales. While it is well established that the carbon-concentration feedback is nearly constant and the carbon-climate feedback becomes increasingly negative in Earth system models (ref example 11 Arora et al., 2013, as numbered by the original manuscript), I appreciate that the authors have provided observational based estimates for these quantities on decadal and centennial timescales. The most interesting finding of the study with implications to the wider climate research community is the estimate for the climate amplifying effect for the carbon cycle based on observations, and that this estimate is significantly smaller than the estimate based on Earth system models. The implication of this overestimation by the Earth system models is that the threshold for allowable emissions before reaching 1.5 degrees warming is 14% higher than previously estimated.

Overall, in my opinion the study is interesting and has the potential to influence and revise the allowable carbon emissions before reaching warming targets. The methods, for the most part, are reasonable and appropriately demonstrated. However, in my opinion the methods presentation can be improved to make it easier to follow and more intuitive, and I have some further queries about the validity/explanation of some of the assumptions as I explain below. My biggest concern is overstatements about the potential overestimation of the amplification effect of carbon cycle based on Earth system models given the very large uncertainties both in the observational and model based estimates for γ , but most importantly all the assumptions that the observational estimates rely on. I am also somewhat reserved on how meaningful it is to directly compare carbon-climate feedback estimates based on different approaches and from different eras, and to what degree estimates from the pre-industrial apply to the future state. These limitations should be highlighted better when discussing the comparisons between observations and model estimates and in reference to the manuscript statement about revising the allowable emissions before reaching specific warming targets. Hence, I recommend revisions before the manuscript can be accepted for publication.

Specific Comments:

1. Use of notation ζ and η : Parameters ζ and η are simply the inverse of the well-defined and commonly used parameters TCRE and α , respectively, so why not use them as such. For me, and I believe for other readers too, it will be more straightforward to follow the methods if: i) instead of ζ TCRE⁻¹ is used and ii) instead of η α^{-1} is used. This notation will improve the accessibility to both the TCRE community and the community that is familiar with the carbon cycle feedback framework that uses the notation α , β , γ . I appreciate why ζ and η are reported in K/ppm in the manuscript (both figures and equations) as to make the link with pCO₂ explicit. However, I suggest presenting both or at least the TCRE⁻¹ in GtC/K, following the standard practice for the TCRE such that it corresponds to sensitivity to cumulative "carbon" emissions.

2. Line 101 equation 1 (and other equations). In my opinion the 0.472 conversion factor should not be in the equation as a simple number without units. I suggest to define the conversion from ppm to GtC as constant c (or whatever notation is chosen for this constant), and rather use its inverse accordingly in the equation such that the TCRE⁻¹ is expressed and reported in the figures as GtC/K which is the standard practice for the TCRE. The same carries over and applies for other equations and methods, for example in the carbon inventory changes (equation 3) cumulative emissions for carbon should be reported in GtC and the 0.472 should be substituted with a constant conversion factor notation that is rather assigned in ΔCA .

3. Line 181, Equation (2). I am not sure I agree with this equation and its derivation, at least not without further clarifications. This equation is derived in the methods (lines 494-500) by assuming that at pre-industrial atmospheric CO₂ remains relatively constant such that ΔCA is effectively zero, and there are no anthropogenic emissions and no related warming such that ΔTA is effectively zero. With the above assumptions then, the $TCRE^1$ is assumed effectively zero (correctly) but then an expression is derived for $\Delta CA/\Delta TA$ ($\alpha-1$). $\Delta CA/\Delta TA$ at the pre-industrial limit should at least be undetermined following the explicit assumption of $\Delta TA = 0$. I am confused as to what the meaning of this quantity is as I thought that Δ is change relatively to the pre-industrial. Maybe this is solved in terms of limits, but I think that the derivation of this equation, its meaning and what Δ represents should be more explicit.

4. Lines 242-254 and line 582: It should be noted here that the non-linearity of the carbon cycle feedback is important. Hence the estimates of β and γ using the fully coupled (COU) and biogeochemically coupled (BGC) runs are different than the estimates using the biogeochemically coupled (BGC) and the radiative coupled (RAD) runs, particularly for γ (Schwinger et al., 2014, Arora et al. 2019). My understanding is that β and γ for CMIP5 are estimates from the BGC and RAD runs, while β and γ for the C4MIP are estimated from the COU and BGC runs. I think that this should be at least mentioned and flagged when comparing with the observational based estimates. The authors should discuss using which runs (RAD+BGC or COU+BGC) do the study's method for the observational based estimates correspond to best, and the caveat of comparing these metrics estimated without considering the effect of non-linearity. At least for the ocean, I know that the estimates of γ when using the radiatively coupled run are always smaller in magnitude (less negative) than when using the coupled and the biogeochemically coupled runs (Schwinger et al., 2014, table 2, and Arora et al., 2019).

5. Line 252-255: I am not sure that the comparison between the models and the observational based estimates for γ is fair here. The CMIP5 model runs are for 140 years only, while the observations are for 850 years. γ experiences significant change with timescales as shown in figure 3, and continuously decreases in the future scenario in the Earth system models (Arora et al., 2013 and the supplementary figure 4b). If the model runs continued for 850 years, they will likely lead to a γ estimate for the centennial timescale that is more negative. Hence in my opinion the observations-based and the model-based estimates refer to different timescales which exaggerates their differences. This is later mentioned as an aside in lines 336-339 but I think this limitation should be explicit and clearly expressed along with statements such as 45% smaller in line 254.

6. Lines 283-285: I am unsure if the link with the upcoming sixth IPCC report where different model versions than in CMIP5 and C4MIP are used is appropriate here. I think for such a statement the β and γ based on CMIP6 models should at least be reported and given in a table in the supplementary like Table 3. Given the large uncertainty and all the assumptions I think that this statement is a stretch even if it was referring to the AR5 and the CMIP5 models.

7. Lines 290-296: The allowable emissions both based on model calibrations and based on the study's observational estimates should be presented with an uncertainty not just an average number and a percentage based on mean values to avoid any misconceptions.

8. Lines 315-327 and associated methods material lines 585-604. In my opinion the box model does not offer any new insight and should be removed from the manuscript. Instead, the authors should use the space to highlight the limitations for the comparison between γ from the observations during the pre-industrial and from earth system models forced by a future scenario.

9. Line 514-516 going from equation 11 to 12: I am not sure I agree with this, moving from a discrete difference form to a differential form, unless there is an explicit assumption here that β and γ are constant/time independent.

10. Line 562: I do not understand how the BA approach of Arora et al., 2013 is used here. The BA approach is used to relate fluxes with inventories/warming such that it leads to feedback metrics B and Γ of the form of $\text{GtC year}^{-1} \text{ ppm}^{-1}$ and $\text{GtC year}^{-1} \text{ K}^{-1}$. These metrics can then be link to the β and γ of the FEA approach (as defined in Arora et al. 2013). Maybe I am misunderstanding something, but can you please clarify how equations 26 and 27 are related to the BA approach and how they are derived?

11. Figure 4: I am a little surprised by panel b in figure 4. As I understand, these estimates are based on CMIP5 like the ones in Arora et al., 2013. However, figure 4 indicates that γ only slightly decreases or even remains stable on timescales larger than about 10-40 years. Based on my interpretation of Figure 6 panels d, e, f in Arora et al, 2013 (in combination with the temperature change with time in Figure 2 panel b in Arora et al, 2013) I was expecting a more substantial decrease (increase in magnitude) for γ on time scales longer than 40 years. I also do not understand why Uvic appears to be almost constant on different timescales as to me it appears to have a long-term increase signal in its magnitude in figure 6 in Arora et al, 2013. I may misunderstand or may be missing something in terms of how you generate these signals, so can you please explain.

12. γ estimates from the pre-industrial applying to the future state: In my understanding the carbon-climate feedback reflects the effect of physical and biogeochemical mechanisms. The authors have some discussion about this in lines 211-219 when it is mentioned that γ is sensitive to the state of the mean climate.

For the ocean, in the future state simulations γ encapsulates both the direct effect of warming on solubility but also the indirect effect of warming on stratification/reduction of the overturning and on biological productivity. Part of the large spread of γ in the Earth system models is associated with these models having a different representation of these processes, like different solution /parameterization for ocean's biology and different reduction in the overturning circulation. The carbon-climate feedback is probably dominated by different processes, particularly for the ocean, during 1000-1850 when for example the overturning did not experienced any long term decline due to warming as in the future. Schwinger et al., 2014 have shown that when estimating γ from the radiatively coupled simulation, the reduction in the overturning leads to an increase in carbon in the ocean below 500 meters which consequently decreases the magnitude of negative γ . Hence, I am wondering isn't it expected that the estimates of γ from the radiatively coupled simulation with future climate change will always be smaller on long timescales than the estimates based on the pre-industrial when there is no reduction in the circulation? Please can you clarify. I think some discussion should be included about directly comparing estimates of γ from different eras when different mechanisms dominated this γ .

Schwinger, J., J.F. Tjiputra, C. Heinze, L. Bopp, et al., 2014: Nonlinearity of Ocean Carbon Cycle Feedbacks in CMIP5 Earth System Models. *J. Climate*, 27, 3869–3888, <https://doi.org/10.1175/JCLI-D-13-00452.1>

Arora, V. K., et al.,: Carbon-concentration and carbon-climate feedbacks in CMIP6 models, and their comparison to CMIP5 models, *Biogeosciences Discuss.*, <https://doi.org/10.5194/bg-2019-473>, in review, 2019.

Reviewer #2 (Remarks to the Author):

Zhang and colleagues use observations over the past millennium to estimate several carbon cycle feedback parameters that are uncertain in IPCC-class models. They use reconstructions of temperature and atmospheric CO₂ to retrieve parameters that regulate the sensitivity of carbon losses from land and the oceans to a temperature increase (γ) and the sensitivity of carbon storage on

land and in the oceans to a 1 ppm change in CO₂. The paper makes a nice conceptual contribution by linking the concept of the transient climate response of cumulative emissions (TCRE) with gamma, beta, and another carbon cycle parameter, alpha, the sensitivity of temperature to a change in atmospheric CO₂. I think this is new and interesting. They use this equation and observations to estimate beta and gamma for the historical era, and I think this is really interesting and important.

In the abstract, the authors claim that the beta sensitivity term is constant across time scales, around 3 PgC/ppm and that the gamma parameter becomes more negative as a function of timescale from about -30 PgC/K over a period of a decade to less than -100 Pg C/K over a century. The authors then claim that the IPCC class models have over estimated gamma and the gain of the carbon climate feedback. As a consequence, the authors argue that collectively, we have more allowable fossil fuel emissions (14%) before we cross the warming thresholds for the Paris agreement.

It is with the assertion of a constant beta across timescales and their assertion that the gamma derived from a period with a low emissions state that can be compared to model estimates of gamma from high emissions trajectories that for me raises important conceptual concerns.

Specifically, while beta is remarkably constant as a function of time scale from the historical observations shown in Fig 2c, this does not mean it would be the same for another trajectory of CO₂ forcing, such as the one from the 1% CMIP5/CMIP6 or the A2 scenario from C4MIP. This is because the rate of CO₂ uptake from the atmosphere by the land and ocean system in response to a 1 ppm increase depends on the timescale over which the ocean and land pools are allowed to equilibrate. So the comparison in Figure 3a between the observations and the different model estimates of beta I believe to be flawed.

It does not surprise me that the CMIP5 models have a smaller beta because the rate of CO₂ increase is much faster for the 1% per year idealized CO₂ trajectory than what has occurred during the past 100 years, allowing less time for carbon to move through deeper ocean layers and more slowly turning over coarse woody debris and soil pools in the models. Gloor et al. (2010) explored this phenomena as it relates to interpretation of the airborne fraction and the conclusions from this analysis are critically relevant here. It would be better to capture the behavior of an individual model from CMIP5 using reduced complexity box model, and then force this model with the observed trajectory of atmospheric CO₂ (and recompute beta (and gamma)) to make a fairer comparison between the observations and the models.

For the same reason, I don't believe its fair to use the beta from the observational record to then separate out gamma from the observations during the 1000-year pre-industrial era. Beta is probably much larger that that derived from the industrial era, and perhaps the estimates of gamma on longer time scales are considerably underestimated.

For gamma, another important issue is raised by Schwinger et al. 2014. These authors show that the barriers to CO₂ inflow from ocean mixed later shoaling to transient CO₂ in a fully coupled simulation generates a fundamentally different gamma than the gamma obtained from response of the ocean to warming in the absence of changing CO₂ (ie diagnosing gamma from the radiative run rather than the difference between the fully coupled and biogeochemically coupled simulations). It is the latter that may be most analogous to the evolution of the climate-carbon cycle system over the last millennium.

The conclusion by the authors that the gain of the climate carbon feedback is too large (mean of 0.13 from CMIP5 depends on comparisons between models and observations that have fundamentally different trajectories of CO₂ and temperature forcing. For this reason, I do not the implications regarding the Paris Accord are at all supported by the authors' analysis.

It could be right, but to prove it, a model that mimics the behavior of the CMIP5 carbon cycle models would need to then be used to simulate the observed 20th century historical period with the identical

CO2 forcing and warming. And the same model should be used for the future projection of allowable emissions to match the Paris Accord. It might be tricky to account for other forcing agents (aerosols, CH4, etc.) in the delta temperature here, but this seems like it has to be the path forward. This could be done independently of the analysis of the last millennium and I might suggest the authors consider this for narrowing and strengthening the analysis, which has interesting and novel elements, but currently makes many comparisons for parameters that to me, appear fundamentally tied to the specific model scenarios and observational periods from which they are generated.

Specific comments:

Abstract. Beta is a parameter that contributes to the carbon concentration feedback, but is not the carbon concentration feedback itself. Same for gamma, please consider rewriting this sentence. The carbon climate feedback, for example, depends on both gamma and beta as the authors later show and understand. So please consider revising nomenclature here and in introduction (line 65).

"From the perspective of the atmosphere, beta is positive and gamma is negative." Line 70. Isn't this the opposite, where beta is positive defined from the perspective of accumulation in the land and ocean? Same for gamma (a negative gamma means loss from the land and ocean reservoirs, but a positive gain in the atmospheric carbon pool).

Equation 1. 'm' has been used to represent the conversion ratio of Pg C/ppm in past carbon cycle work (I think in Arora et al.). Or maybe its the inverse. Anyway, using m might be better than inserting a 0.472 constant in many places.

Lines 132-136. The reason n is so much larger for contemporary and future periods also has to do with radiative forcing from CO2 saturating in the wings of the 8-12 um outgoing longwave band at higher absolute CO2 levels.

I don't understand in the text and in Figures 2 and 3 how the Nyquist frequency in Fourier analysis factors in. The x axis in Figure 2 spans over 100 years (a, b) and Figure 3 is 1000 years, even though the record for the historical is about 140 years, and in Fig 1 the millennium period considered is about 850 years. Please revise or describe how it is possible to resolve (and show error bars for) estimates that have a period the same as length of the observed record.

Line 134. "Over ... " That is a really long sentence and I got lost in the middle of it. Maybe some part of a sentence was cut out or lost here?

References:

Gloor, M., Sarmiento, J. L., and Gruber, N.: What can be learned about carbon cycle climate feedbacks from the CO2 airborne fraction?, *Atmos. Chem. Phys.*, 10, 7739–7751, <https://doi.org/10.5194/acp-10-7739-2010>, 2010.

Schwinger, J., et al. (2014), Nonlinearity of ocean carbon cycle feedbacks in CMIP5 Earth system models, *J. Clim.*, 27(11), 3869–3888, doi:10.1175/jcli-d-13-00452.1.

Reviewer #3 (Remarks to the Author):

The study is addressing an important topic of the climate-carbon cycle feedback. The manuscript is well written, but the study has ignored the inherent nonlinearity of the carbon-cycle framework that is set out in a substantial study by Schwinger et al. (2014), Nonlinearity of Ocean Carbon Cycle Feedbacks in CMIP5 Earth system models, Journal of Climate.

Unfortunately I have a problem with the central part of the manuscript with its focus on time-dependence of the climate-carbon cycle feedback. The study ignores how the carbon-cycle feedback parameters are defined and the inherent nonlinearity in their framework, which is clearly set out in Schwinger et al. (2014). In this study, the carbon-cycle feedback parameters, beta and gamma, are based on a Taylor expansion relative to the pre-industrial state; see equation (2) in Schwinger et al. (2014). This approach was taken by the original study of Friedlingstein et al. (2003), but is more completely set out by Schwinger et al. (2014). In more detail, the change in the carbon inventory depends on a linear sum of first order differential terms involving T and CO₂ plus further second order and higher order differential terms, where all the differential terms are evaluated relative to the preindustrial. The beta and gamma terms are defined by the first order differential terms evaluated at the time of the preindustrial with the second order and higher differential terms neglected, such that $\beta = dF/dCO_2$ at the pre industrial and $\gamma = dF/dT$ at the preindustrial, where F is a function defining the climate system. Schwinger et al. (2014) explicitly state that the shortcoming of this approach is that there is no accounting of time dependence of inventory changes. In addition, Schwinger et al. (2014) demonstrate that the ocean carbon-cycle feedbacks are inherently nonlinear.

Given the Schwinger et al. (2014) study, I am not convinced that the present manuscript is robust. The manuscript estimates the time-dependence of the carbon cycle parameters by a Fourier series fit over different time periods. However, the beta and gamma parameters are then not still evaluated at the preindustrial as they should be, but instead are evaluated at the instantaneous time. If the terms are evaluated at the instantaneous time, then the original Taylor expansion does not hold that was used to define beta and gamma.

Evaluating the beta and gamma terms at different times probably effectively means that the neglected high-order differential terms are being melded into their estimates, so that there is an issue of errors arising from the nonlinearity of the framework.

I am aware that there are prior studies that have evaluated the time-dependence of the carbon-cycle feedback parameters, but the Frank et al. (2010) study and the Willeit et al. (2014) were either before or unaware of the Schwinger et al. (2014) study.

The authors can of course choose to evaluate these differentials dF/dCO_2 and dF/dT at any time, but they should not then equate them to beta and gamma, or expect the actual linearisation of the carbon budget to hold, so that the wider implications of their study is then lost.

My other concerns are more minor. The theory introduced in (1) and the Methods in equation (3) would be clearer if all carbon inventories were quoted in GtC or PgC, rather than have the atmospheric inventory and carbon emission in ppm. Making the units the same for all the carbon variables (that have the same symbol) would avoid the 0.472 conversion factors being included.

The transient climate response to emissions, TCRE, is a widely used climate metric. The manuscript would be better advised to focus on that variable, rather than its reciprocal.

In the methods, equations (4) and (5) should be estimated at the same reference time, usually taken to be the pre industrial.

In the methods, the variables that are time dependent should be explicitly defined in equations (3) to (7). Based on Schwinger et al. (2014), beta and gamma terms should not be time dependent.

In summary, the manuscript is focussing on evaluating the time dependence of the carbon-cycle parameters without taking into account the time state that these differentials are evaluated at and ignoring the nonlinearity from the neglected higher-order terms. The study needs to reconcile their approach with the Schwinger et al. (2014) study that highlights the inherent nonlinearity of the carbon-cycle framework and the requirement to evaluate beta and gamma at the same reference time. While the manuscript makes many inferences for beta and gamma for different time periods, it is difficult to judge their value unless the estimates are referenced to the same time point and the error from the neglected nonlinear terms are accounted for.

To Reviewer #1:

[Comment A1] This study uses the well-known linear carbon cycle feedback framework with the novelty that it applies a Fourier analysis to reveal the variability of the carbon cycle feedbacks across timescales. This analysis is applied to reconstructions based on observations and Earth system model outputs. The study finds that while the carbon-concentration feedback is relatively constant, the carbon climate feedback varies becoming more negative on centennial timescales. While it is well established that the carbon concentration feedback is nearly constant and the carbon-climate feedback becomes increasingly negative in Earth system models (ref example 11 Arora et al., 2013, as numbered by the original manuscript), I appreciate that the authors have provided observational based estimates for these quantities on decadal and centennial timescales. The most interesting finding of the study with implications to the wider climate research community is the estimate for the climate amplifying effect for the carbon cycle based on observations, and that this estimate is significantly smaller than the estimate based on Earth system models. The implication of this overestimation by the Earth system models is that the threshold for allowable emissions before reaching 1.5 degrees warming is 14% higher than previously estimated.

[Response A1]: We thank the reviewer very much for highlighting the importance of our work of the observation-based estimates for the climate-carbon cycle feedback parameters and of the estimate for the climate amplifying effect. According to your comments, we have changed the title of our manuscript to “A small climate-amplifying effect of climate-carbon cycle feedback”, and strengthened the comparison of the feedback gain between historical period and future high emission scenarios.

[Comment A2] Overall, in my opinion the study is interesting and has the potential to influence and revise the allowable carbon emissions before reaching warming targets. The methods, for the most part, are reasonable and appropriately demonstrated. However, in my opinion the methods presentation can be improved to make it easier to follow and more intuitive, and I have some further queries about the validity/explanation of some of the assumptions as I explain below. My biggest concern is overstatements about the potential

overestimation of the amplification effect of carbon cycle based on Earth system models given the very large uncertainties both in the observational and model based estimates for γ , but most importantly all the assumptions that the observational estimates rely on. I am also somewhat reserved on how meaningful it is to directly compare carbon-climate feedback estimates based on different approaches and from different eras, and to what degree estimates from the pre-industrial apply to the future state. These limitations should be highlighted better when discussing the comparisons between observations and model estimates and in reference to the manuscript statement about revising the allowable emissions before reaching specific warming targets. Hence, I recommend revisions before the manuscript can be accepted for publication.

[Response A2]: As suggested by the reviewer, we have carefully revised the paper to make our paper clearer and to avoid the overstating the results of this study.

(1) We agree that the estimates of the amplification effect of carbon cycle (feedback gain factor, g in our study) remains highly uncertain for both the observational and modeled-based results. Both carbon-concentration feedback (β) and carbon-climate feedback (γ) contribute to the amplification effect, therefore, the large uncertainty in carbon-climate feedback would leads to large uncertainty in the amplification effect. In this study, our calculation of the feedback gain is based on the equation (Gregory et al., 2009): $g = 1 - 1/[AF(1 + \beta)]$, and the cumulative airborne fraction (AF): $AF = 1/(1 + \beta + \alpha\gamma^*)$. These two equations show that the gain factor is driven by β , γ^* , and α , the sensitivity of climate to atmospheric CO_2 . Our estimates of these parameters for the industrial period of 1880-2017 showed that the β dominated most of (>95%) variation of AF and g over the 1880-2017, rather than the γ . The observation-based estimates of α (0.005 K GtC^{-1}) and γ^* (-10.9 GtC K^{-1}) lead to a small $\alpha\gamma^*$ ($\sim 0.05 \text{ GtC GtC}^{-1}$), while the β is $\sim 1.52 \text{ GtC GtC}^{-1}$, meaning that $\beta \gg \alpha\gamma^*$, indicating the for the historical period 1880-2017, the uncertainty in the amplification effect was mainly dominated by the uncertainty in carbon-concentration feedback (β). The uncertainty in β results from the uncertainties in CO_2 emissions from fossil fuel combustion and land-use change estimates and errors in the temperature datasets and the regression in equation (2) in the revised manuscript. Therefore, we showed the observation-based feedback gain (g) was very small (0.01 ± 0.05) for the 1880-2017 based on the observational β . In the

revised version, we also estimated feedback gain (g) from C4MIP models for the same period 1880-2017 to be 0.09 ± 0.04 (see Supplementary Figs. 7-8 in the revised version), while the uncertainty in the C4MIP-based g was mainly contributed by C4MIP-based β from the large spread between 11 C4MIP models.

(2) We also agree that comparison of different estimates of γ from different studies or over different periods can be problematic. In the revised manuscript, we avoided the direct comparison between them, and pointed out the limited comparability between them (Lines 330-333 in the revised version). When a comparison is made, we used the same period, such as the observation-based γ and model-based γ for the same period 1880-2017, or used the same approach, such as the model-based γ for periods between historical period and future high emission scenarios (please check for revisions in Lines 319-330 in the revised version).

Specific Comments:

[Comment A3] 1. Use of notation ζ and η : Parameters ζ and η are simply the inverse of the well-defined and commonly used parameters $TCRE$ and α , respectively, so why not use them as such. For me, and I believe for other readers too, it will be more straightforward to follow the methods if: i) instead of ζ $TCRE^{-1}$ is used and ii) instead of η α^{-1} is used. This notation will improve the accessibility to both the $TCRE$ community and the community that is familiar with the carbon cycle feedback framework that uses the notation α , β , γ . I appreciate why ζ and η are reported in K/ppm in the manuscript (both figures and equations) as to make the link with pCO_2 explicit. However, I suggest presenting both or at least the $TCRE^{-1}$ in GtC/K , following the standard practice for the $TCRE$ such that it corresponds to sensitivity to cumulative “carbon” emissions.

[Response A3]: Thanks for this constructive comment. We agree that $TCRE$ and α are well-defined and more commonly used in the climate change and carbon cycle research community. We have used the notations α , β , γ and $TCRE$ to build the new climate-carbon cycle feedback framework (Equation (2)) in the revised version. We replaced all ζ with $TCRE^{-1}$ in GtC/K and η with α^{-1} in GtC/K in text and figures for the analysis of the industrial period. We keep the use of η in the analysis for the pre-industrial period. As η has been frequently used in previous studies (Frank et al., 2010; Willeit et al., 2014). We compared the

estimated η from our method to those from previous studies (Frank et al., 2010) for the pre-industrial period in the revised version (Lines 228-233).

[Comment A4] 2. Line 101 equation 1 (and other equations). In my opinion the 0.472 conversion factor should not be in the equation as a simple number without units. I suggest to define the conversion from ppm to GtC as constant c (or whatever notation is chosen for this constant), and rather use its inverse accordingly in the equation such that the $TCRE^{-1}$ is expressed and reported in the figures as GtC/K which is the standard practice for the TCRE. The same carries over and applies for other equations and methods, for example in the carbon inventory changes (equation 3) cumulative emissions for carbon should be reported in GtC and the 0.472 should be substituted with a constant conversion factor notation that is rather assigned in ΔCA .

[Response A4]: Agreed. We have defined the conversion factor as $m=2.12$ GtC ppm⁻¹ (m was used in Arora et al. 2013) in the revised version. As ΔC_E , and ΔC_A have been reported in GtC, and $TCRE^{-1}$ in GtC/K in the revised version, m will not appear in equations (1-2) in the manuscript, but in equation (3) for the analysis of pre-industrial period, m is used to convert η in ppm K⁻¹ to GtC K⁻¹.

[Comment A5] 3. Line 181, Equation (2). I am not sure I agree with this equation and its derivation, at least not without further clarifications. This equation is derived in the methods (lines 494-500) by assuming that at pre-industrial atmospheric CO₂ remains relatively constant such that ΔCA is effectively zero, and there are no anthropogenic emissions and no related warming such that ΔTA is effectively zero. With the above assumptions then, the $TCRE^{-1}$ is assumed effectively zero (correctly) but then an expression is derived for $\Delta CA/\Delta TA$ (α^{-1}). $\Delta CA/\Delta TA$ at the pre-industrial limit should at least be undetermined following the explicit assumption of $\Delta TA = 0$. I am confused as to what the meaning of this quantity is as I thought that Δ is change relatively to the pre-industrial. Maybe this is solved in terms of limits, but I think that the derivation of this equation, its meaning and what Δ represents should be more explicit.

[Response A5]: Thanks very much for this detailed comment. We agree that the assumptions

for equation (3) (or equation (2) in the previous version) need further clarifications. Firstly, we agree that both ΔC_A and ΔT_A were not well defined previously. In the revised manuscript, we stated that ΔC_A and ΔT_A were the changes in atmospheric CO₂ concentration and surface air temperature over a time period of Δt . Thus, for the analysis of the industrial period 1850-2017, the Δ represents changes relative to the year 1850. But for the analysis of the pre-industrial period 1000-1850, then the Δ represents changes relative to the year 1000. Therefore, the ΔC_A and ΔT_A for the pre-industrial period 1000-1850 were calculated from $C_A^{1850} - C_A^{1000}$ and $T_A^{1850} - T_A^{1000}$ respectively. From the observations as showed in Fig.1, we know that both ΔC_A and ΔT_A were not zero over 1000-1850, thus the $\Delta C_A/\Delta T_A$ (α^{-1} or η) and their decompositions on different timescales can be derived from observations (Supplementary Figs. 3-6). Secondly, the assumptions for equation (3) in the revised version are: i) Over the pre-industrial period 1000-1850, the anthropogenic CO₂ emissions was relatively small ($\Delta C_E \approx 0$), only contributed from the early land use such as during the Litter Ice Age period; ii) The small anthropogenic CO₂ emissions over 1000-1850 have not induced global warming. From Fig.1b, we can find that the temperature datasets over 1000-1850 actually shows decreasing trends, thus we assumed that change in temperature (ΔT_A) during 1000-1850 was almost dominated by the climate variability and its interaction with natural CO₂ variations (ΔC_A). Then equation (3) can be derived from equation (6) when $\Delta C_E \approx 0$ in the Methods in the revised version.

[Comment A6] 4. Lines 242-254 and line 582: It should be noted here that the non-linearity of the carbon cycle feedback is important. Hence the estimates of β and γ using the fully coupled (COU) and biogeochemically coupled (BGC) runs are different than the estimates using the biogeochemically coupled (BGC) and the radiative coupled (RAD) runs, particularly for γ (Schwinger et al., 2014, Arora et al. 2019). My understanding is that β and γ for CMIP5 are estimates from the BGC and RAD runs, while β and γ for the C4MIP are estimated from the COU and BGC runs. I think that this should be at least mentioned and flagged when comparing with the observational based estimates. The authors should discuss using which runs (RAD+BGC or COU+BGC) do the study's method for the observational based estimates correspond to best, and the caveat of comparing these metrics estimated

without considering the effect of non-linearity. At least for the ocean, I know that the estimates of γ when using the radiatively coupled run are always smaller in magnitude (less negative) than when using the coupled and the biogeochemically coupled runs (Schwinger et al., 2014, table 2, and Arora et al., 2019).

[Response A6]: Thanks for this very constructive comment. the nonlinearity of the carbon cycle feedback was ignored in previous version. Following reviewer’s suggestions and the definitions based on refs. (Arora et al., 2019; Schwinger et al., 2014). In the revised version, we included the consideration of the nonlinearity feedback term in our climate-carbon cycle feedback analysis framework, see equations (1-2) and equations (5-8). As defined by Schwinger et al., 2014, when assuming the carbon stock in biosphere ($C_B = C_L + C_O$) at the reference climate state as a function of climate and CO₂: $C_B = F(C_A, T_A)$, then β , γ , and $f(\beta, \gamma)$ can be expressed as the 1st order and 2nd order coefficients of the Taylor series of C_B since the initial time ($t = 0$): $\beta = \frac{\partial F}{\partial C_A} \Big|_0$, $\gamma = \frac{\partial F}{\partial T_A} \Big|_0$, and $f(\beta, \gamma) = \frac{\partial^2 F}{\partial C_A \partial T_A} \Big|_0 + \frac{1}{2} \frac{\partial^2 F}{\partial C_A^2} \Big|_0 \frac{\Delta C_A}{\Delta T_A} + \frac{1}{2} \frac{\partial^2 F}{\partial T_A^2} \Big|_0 \frac{\Delta T_A}{\Delta C_A} + R_3$. The nonlinear feedback $f(\beta, \gamma)$ in this study represents the 2nd and high-order terms of the Taylor expansion. As previous studies mainly focused on the nonlinearity of the carbon-climate (γ -) feedback (Gregory et al., 2009; Schwinger et al., 2014; Zickfeld et al., 2011), in this study, we combined the γ -feedback and the atmospheric CO₂ change’s impacts on the nonlinear feedback as: $\gamma^* = \gamma + f(\beta, \gamma)\Delta C_A$ (see revisions on Lines 540-576 in Methods in the revised version). Because we cannot separate the nonlinear feedback contribution from observation-based estimates from our current feedback analysis, we quantified the nonlinear feedback term $f(\beta, \gamma)$ and its contribution to the γ -feedback $f(\beta, \gamma)\Delta C_A$ from the CMIP5 models’ three groups of simulations: the COU, BGC and RAD runs, and the COU and BGC runs of C4MIP models. Thus, we defined and calculated the direct β -feedback from the BGC simulations ($\beta^{BGC} = \Delta C_B^{BGC} / \Delta C_A^{BGC}$) and the direct γ -feedback from the RAD simulations ($\gamma^{RAD} = \Delta C_B^{RAD} / \Delta T_A^{RAD}$) and the indirect γ -feedback from the COU-BGC simulations ($\gamma^{COU-BGC} = (\Delta C_B^{COU} - \Delta C_B^{BGC}) / \Delta T_A^{COU}$), for the observation-overlapped period of 1880-2017 and the future emission scenario of 2018-2100 for the C⁴MIP models, and the 1pctCO₂ 140-year period for the CMIP5 models, respectively. We also estimated the nonlinear feedback from

the difference between COU simulations and the BGC and RAD simulations ($f(\beta, \gamma) = [\Delta C_B^{COU} - (\Delta C_B^{BGC} + \Delta C_B^{RAD})] / \Delta C_A^{COU} \Delta T_A^{COU}$) and its contribution to γ -feedback ($f(\beta, \gamma) \Delta C_A^{COU}$) for the CMIP5 models. Results have been showed in Fig. 3a, c and the Supplementary Tables (3-4) in the revised version. We estimated that the CMIP5-based nonlinear feedback $f(\beta, \gamma)$ for the 1pctCO₂ 140-year period was $-11.22 \pm 11.72 \times 10^{-3}$ GtC ppm⁻¹ K⁻¹, and its contribution to the γ -feedback was -9.6 ± 10.03 GtC K⁻¹, which means that the γ^{RAD} was about 15% smaller in magnitude than the $\gamma^{COU-BGC}$ feedback. We also have added the results in the manuscript (Lines 280-333).

[Comment A7] 5. Line 252-255: I am not sure that the comparison between the models and the observational based estimates for γ is fair here. The CMP5 model runs are for 140 years only, while the observations are for 850 years. γ experiences significant change with timescales as shown in figure 3, and continuously decreases in the future scenario in the Earth system models (Arora et al., 2013 and the supplementary figure 4b). If the model runs continued for 850 years, they will likely lead to a γ estimate for the centennial timescale that is more negative. Hence in my opinion the observations-based and the model-based estimates refer to different timescales which exaggerates their differences. This is later mentioned as an aside in lines 336-339 but I think this limitation should be explicit and clearly expressed along with statements such as 45% smaller in line 254.

[Response A7]: Thanks for this comment. We have realized that it's unfair to compare the model-based estimates for γ with the pre-industrial observation-based γ . Following your suggestion, we have pointed out that the model-based γ and observation-based γ refer to different time periods and timescales, there are very limited comparability between them (Lines 330-333 in the manuscript). In the revised version, we have removed the direct comparison between them.

[Comment A8] 6. Lines 283-285: I am unsure if the link with the upcoming sixth IPCC report where different model versions than in CMIP5 and C4MIP are used is appropriate here. I think for such a statement the β and γ based on CMIP6 models should at least be reported and

given in a table in the supplementary like Table 3. Given the large uncertainty and all the assumptions I think that this statement is a stretch even if it was referring to the AR5 and the CMIP5 models.

[Response A8]: Thanks for pointing out this issue here. We agree that the earth system models of the upcoming sixth IPCC report (CMIP6) have been upgraded to the new versions which are different from CMIP5 models. As shown by ref. (Arora et al., 2019), CMIP6 models with nitrogen cycle coupled with carbon cycle showed smaller β and γ in magnitude, and other metric also could have changed. In our study, as we have not conducted the comparisons from CMIP6 models (due to that the CMIP6 C4MIP datasets are not current available for download), we have removed the related statements in the revised version.

[Comment A9] 7. Lines 290-296: The allowable emissions both based on model calibrations and based on the study's observational estimates should be presented with an uncertainty not just an average number and a percentage based on mean values to avoid any misconceptions.

[Response A9]: Agreed and revised according to this suggestion. In the revised version, we calculated the underestimation of allowable emissions based the overestimation of feedback amplification effect ($G = \frac{1}{1-g} \approx 1 + g$) and feedback gain (g) based on C⁴MIP models than those of observation-based values. The uncertainty in G for C⁴MIP models is $\delta_{G_{MOD}} = \delta_{g_{MOD}}$, and for observation is $\delta_{G_{OBS}} = \delta_{g_{OBS}}$. Thus, the relative change in G from C⁴MIP models compare to observation is $(G_{MOD} - G_{OBS})/G_{OBS} * 100\%$, and its uncertainty is $\frac{G_{MOD}}{G_{OBS}} * \sqrt{\left(\frac{\sigma_{G_{MOD}}}{G_{MOD}}\right)^2 + \left(\frac{\sigma_{G_{OBS}}}{G_{OBS}}\right)^2} * 100\%$. Using the observation-based g (0.01 ± 0.05) and the new C⁴MIP-based g (0.09 ± 0.04) for the same period 1880-2017, we estimated allowable emissions would be $9 \pm 7\%$ more, or 125 ± 8 GtC. Please check these revisions in the revised manuscript (Lines 362-368, 383-388, 721-724).

[Comment A10] 8. Lines 315-327 and associated methods material lines 585-604. In my opinion the box model does not offer any new insight and should be removed from the manuscript. Instead, the authors should use the space to highlight the limitations for the comparison between γ from the observations during the pre-industrial and from earth system

models forced by a future scenario.

[Response A10]: Thanks for this comment. We have removed the box model part from the Discussions. The box model was used to diagnose whether our approach for estimating β and γ from observations (equation (2)) is solid. The result from box model showed that observation-based β and γ can successfully predict CO₂ and temperature which close the observed values.

[Comment A11] 9. Line 514-516 going from equation 11 to 12: I am not sure I agree with this, moving from a discrete difference form to a differential form, unless there is an explicit assumption here that β and γ are constant/time independent.

[Response A11]: Thanks for this detailed comment. We have revised equation (12) by moving from a discrete difference form to a partial differential form (i.e., $\frac{\partial C_A}{\partial t} = \frac{\Delta C_A}{\Delta t}$ for $\Delta t \rightarrow 0$, $\Delta C_A \rightarrow 0$). In the revised version, the equation (11) is $\frac{\Delta C_E}{\Delta t} = \frac{\Delta C_A}{\Delta t} + \beta \frac{\Delta C_A}{\Delta t} + \gamma \frac{\Delta T_A}{\Delta t} + f(\beta, \gamma) \Delta C_A \frac{\Delta T_A}{\Delta t}$, which includes a nonlinear feedback term. Then equation (12) is $\frac{\partial C_E}{\partial t} = (1 + \beta) \frac{\partial C_A}{\partial t} + \gamma^* \frac{\partial T_A}{\partial t}$, where $\gamma^* = \gamma + f(\beta, \gamma) \Delta C_A$. This holds when $\beta = \beta(t)$ or $\gamma = \gamma(t)$. On the other hand, we estimated that the nonlinear feedback term only made a small contribution (15%) to the γ^* -feedback and a negligible contribution (3%) to the β -feedback from the CMIP5 1pctCO₂ experiments (see Supplementary Table 4 in revised version).

[Comment A12] 10. Line 562: I do not understand how the BA approach of Arora et al., 2013 is used here. The BA approach is used to relate fluxes with inventories/warming such that it leads to feedback metrics B and Γ of the form of GtC year-1 ppm-1 and GtC year-1 K-1 . These metrics can then be link to the β and γ of the FEA approach (as defined in Arora et al. 2013). Maybe I am misunderstanding something, but can you please clarify how equations 26 and 27 are related to the BA approach and how they are derived?

[Response A12]: Thanks for pointing out this unclear definition. These metrics were actually linked to the β and γ of the FEA approach as defined in Arora et al. 2013 (Arora et al., 2013). In the revised version, we re-defined β and γ from different approaches (Arora et al., 2019):

$\beta^{BGC} \approx \Delta C_B^{BGC} / \Delta C_A^{BGC}$ from the biogeochemically-coupled (BGC) simulations of C⁴MIP and CMIP5 models, $\gamma^{COU-BGC} \approx (\Delta C_B^{COU} - \Delta C_B^{BGC}) / \Delta T_A^{COU}$ from fully-coupled (COU) and biogeochemically-coupled (BGC) simulations of C⁴MIP models, and $\gamma^{RAD} = \Delta C_B^{RAD} / \Delta T_A^{RAD}$ for the radiatively-coupled (RAD) simulations from CMIP5 models. Please check these revisions in the revised manuscript (Lines 663-679). Because the FEA approach was only used to calculate the β^{BGC} and $\gamma^{COU-BGC}$ over the whole study period as used in Arora et al. 2013 (Arora et al., 2013), which means that the Δt is 140 years, and ΔC is the carbon stock change between the first year and the last year. In our study, we also want to calculate β_k and γ_k on different timescales (Figure A1), e.g., $\Delta t=2, 5, 10, 30, 50, 100, 140$ years. To this end, we developed equations (S1-S6, in Texts.1-2, Supplementary Information). For example, to calculate the β_{10yr} and γ_{10yr} on the timescale of 10 years, i.e. $\Delta t=10$ year, the whole study period 140 years has a number of 14 of $\Delta t=10$ year if each time interval dose not overlap, from equation 26, we actually calculate the average of all 14 β^{BGC} at 10 year: $\frac{1}{14} \sum \beta^{BGC}_i = \frac{1}{14} \sum \frac{\Delta C_{B,i}^{BGC}}{\Delta C_{A,i}^{BGC}}, i = 1,14.$

Figure A1. A diagram to show the difference of calculating β^{BGC} and γ^{RAD} for different time periods (the FEA approach) and calculating β_k^{BGC} and γ_k^{RAD} on different time scales (our approach). This figure has been added in the revised version (Supplementary Fig. 10).

[**Comment A13**] 11. Figure 4: I am a little surprised by panel b in figure 4. As I understand, these estimates are based on CMIP5 like the ones in Arora et al., 2013. However, figure 4 indicates that γ only slightly decreases or even remains stable on timescales larger than about 10-40 years. Based on my interpretation of Figure 6 panels d, e, f in Arora et al, 2013 (in combination with the temperature change with time in Figure 2 panel b in Arora et al, 2013) I was expecting a more substantial decrease (increase in magnitude) for γ on time scales longer than 40 years. I also do not understand why Uvic appears to be almost constant on different timescales as to me it appears to have a long-term increase signal in its magnitude in figure 6 in Arora et al, 2013. I may misunderstand or may be missing something in terms of how you generate these signals, so can you please explain.

[**Response A13**]: Thanks for this detailed comment. The γ_k^{RAD} across timescales in Figure 4b in the manuscript differ from the results in Figure 6 panels d, e, f in Arora et al, 2013, because we calculated it differently. Figure A1 shows the difference between the FEA approach as used in Arora et al, 2013 and our approach for calculating γ_k^{RAD} on different timescales. When calculating γ^{RAD} over a time period Δt using the FEA approach, we calculated the changes in carbon stocks from the first simulation year (t_0):

$$\gamma^{RAD} = \frac{\Delta C_B^{RAD}}{\Delta T_A^{RAD}} = \frac{C_B^{RAD}(t_0 + \Delta t) - C_B^{RAD}(t_0)}{T_A^{RAD}(t_0 + \Delta t) - C_B^{RAD}(t_0)}.$$

As is shown in Figure A1, when setting $\Delta t = 5, 10, 20, 30, \dots$ years, we can calculate γ^{RAD} over the periods of simulation year 1-5, 1-10, 1-20, 1-30, etc., respectively. As shown in Figure A2a, we calculated the $\gamma^{RAD} = \Delta C_B^{RAD} / \Delta T_A^{RAD}$ for the radiatively-coupled (RAD) simulations from CMIP5 models using 1pctCO₂ 140 simulation years over different periods since the first simulation year. Figure A2a shows similar results to the results in Figure 6 panels d, e, f in Arora et al, 2013. Most models in Figure A2a show a substantial decrease (increase in magnitude) for γ^{RAD} on time scales longer than 40 years. The Uvic ESCM2.9 also shows substantial decreased γ^{RAD} on time scales from 2 to 120 years. However, in our approach for calculating γ_k^{RAD} on different timescales, we calculate the average of γ^{RAD} from many time intervals of Δt with the same length (Figure A1). The Δt does not only refer to the first simulation year (t_0), but also refer to the simulation year of $t_0 + i\Delta t, i =$

0,1,2,3 ... n. Then, the γ_k^{RAD} on the timescale of Δt is

$$\gamma_{k=\Delta t}^{RAD} = \frac{1}{n} \sum \gamma_i^{RAD} = \frac{1}{n} \sum \frac{\Delta C_{B,i}^{RAD}}{\Delta T_{A,t}^{RAD}} = \frac{1}{n} \sum \frac{C_B^{RAD}(t_0+(i+1)\Delta t) - C_B^{RAD}(t_0+i\Delta t)}{T_A^{RAD}(t_0+(i+1)\Delta t) - C_B^{RAD}(t_0+i\Delta t)}$$

Our results showed the γ_k^{RAD} on timescales longer than 10 years became larger on magnitude (more negative) as shown in Supplementary Figs. 7-8 in the revised version. This is because that the γ^{RAD} calculated from the reference time ($t_0 + i\Delta t$) became larger in magnitude when $i\Delta t$ increase (Figure A2b). This consequently made the averaged value (γ_k^{RAD}) larger in magnitude than the γ^{RAD} calculated from the first simulation year as the reference time. This result also means that in CMIP5 models, the carbon stocks in land and ocean over the later periods of the 1pctCO2 simulations experiment are more sensitive to changes in surface temperature than the early periods.

Figure A2. Comparison of γ^{RAD} from different reference times ($t_0 + i\Delta t$). **a.** the γ^{RAD} for the simulation intervals ($t_0 + \Delta t$) since the first simulation year (t_0), $\Delta t = 2, 3, \dots, 120$ years. γ^{RAD} were calculated using the FEA approach. **b.** the γ^{RAD} for the simulation periods (Δt) since the new reference years ($t_0 + 5\Delta t$), $\Delta t = 2, 3, \dots, 120$ years.

[Comment A14] 12. γ estimates from the pre-industrial applying to the future state: In my understanding the carbon-climate feedback reflects the effect of physical and biogeochemical mechanisms. The authors have some discussion about this in lines 211-219 when it is mentioned that γ is sensitive to the state of the mean climate. For the ocean, in the future state simulations γ encapsulates both the direct effect of warming on solubility but also the indirect

effect of warming on stratification/reduction of the overturning and on biological productivity. Part of the large spread of γ in the Earth system models is associated with these models having a different representation of these processes, like different solution /parameterization for ocean's biology and different reduction in the overturning circulation.

The carbon-climate feedback is probably dominated by different processes, particularly for the ocean, during 1000-1850 when for example the overturning did not experienced any long term decline due to warming as in the future. Schwinger et al., 2014 have shown that when estimating γ from the radiatively coupled simulation, the reduction in the overturning leads to an increase in carbon in the ocean below 500 meters which consequently decreases the magnitude of negative γ . Hence, I am wondering isn't it expected that the estimates of γ from the radiatively coupled simulation with future climate change will always be smaller on long timescales than the estimates based on the pre-industrial when there is no reduction in the circulation? Please can you clarify. I think some discussion should be included about directly comparing estimates of γ from different eras when different mechanisms dominated this γ .

[Response A14]: Thanks for this constructive comment. We agree that the carbon-climate feedback is sensitive to the state of the mean climate the responses of physical and biogeochemical processes, particularly in the ocean. In the revised version, we have avoided the direct comparison between the pre-industrial observation-based γ and the CMIP5 or C4MIP model-based γ . What we can learn from the results of the pre-industrial observation-based γ derived from the quasi-equilibrium climate state for 1000-1850 is that γ -feedback would increase (become more negative) from decadal to centennial scales. But for the industrial period 1880-2017 and future high emissions scenarios, the climate state is/will be not on an equilibrium climate state, as mentioned by the reviewer, change in the carbon-climate (γ -) feedback depends on both the warming transient climate and increasing CO₂. The warming has both direct effects on terrestrial ecosystem respirations (Cheng et al., 2017; Schuur et al., 2015) and sea water solubility, which makes γ more negative, and indirect effects on biological productivity and phenology (Piao et al., 2019) and on stratification/reduction of the overturning (Schwinger et al., 2014; Zickfeld et al., 2011), which makes γ less negative. On the other hand, the overturning transports the increased

CO₂ from sea surface to deep ocean. But the warming reduces the overturning, hence the carbon transport. This interaction between warming and increasing CO₂ shows a nonlinear feedback contribution to the γ -feedback. Therefore, change in the nonlinear feedback contribution depends on the reduction in carbon transport due to decreased overturning. When there is no reduction in the overturning circulation, the ocean nonlinear feedback contribution would become smaller. While the γ -feedback would continue to grow because of the warming impacts on terrestrial ecosystem respirations and sea water solubility. In the revised version, we showed that the estimated $\gamma^{COU-BGC}$ from eleven C⁴MIP models were on magnitude increased from -27.52 ± 11.93 GtC K⁻¹ for the historical period of 1880-2017 to -62.27 ± 32.17 GtC K⁻¹ for the near RCP8.5 scenario of 2018-2100. For the 1pctCO₂ high emission scenario, the CMIP5-based $\gamma^{COU-BGC}$ was -70.14 ± 32.43 GtC K⁻¹, and the direct γ -feedback γ^{RAD} was -65.08 ± 30.74 GtC K⁻¹, and the nonlinear feedback contribution was -9.6 ± 10.03 GtC K⁻¹, about $15 \pm 17\%$ of the γ -feedback. Thus, the γ -feedback at a new equilibrium state of the warmer climate and higher CO₂ world of high emissions scenarios would become much larger on magnitude than the γ -feedback for the industrial or pre-industrial periods, and likely play the dominant role to enlarge the feedback gain and the amplification effect. We have included more discussions in the revised version (Lines 406-434).

Reference:

- Arora, V.K. et al., 2013. Carbon–Concentration and Carbon–Climate Feedbacks in CMIP5 Earth System Models. *J. Clim.*, 26(15): 5289-5314.
- Arora, V.K. et al., 2019. Carbon-concentration and carbon-climate feedbacks in CMIP6 models, and their comparison to CMIP5 models. *Biogeosciences Discuss.*, in review.
- Cheng, L. et al., 2017. Warming enhances old organic carbon decomposition through altering functional microbial communities. *ISME J*, 11(8): 1825-1835.

- Frank, D.C. et al., 2010. Ensemble reconstruction constraints on the global carbon cycle sensitivity to climate. *Nature*, 463(7280): 527-30.
- Gregory, J.M., Jones, C.D., Cadule, P. and Friedlingstein, P., 2009. Quantifying Carbon Cycle Feedbacks. *J. Clim.*, 22(19): 5232-5250.
- Hajima, T., Tachiiri, K., Ito, A. and Kawamiya, M., 2014. Uncertainty of Concentration–Terrestrial Carbon Feedback in Earth System Models*. *J. Clim.*, 27(9): 3425-3445.
- Piao, S. et al., 2019. Plant phenology and global climate change: Current progresses and challenges. *Glob. Change Biol.*, 25(6): 1922-1940.
- Randerson, J.T. et al., 2015. Multicentury changes in ocean and land contributions to the climate-carbon feedback. *Glob. Biogeochem. Cycles*, 29(6): 744-759.
- Schuur, E.A.G. et al., 2015. Climate change and the permafrost carbon feedback. *Nature*, 520(7546): 171-9.
- Schwinger, J. et al., 2014. Nonlinearity of Ocean Carbon Cycle Feedbacks in CMIP5 Earth System Models. *J. Clim.*, 27(11): 3869-3888.
- Willeit, M., Ganopolski, A., Dalmonech, D., Foley, A.M. and Feulner, G., 2014. Time-scale and state dependence of the carbon-cycle feedback to climate. *Clim. Dyn.*, 42(7-8): 1699-1713.
- Zickfeld, K., Eby, M., Matthews, H.D., Schmittner, A. and Weaver, A.J., 2011. Nonlinearity of Carbon Cycle Feedbacks. *J. Clim.*, 24(16): 4255-4275.

To Reviewer #2:

[Comment B1] Zhang and colleagues use observations over the past millennium to estimate several carbon cycle feedback parameters that are uncertain in IPCC-class models. They use reconstructions of temperature and atmospheric CO₂ to retrieve parameters that regulate the sensitivity of carbon losses from land and the oceans to a temperature increase (γ) and the sensitivity of carbon storage on land and in the oceans to a 1 ppm change in CO₂. The paper makes a nice conceptual contribution by linking the concept of the transient climate response of cumulative emissions (TCRE) with γ , β , and another carbon cycle parameter, α , the sensitivity of temperature to a change in atmospheric CO₂. I think this is new and interesting. They use this equation and observations to estimate β and γ for the historical era, and I think this is really interesting and important.

[Response B1]: We thank the reviewer very much for highlighting the importance of our work on the method and the observational estimates of β and γ . In the new version, we have carefully revised the manuscript and the supporting information document according to your constructive comments.

[Comment B2] In the abstract, the authors claim that the β sensitivity term is constant across time scales, around 3 PgC/ppm and that the γ parameter becomes more negative as a function of timescale from about -30 PgC/K over a period of a decade to less than -100 Pg C/K over a century. The authors then claim that the IPCC class models have overestimated γ and the gain of the carbon climate feedback. As a consequence, the authors argue that collectively, we have more allowable fossil fuel emissions (14%) before we cross the warming thresholds for the Paris agreement.

It is with the assertion of a constant β across timescales and their assertion that the γ derived from a period with a low emissions state that can be compared to model estimates of γ from high emissions trajectories that for me raises important conceptual concerns.

Specifically, while β is remarkably constant as a function of time scale from the historical observations shown in Fig 2c, this does not mean it would be the same for another trajectory of CO₂ forcing, such as the one from the 1% CMIP5/CMIP6 or the A2 scenario

from CMIP. This is because the rate of CO₂ uptake from the atmosphere by the land and ocean system in response to a 1 ppm increase depends on the timescale over which the ocean and land pools are allowed to equilibrate. So the comparison in Figure 3a between the observations and the different model estimates of beta I believe to be flawed.

It does not surprise me that the CMIP5 models have a smaller beta because the rate of CO₂ increase is much faster for the 1% per year idealized CO₂ trajectory than what has occurred during the past 100 years, allowing less time for carbon to move through deeper ocean layers and more slowly turning over coarse woody debris and soil pools in the models. Gloor et al. (2010) explored this phenomena as it relates to interpretation of the airborne fraction and the conclusions from this analysis are critically relevant here. It would be better to capture the behavior of an individual model from CMIP5 using reduced complexity box model, and then force this model with the observed trajectory of atmospheric CO₂ (and recompute beta (and gamma)) to make a fairer comparison between the observations and the models.

[Response B2]: Thanks very much for this constructive comment. We agree that while the observational beta is found nearly constant across time scale, it does not mean that this is the same for another trajectory of CO₂ forcing. Firstly, As explained by Gloor et al. (2010) and Raupach et al. (2013), the beta keeping constant over time is because of the exponential growth (with $y = 0.27e^{0.018t}$) of atmospheric CO₂ or CO₂ emission during the 1880-2017 period. To explain this phenomenon, Raupach et al. (2013) proposed the LinExp theory, in which the carbon-climate system over the industrial period can be approximated as a linear system (Lin) of carbon cycle forced by the exponentially growing CO₂ emissions (Exp) over 1880-2017, then all ratios of responses to forcings are constant, e.g. the $AF = \Delta C_A / \Delta C_E$, or $\beta = \Delta C_B / \Delta C_A$. Because the CMIP5 1pctCO2 experiment (1% CO₂ growth per year) and the C⁴MIP A2 scenario experiment (~0.7% CO₂ growth per year) were forced by the exponentially growing atmospheric CO₂, the estimated beta for both CMIP5 and C⁴MIP were also nearly constant over time scale (Supplementary Figs. 7-8 in revised version). Secondly, we agree that it is unfair to compare the observational and model-based beta from the CO₂ trajectories with different growth rates, as the beta become smaller under higher rates of CO₂ increase. In the revised version, we calculated C⁴MIP-based β (i.e. β^{BGC}) for two time

period: 1880-2017 and 2018-2100. Then we compared the observational beta with C⁴MIP-based β^{BGC} for the same period of 1880-2017. Result shows that the C⁴MIP-based β^{BGC} for 1880-2017 is 3.07 ± 0.68 GtC ppm⁻¹, close to the observation-based β (3.22 ± 0.32 GtC ppm⁻¹). While the C⁴MIP-based β^{BGC} for 2018-2100 is 2.32 ± 0.59 GtC ppm⁻¹, about 24% smaller than that for 1880-2017 (Fig. 3a in revised version). Please check the revisions in Lines 303-317 in the revised manuscript.

[Comment B3] For the same reason, I don't believe it's fair to use the beta from the observational record to then separate out gamma from the observations during the 1000-year pre-industrial era. Beta is probably much larger than that derived from the industrial era, and perhaps the estimates of gamma on longer time scales are considerably underestimated.

[Response B3]: We agree that the beta derived from the 1000-year pre-industrial period would likely be larger than the observational beta from the industrial period of 1880-2017, due to that higher growth rates of atmospheric CO₂ would induce smaller beta as shown in our results and previous modeling studies (Hajima et al., 2014; Randerson et al., 2015). Here, in this study, we could not estimate beta directly from the observations during the 1000-year pre-industrial period using our Fourier analysis-based climate-carbon cycle feedback framework (equation (2)), as the TCRE⁻¹ for the pre-industrial period may not exist (emissions were small and may have no significant effect on global temperature). Therefore, to draw out the timescale-dependence of gamma across time scales for the 1000-year pre-industrial period, we applied the industrial observational beta to the pre-industrial period (equation (3)). We also have done an error sensitivity analysis to test how much uncertainty in gamma for the pre-industrial period would increase from the uncertainty in beta. We show that an overestimation (or underestimation) of 50% in β would induce an underestimation (or overestimation) of about 30% in γ on timescales over 10 to 1000 years for the PILM period. This change of 30% (~ 25 GtC K⁻¹) in γ for the pre-industrial period induced from the 50% uncertainty in β is still smaller than those from the large divergences in ice-core CO₂ records and reconstructed temperatures (Figure B1, or Supplementary Fig.5 in revised version). This assumption might be acceptable, as we are focusing on the timescale-dependence of gamma across time scales for the pre-industrial period.

Figure B1. Error sensitivity analysis of γ on different timescales in response to changes in β . The β is $3.22 \text{ GtC ppm}^{-1}$ with an uncertainty of 10%, estimated from observational datasets over 1880-2017. We applied this value of β to estimate pre-industrial γ on timescales from 10 to 1000 years using the EnOBS ensemble η across timescales from $>1,500$ members (based on 3 ice-core CO_2 records $\times 521$ calibrated temperature reconstructions (Frank et al., 2010)). This figure shows that when the β has overestimations (or underestimations) of 10% to 50% of its industrial observation-based value ($3.22 \text{ GtC ppm}^{-1}$), the γ shows underestimations (or overestimations) of about 6% to 30%. This largest change of 30% ($\sim 25 \text{ GtC K}^{-1}$) in $\gamma_{100\text{yr}}$ is still smaller than the uncertainty of $\gamma_{100\text{yr}}$ (a standard deviation of 41.90 GtC K^{-1}) that was mainly caused by the large divergences in the three ice-core CO_2 records and reconstructed temperatures.

[Comment B4] For gamma, another important issue is raised by Schwinger et al. 2014. These authors show that the barriers to CO_2 inflow from ocean mixed layer shoaling to transient CO_2 in a fully coupled simulation generates a fundamentally different gamma than the gamma obtained from response of the ocean to warming in the absence of changing CO_2 (ie diagnosing gamma from the radiative run rather than the difference between the fully coupled and biogeochemically coupled simulations). It is the latter that may be most analogous to the evolution of the climate-carbon cycle system over the last millennium.

[Response B4]: Thanks very much for this constructive comment. We have realized that the nonlinearity of climate-carbon feedback as raised by Schwinger et al. 2014 was ignored in our previous version. In the revised version, have made major revisions to address the “nonlinear feedback” issue. Firstly, we renewed our climate-carbon cycle feedback framework by incorporating the nonlinearity feedback term (see equations (1-2) and equations (5-8) in the revised version). As defined by Schwinger et al., 2014, when assuming the carbon stock in biosphere ($C_B = C_L + C_O$) at the reference climate state as a function of climate and CO₂: $C_B = F(C_A, T_A)$, then β , γ , and $f(\beta, \gamma)$ can be expressed as the 1st order and 2nd order coefficients of the Taylor series of C_B since the initial time ($t=0$): $\beta = \frac{\partial F}{\partial C_A} \Big|_0$, $\gamma = \frac{\partial F}{\partial T_A} \Big|_0$, and $f(\beta, \gamma) = \frac{\partial^2 F}{\partial C_A \partial T_A} \Big|_0 + \frac{1}{2} \frac{\partial^2 F}{\partial C_A^2} \Big|_0 \frac{\Delta C_A}{\Delta T_A} + \frac{1}{2} \frac{\partial^2 F}{\partial T_A^2} \Big|_0 \frac{\Delta T_A}{\Delta C_A} + R_3$. The nonlinear feedback $f(\beta, \gamma)$ in this study represents the 2nd and high-order terms of the Taylor expansion. As previous studies mainly focused on the nonlinearity of the carbon-climate (γ -) feedback (Gregory et al., 2009; Schwinger et al., 2014; Zickfeld et al., 2011), in this study, we combined the γ -feedback and the atmospheric CO₂ change’s impacts on the nonlinear feedback as: $\gamma^* = \gamma + f(\beta, \gamma)\Delta C_A$ (see revisions on Lines 543-567 in Methods in the revised version). Secondly, we tried to quantify the nonlinear feedback term $f(\beta, \gamma)$ and its contribution to the γ -feedback $f(\beta, \gamma)\Delta C_A$ from the CMIP5 models’ three groups of simulations: the COU, BGC and RAD runs, and the COU and BGC runs of C4MIP models. We defined and calculated the direct β -feedback from the BGC simulations ($\beta^{BGC} \approx \Delta C_B^{BGC} / \Delta C_A^{BGC}$) and the direct γ -feedback from the RAD simulations ($\gamma^{RAD} = \Delta C_B^{RAD} / \Delta T_A^{RAD}$) and the indirect γ -feedback from the COU-BGC simulations ($\gamma^{COU-BGC} \approx (\Delta C_B^{COU} - \Delta C_B^{BGC}) / \Delta T_A^{COU}$), for the observation-overlapped period of 1880-2017 and the future emission scenario of 2018-2100 for the C⁴MIP models, and the 1pctCO₂ 140-year period for the CMIP5 models, respectively (equations (22-26) in the revised version). We also estimated the nonlinear feedback from the difference between COU simulations and the BGC and RAD simulations ($f(\beta, \gamma) = [\Delta C_B^{COU} - (\Delta C_B^{BGC} + \Delta C_B^{RAD})] / \Delta C_A^{COU} \Delta T_A^{COU}$) and its contribution to γ -feedback ($f(\beta, \gamma)\Delta C_A^{COU}$) for the CMIP5 models. Results have been showed in Fig. 3a, c and the Supplementary Tables (3-4) in the revised version. We estimated that the CMIP5-based nonlinear feedback $f(\beta, \gamma)$ for the 1pctCO₂ 140-year period was $-11.22 \pm 11.72 \times 10^{-3}$ GtC

$\text{ppm}^{-1} \text{K}^{-1}$, and its contribution to the γ -feedback was $-9.6 \pm 10.03 \text{ GtC K}^{-1}$, which means that the γ^{RAD} was about 15% smaller in magnitude than the $\gamma^{COU-BGC}$ feedback. We also have added the results in the manuscript (Lines 280-333).

[Comment B5] The conclusion by the authors that the gain of the climate carbon feedback is too large (mean of 0.13 from CMIP5 depends on comparisons between models and observations that have fundamentally different trajectories of CO₂ and temperature forcing. For this reason, I do not the implications regarding the Paris Accord are at all supported by the authors' analysis.

It could be right, but to prove it, a model that mimics the behavior of the CMIP5 carbon cycle models would need to then be used to simulate the observed 20th century historical period with the identical CO₂ forcing and warming. And the same model should be used for the future projection of allowable emissions to match the Paris Accord. It might be tricky to account for other forcing agents (aerosols, CH₄, etc.) in the delta temperature here, but this seems like it has to be the path forward. This could be done independently of the analysis of the last millennium and I might suggest the authors consider this for narrowing and strengthening the analysis, which has interesting and novel elements, but currently makes many comparisons for parameters that to me, appear fundamentally tied to the specific model scenarios and observational periods from which they are generated.

[Response B5]: We agree that it is unfair to compare the observational and model-based beta from the CO₂ trajectories with different growth rates. In the revised version, we calculated C⁴MIP-based parameters for two time period: 1880-2017 and 2018-2100. We compared the observation-based and C⁴MIP-based feedback parameters for the same period 1880-2017 with the same CO₂ trajectory.

Result shows that the C⁴MIP-based β^{BGC} for 1880-2017 is $3.07 \pm 0.68 \text{ GtC ppm}^{-1}$, close to the observation-based β ($3.22 \pm 0.32 \text{ GtC ppm}^{-1}$). While the C⁴MIP-based gain factor (g) for the same period 1880-2017 was 0.09 ± 0.04 , larger than the observational value (0.01 ± 0.05) by about an order of magnitude (Fig. 3c in the revised version). As a result, the observation-based and C⁴MIP-based feedback amplification G ($G = 1/(1 - g)$, see Methods) are 1.01 ± 0.05 and 1.10 ± 0.04 respectively, suggesting the modelled amplification

effect is about $9\pm 7\%$ larger. Using the observation-based g (0.01 ± 0.05) and the new C⁴MIP-based g (0.09 ± 0.04) for the same period 1880-2017, we estimated allowable emissions would be $9\pm 7\%$ more, or 125 ± 8 GtC. Please check these revisions in the revised manuscript (Lines 362-368, 383-388).

Specific comments:

[Comment B6] *Abstract. Beta is a parameter that contributes to the carbon concentration feedback, but is not the carbon concentration feedback itself. Same for gamma, please consider rewriting this sentence. The carbon climate feedback, for example, depends on both gamma and beta as the authors later show and understand. So please consider revising nomenclature here and in introduction (line 65).*

[Response B6]: Thanks for this detailed comment. We agree that both beta and gamma parameters which represent the sensitivities of land and ocean carbon storages to 1 ppm change in atmospheric CO₂ and 1 K change in global temperature, respectively. Following the use by ref. (Gregory et al., 2009), we have revised the manuscript to notate the carbon-concentration feedback response parameter as β and the carbon-climate feedback response parameter as γ . See Lines 36-37, 66-68.

[Comment B7] *“From the perspective of the atmosphere, beta is positive and gamma is negative.” Line 70. Isn't this the opposite, where beta is positive defined from the perspective of accumulation in the land and ocean? Same for gamma (a negative gamma means loss from the land and ocean reservoirs, but a positive gain in the atmospheric carbon pool).*

[Response B7]: Agreed and we have it to “From the perspective of the land and ocean reservoirs, beta is positive and gamma is negative.” In the revised version (Lines 71-72)

[Comment B8] *Equation 1. 'm' has been used to represent the conversion ratio of Pg C/ppm in past carbon cycle work (I think in Arora et al.). Or maybe its the inverse. Anyway, using m might be better than inserting a 0.472 constant in many places.*

[Response B8]: Agreed and have defined $m=2.12$ GtC ppm⁻¹ in the revised version.

[Comment B9] Lines 132-136. The reason n is so much larger for contemporary and future periods also has to do with radiative forcing from CO₂ saturating in the wings of the 8-12 μm outgoing longwave band at higher absolute CO₂ levels.

[Response B9]: Agreed. Thanks for this comment and we have added related explanations in the revised version (please check Lines 191-195).

[Comment B10] I don't understand in the text and in Figures 2 and 3 how the Nyquist frequency in Fourier analysis factors in. The x axis in Figure 2 spans over 100 years (a, b) and Figure 3

is 1000 years, even though the record for the historical is about 140 years, and in Fig 1 the millennium period considered is about 850 years. Please revise or describe how it is possible to resolve (and show error bars for) estimates that have a period the same as length of the observed record.

[Response B10]: Thanks for this detailed comment. The Figure 2a-b in the manuscript shows that amplitudes of CO₂ and T_A for 1880-2017 from Fourier analysis, thus the x axis in Figure 2 spans over 1-140 years. While the Figure 3b-d (now has been moved to the Supporting Information document, Supplementary Fig. 6) shows the η and γ estimates across timescales of 10-850 years for the 1000-1850 from Fourier analysis, thus the x axis in Supplementary Fig. 6 spans over 1-1000 years. One can find that the maximal value (850 years) of timescale of η and γ do not reach the maximal major tick mark (the 10³ years), it is just a bit larger than the fourth minor tick mark (the 800-year) between the 10² and the 10³ years.

[Comment B11] Line 134. "Over ... " That is a really long sentence and I got lost in the middle of it. Maybe some part of a sentence was cut out or lost here?

[Response B11]: Thanks for this detailed comment. We have removed this sentence, and the related sentence has been shown in Lines 184-191 in the revised version, which reads “*The reason for this large discrepancy in η between the pre-industrial and industrial periods is likely a consequence of the nonlinear dependence of radiative forcing on atmospheric CO₂ concentrations (Myhre et al., 1998), and the temperature change in response to the increase*

in atmospheric CO₂ during the industrial era has not reached steady state. It is also well known that equilibrium climate sensitivity is often considerably larger than the transient climate sensitivity (He et al., 2017)."

Reference:

- Hajima, T., Tachiiri, K., Ito, A. and Kawamiya, M., 2014. Uncertainty of Concentration–Terrestrial Carbon Feedback in Earth System Models*. *J. Clim.*, 27(9): 3425-3445.
- Gloor, M., Sarmiento, J. L. & Gruber, N. What can be learned about carbon cycle climate feedbacks from the CO₂ airborne fraction? *Atmospheric Chemistry and Physics* 10, 7739-7751 (2010).
- Randerson, J.T. et al., 2015. Multicentury changes in ocean and land contributions to the climate-carbon feedback. *Glob. Biogeochem. Cycles*, 29(6): 744-759.
- Raupach, M. R. The exponential eigenmodes of the carbon-climate system, and their implications for ratios of responses to forcings. *Earth System Dynamics* 4, 31-49 (2013).
- He, J., Winton, M., Vecchi, G., Jia, L. and Rugenstein, M., 2017. Transient Climate Sensitivity Depends on Base Climate Ocean Circulation. *J. Clim.*, 30(4): 1493-1504.
- Myhre, G., Highwood, E.J., Shine, K.P. and Stordal, F., 1998. New estimates of radiative forcing due to well mixed greenhouse gases. *Geophys. Res. Lett.*, 25(14): 2715-2718.

To Reviewer #3:

[Comment C1] The study is addressing an important topic of the climate-carbon cycle feedback. The manuscript is well written, but the study has ignored the inherent nonlinearity of the carbon-cycle framework that is set out in a substantial study by Schwinger et al. (2014), Nonlinearity of Ocean Carbon Cycle Feedbacks in CMIP5 Earth system models, Journal of Climate.

Unfortunately I have a problem with the central part of the manuscript with its focus on time-dependence of the climate-carbon cycle feedback. The study ignores how the carbon-cycle feedback parameters are defined and the inherent nonlinearity in their framework, which is clearly set out in Schwinger et al. (2014). In this study, the carbon cycle feedback parameters, beta and gamma, are based on a Taylor expansion relative to the pre-industrial state; see equation (2) in Schwinger et al. (2014). This approach was taken by the original study of Friedlingstein et al. (2003), but is more completely set out by Schwinger et al. (2014). In more detail, the change in the carbon inventory depends on a linear sum of first order differential terms involving T and CO₂ plus further second order and higher order differential terms, where all the differential terms are evaluated relative to the preindustrial. The beta and gamma terms are defined by the first order differential terms evaluated at the time of the preindustrial with the second order and higher differential terms neglected, such that $\beta = dF/dCO_2$ at the pre industrial and $\gamma = dF/dT$ at the preindustrial, where F is a function defining the climate system. Schwinger et al. (2014) explicitly state that the shortcoming of this approach is that there is no accounting of time dependence of inventory changes. In addition, Schwinger et al. (2014) demonstrate that the ocean carbon-cycle feedbacks are inherently nonlinear.

Given the Schwinger et al. (2014) study, I am not convinced that the present manuscript is robust. The manuscript estimates the time-dependence of the carbon cycle parameters by a Fourier series fit over different time periods. However, the beta and gamma parameters are then not still evaluated at the preindustrial as they should be, but instead are evaluated at the instantaneous time. If the terms are evaluated at the instantaneous time, then the original Taylor expansion does not hold that was used to define beta and gamma.

Evaluating the beta and gamma terms at different times probably effectively means that

the neglected high-order differential terms are being melded into their estimates, so that there is an issue of errors arising from the nonlinearity of the framework.

I am aware that there are prior studies that have evaluated the time-dependence of the carbon-cycle feedback parameters, but the Frank et al. (2010) study and the Willeit et al. (2014) were either before or unaware of the Schwinger et al. (2014) study. The authors can of course choose to evaluate these differentials dF/dCO_2 and dF/dT at any time, but they should not then equate them to beta and gamma, or expect the actual linearisation of the carbon budget to hold, so that the wider implications of their study is then lost.

[Response C1]: Many thanks to the reviewer for this constructive comment. Following this comment, we have realized that the nonlinearity of climate-carbon feedback as raised by Schwinger et al. 2014 was ignored in our previous version. In the revised version, we have made major revisions to address the “nonlinear feedback” issue.

Firstly, we updated our climate-carbon cycle feedback framework by incorporating the nonlinearity feedback term as a function of β and γ parameters, i.e. $f(\beta, \gamma)$ in a unit of $GtC\ ppm^{-1}\ K^{-1}$ or $GtC\ GtC^{-1}\ K^{-1}$, in a CO_2 emission-driven coupled climate-carbon cycle system (see also equations (1-2) and equations (5-8) in the revised version),

$$\Delta C_E = \Delta C_A + \beta \Delta C_A + \gamma \Delta T_A + f(\beta, \gamma) \Delta C_A \Delta T_A \quad (C1)$$

As defined by Schwinger at al., 2014, when assuming the carbon stock in biosphere ($C_B = C_L + C_O$) at the reference climate state as a function of climate and CO_2 : $C_B = F(C_A, T_A)$, then β , γ , and $f(\beta, \gamma)$ can be expressed as the 1st order and 2nd order coefficients of the Taylor series of C_B since the initial time ($t=0$): $\beta = \frac{\partial F}{\partial C_A} \Big|_0$, $\gamma = \frac{\partial F}{\partial T_A} \Big|_0$, and $f(\beta, \gamma) = \frac{\partial^2 F}{\partial C_A \partial T_A} \Big|_0 + \frac{1}{2} \frac{\partial^2 F}{\partial C_A^2} \Big|_0 \frac{\Delta C_A}{\Delta T_A} + \frac{1}{2} \frac{\partial^2 F}{\partial T_A^2} \Big|_0 \frac{\Delta T_A}{\Delta C_A} + R_3$. The initial time for the study for the industrial period 1880-2017 was set at 1880 for both observations and C4MIP models. The nonlinear feedback $f(\beta, \gamma)$ in this study represents the 2nd and high-order terms of the Taylor expansion as presented in Schwinger at al., 2014. As previous studies mainly focused on the nonlinearity of the carbon-climate (γ -) feedback (Gregory et al., 2009; Schwinger et al., 2014; Zickfeld et al., 2011), in this study, we combined the γ -feedback and the atmospheric CO_2 change’s impacts on the nonlinear feedback as: $\gamma^* = \gamma + f(\beta, \gamma) \Delta C_A$ (see revisions on Lines 543-567 in Methods in the revised version), and,

$$\Delta C_E = (1 + \beta)\Delta C_A + \gamma^*\Delta T_A. \quad (C2)$$

Then, using the Fourier analysis-based approach raised by our study (equation (2) and Fig.2a in the revised version), we estimated the value of β for the 1880-2017 to be 3.22 ± 0.32 GtC ppm⁻¹. From equation (C2), we then estimated the γ^* to be -10.9 ± 3.6 GtC K⁻¹ for the period 1880-2017.

Secondly, we found it difficult to directly separate the nonlinear feedback contribution from observation-based γ^* from our current feedback analysis. We then tried to quantify the nonlinear feedback term $f(\beta, \gamma)$ and its contribution to the γ -feedback $f(\beta, \gamma)\Delta C_A$ from the CMIP5 models' three groups of simulations: the COU, BGC and RAD runs, and the COU and BGC runs of C4MIP models. We defined and calculated the direct β -feedback from the BGC simulations ($\beta^{BGC} \approx \Delta C_B^{BGC} / \Delta C_A^{BGC}$) and the direct γ -feedback from the RAD simulations ($\gamma^{RAD} = \Delta C_B^{RAD} / \Delta T_A^{RAD}$) and the indirect γ -feedback from the COU-BGC simulations ($\gamma^{COU-BGC} \approx (\Delta C_B^{COU} - \Delta C_B^{BGC}) / \Delta T_A^{COU}$), for the observation-overlapped period of 1880-2017 and the future emission scenario of 2018-2100 for the C⁴MIP models, and the 1pctCO₂ 140-year period for the CMIP5 models, respectively (equations (22-26) in the revised version). We also estimated the nonlinear feedback from the difference between COU simulations and the BGC and RAD simulations ($f(\beta, \gamma) = [\Delta C_B^{COU} - (\Delta C_B^{BGC} + \Delta C_B^{RAD})] / \Delta C_A^{COU} \Delta T_A^{COU}$) and its contribution to γ -feedback ($f(\beta, \gamma)\Delta C_A^{COU}$) for the CMIP5 models. Results have been showed in Fig. 3a, c and the Supplementary Tables (3-4) in the revised version. We estimated that the CMIP5-based nonlinear feedback $f(\beta, \gamma)$ for the 1pctCO₂ 140-year period was $-11.22\pm 11.72\times 10^{-3}$ GtC ppm⁻¹ K⁻¹, and its contribution to the γ -feedback was -9.6 ± 10.03 GtC K⁻¹, which means that the γ^{RAD} was about 15% smaller in magnitude than the $\gamma^{COU-BGC}$ feedback, while its contribution ($f(\beta, \gamma)\Delta T_A$) to the β -feedback is negligible (3%). Thus, we indicate that the non-linear feedback term has negligible effect on the estimate of the slope of $(1 + \beta_k)$ (see Table C1). We have added the results in the manuscript (Lines 280-330).

Table C1. Estimates of β , γ and $f(\beta, \gamma)$ for the nine CMIP5 models from the 1pctCO₂ climate-carbon cycle feedback experiments.

Model	β^{BGC} (GtC ppm ⁻¹)	$f(\beta, \gamma)\Delta T_A$ (GtC ppm ⁻¹)	$\gamma^{COU-BGC}$ (GtC K ⁻¹)	γ^{RAD} (GtC K ⁻¹)	$f(\beta, \gamma)\Delta C_A$ (GtC K ⁻¹)	$f(\beta, \gamma)$ (GtC ppm ⁻¹ K ⁻¹)
BCC-CSM1	2.06	-0.045	-89.95	-86.22	-9.71	-11.35×10 ⁻³
CanESM2	1.66	0.004	-75.75	-78.82	0.64	0.75×10 ⁻³
CESM1-BGC	0.96	0.022	-17.27	-23.77	5.04	5.89×10 ⁻³
HadGEM2	1.92	-0.185	-63.34	-44.04	-28.71	-33.57×10 ⁻³
IPSL-CM5A-LR	2.05	-0.036	-63.06	-65.13	-6.08	-7.15×10 ⁻³
MIROC-ESM	1.55	-0.041	-104.15	-100.14	-6.3	-7.41×10 ⁻³
MPI-ESM-LR	2.30	-0.111	-102.38	-88.09	-18.98	-22.19×10 ⁻³
NorESM-ME	1.11	-0.042	-21.69	-14.08	-9.76	-11.41×10 ⁻³
UVic ESCM2.9	1.74	-0.058	-93.67	-85.44	-12.59	-14.60×10 ⁻³
Ensemble Mean	1.71	-0.055	-70.14	-65.08	-9.6	-11.22×10⁻³
+/-S.D.	+/-0.44	0.062	+/-32.43	+/-30.74	+/-10.03	+/-11.72×10⁻³

[Comment C2] My other concerns are more minor. The theory introduced in (1) and the Methods in equation (3) would be clearer if all carbon inventories were quoted in GtC or PgC, rather than have the atmospheric inventory and carbon emission in ppm. Making the units the same for all the carbon variables (that have the same symbol) would avoid the 0.472 conversion factors being included.

[Response C2]: Agreed. Thanks very much for this comment. We have quoted all carbon inventories in GtC in the revised version. The conversion factor $m=2.12$ GtC ppm⁻¹ was applied in equation (3) and equations in the Methods.

[Comment C3] The transient climate response to emissions, TCRE, is a widely used climate metric. The manuscript would be better advised to focus on that variable, rather than its reciprocal.

[Response C3]: Agreed. We have used $TCRE$ or $TCRE^{-1}$ in place of ξ in the revised version.

[Comment C4] In the methods, equations (4) and (5) should be estimated at the same reference time, usually taken to be the pre industrial.

[Response C4]: Agreed. Thanks for this detailed comment. We have mentioned that the reference time is 1850 in the revised version (Lines 548-551).

[Comment C5] In the methods, the variables that are time dependent should be explicitly defined in equations (3) to (7). Based on Schwinger et al. (2014), beta and gamma terms should not be time dependent.

[Response C5]: Thanks for this constructive comment. Following the definition by Schwinger et al. (2014), we defined $\beta = \frac{\partial F}{\partial C_A} |_0$, $\gamma = \frac{\partial F}{\partial T_A} |_0$ as the first order coefficients of the Taylor series of $C_B = F(C_A, T_A)$, where $C_B = C_L + C_O$ is the carbon stock at a reference climate state ($t=0$). For the feedback analysis for the industrial period 1850-2017 of observation and C⁴MIP models, the reference climate state is defined as the climate state at the year 1850. For the analysis for the future CO₂ emissions scenarios of C⁴MIP models, the reference climate state is defined as the climate at the year 2018 simulated by models. Thus, in our study, beta and gamma could change with climate state and are time dependent. Many previous studies have demonstrated that both beta and gamma are timescale dependent. Willeit et al. (2014) and Frank et al. (2010) showed that gamma changed over different time periods before the industrial period. Arora et al. (2013) showed that gamma was getting more negative over time with temperature increase from the 1pctCO₂ CMIP5 experiments. Modeling studies from Gregory et al. (2009), Hajima et al. (2013), and Randerson et al. (2014) have showed that gamma was time period dependent, and the carbon-concentration feedback parameter (beta) would get smaller with higher growth rates of atmospheric CO₂ or emissions as forcing. Raupach et al. (2014) using the LinExp theory explained that the constancy of the airborne fraction (and beta) over industrial period was caused by nearly exponentially growing CO₂ forced by exponentially growing emissions.

[Comment C6] In summary, the manuscript is focusing on evaluating the time dependence of the carbon-cycle parameters without taking into account the time state that these differentials are evaluated at and ignoring the nonlinearity from the neglected higher order terms. The study needs to reconcile their approach with the Schwinger et al. (2014) study that highlights the inherent nonlinearity of the carbon-cycle framework and the requirement to evaluate beta and gamma at the same reference time. While the manuscript makes many inferences for beta and gamma for different time periods, it is difficult to judge their value unless the estimates are referenced to the same time point and the error from the neglected nonlinear terms are accounted for.

[Response C6]: Thanks very much for this constructive comment. We agree with the reviewer that our previous version of manuscript did not take into account the nonlinearity of climate-carbon cycle feedback term within the carbon-concentration feedback and the climate-carbon feedback. We have made lots of improvements in this version to address the ‘nonlinear’ comment. We reconciled approaches with the Schwinger et al. (2014) study and other studies (Arora et al., 2013, 2019; Zickfeld et al., 2011; Gregory et al., 2009; Friedlingstein et al., 2006), and incorporated the nonlinear feedback term as a function of beta and gamma parameters into the novel Fourier analysis-based climate-carbon cycle feedback framework. Using this new framework, we recalculated the beta and gamma* (including the nonlinear feedback contribution) for the industrial period 1880-2017. We set the same reference time at 1880 for calculating observational and model-based beta and gamma for the period 1880-2017. We estimated the nonlinear feedback term from the CMIP5 1pctCO2 experiments to show that its contribution to the climate-carbon feedback was about 15%, while its contribution ($f(\beta, \gamma)\Delta T_A$) to the carbon-concentration feedback is negligible (3%) (see Table C1 in Response C1).

References:

Arora, V.K. et al., 2013. Carbon–Concentration and Carbon–Climate Feedbacks in CMIP5 Earth System Models. *J. Clim.*, 26(15): 5289-5314.

- Arora, V.K. et al., 2019. Carbon-concentration and carbon-climate feedbacks in CMIP6 models, and their comparison to CMIP5 models. *Biogeosciences Discuss.*, in review.
- Cheng, L. et al., 2017. Warming enhances old organic carbon decomposition through altering functional microbial communities. *ISME J*, 11(8): 1825-1835.
- Frank, D.C. et al., 2010. Ensemble reconstruction constraints on the global carbon cycle sensitivity to climate. *Nature*, 463(7280): 527-30.
- Hajima, T., Tachiiri, K., Ito, A. and Kawamiya, M., 2014. Uncertainty of Concentration–Terrestrial Carbon Feedback in Earth System Models*. *J. Clim.*, 27(9): 3425-3445.
- Gregory, J.M., Jones, C.D., Cadule, P. and Friedlingstein, P., 2009. Quantifying Carbon Cycle Feedbacks. *J. Clim.*, 22(19): 5232-5250.
- Randerson, J.T. et al., 2015. Multicentury changes in ocean and land contributions to the climate-carbon feedback. *Glob. Biogeochem. Cycles*, 29(6): 744-759.
- Raupach, M. R. 2013. The exponential eigenmodes of the carbon-climate system, and their implications for ratios of responses to forcings. *Earth System Dynamics* 4, 31-49.
- Schwinger, J. et al., 2014. Nonlinearity of Ocean Carbon Cycle Feedbacks in CMIP5 Earth System Models. *J. Clim.*, 27(11): 3869-3888.
- Willeit, M., Ganopolski, A., Dalmonch, D., Foley, A.M. and Feulner, G., 2014. Time-scale and state dependence of the carbon-cycle feedback to climate. *Clim. Dyn.*, 42(7-8): 1699-1713.

Zickfeld, K., Eby, M., Matthews, H.D., Schmittner, A. and Weaver, A.J., 2011.

Nonlinearity of Carbon Cycle Feedbacks. *J. Clim.*, 24(16): 4255-4275.

REVIEWER COMMENTS

Reviewer #1 (Remarks to the Author):

Review of "A small climate-amplifying effect of climate-carbon cycle feedback"

As I have described in my review of the previous version of the manuscript, in my opinion the study is interesting and has the potential to influence and revise the allowable carbon emissions before reaching warming targets. The revised version nicely demonstrates how the carbon cycle feedbacks behave on different time scales. The study provides new estimates of the carbon cycle feedbacks based on observations and discuss how comparable or not are these observation-based estimates with those from earth system models, on different time periods and under different CO₂ increase trajectory. The authors have clearly taken my recommendations and comments into consideration and included new analysis along with clarifications, as well as rewritten parts of the manuscript to highlight how meaningful is to compare estimates of beta and gamma from different eras and under different emissions rate. Most of my previous concerns have been addressed in the new manuscript and I appreciate the authors' effort. However I still have some concerns/reservations about the statement for the amplification effect from the carbon cycle feedbacks based on the gain factor estimated from the observations and the C4MIP models, given that these estimates are based on a different method. Hence, I recommend some minor revisions (or at least some clarifications) as I explain below for the manuscript to be accepted for publication.

Specific comments:

1. Estimates of beta and gamma* during 1880-2017 and their comparison: In my understanding beta and gamma* from the observations are estimated based on equation 2, and so essentially by the slope and intercept of alpha_k vs TCRE_k in figure 2. However, beta and gamma during 1880-2017 for the C4MIP models (as reported in lines 303 and 322) are estimated using the FEA approach (with their timescale dependence estimated using supplementary equations S5 and S6). Hence, I wonder how much of the difference between observations and C4MIP is actually due to the method used. In my opinion a more direct comparison between the observations-based and the model-based beta and gamma* will be to estimate them using the same method, such that for the C4MIP you estimate beta and gamma as a slope and an intercept using the alpha_k and TCRE_k from the C4MIP models. Hence, I suggest that for the 1880-2017 time period only, for the C4MIP together with the estimate of beta and gamma* from FEA you report the estimates of beta and gamma based on your approach (equation 2).

2. Estimates of feedback gain, lines 343-368 and associated methods and supplementary. I am somewhat perplexed as why is the airborne fraction used to estimate the gain factor rather directly gamma*, alpha and beta as in equation 9. Then the relative contribution from gamma* and beta to the gain is more straightforward. I appreciate that the argument here is that AF is relative constant but in my understanding this is associated/reflected largely by beta being relative constant. Nevertheless, my main issue is that, in my understanding, the gain factor based on observations is estimated using the beta estimate of 1.52 GtC/GtC based on equation 2. However, the gain factor based on C4MIP is estimated using the beta estimate from the FEA approach. In my opinion, a beta and gamma* estimate for C4MIP derived from the same method as the observation-based estimate (equation 2) should be used (this relates to my above comment 1) to estimate the gain for C4MIP during the 1880-2017. Further, in my opinion equation 9 should be used directly to estimate the gain rather than the airborne fraction.

Minor suggestions for clarity/typos:

3. Line 80 and 84: Use of "idealized model experiment". I understand that the authors mean idealised experiments with Earth system models as I am familiar with the referenced studies. However, I think

for clarity and to avoid any confusion with use of idealised models rather than Earth system models, I suggest to rephrase to something along the lines of “.. Earth system models under idealised experiments”.

4. Lines 172-177. In Supplementary Figure 2, there is a somewhat larger divergence of the predicted and observed temperature after 1980's which you mention at lines 700-701. Is it correct to presume that this divergence after the 1980's is linked with the γ^* not actually being constant during 1880-2017. If yes, maybe the authors could mention in this paragraph (172-177) something along the lines that the larger predicted trend for temperature relative to the observed after 1980's is associated with γ^* treated/assumed as constant in these estimates.

5. Line 213-214 typo: there is an extra that before “ ... the constant value”

Reviewer #3 (Remarks to the Author):

Review of Zhang et al. “A small climate-amplifying effect of climate-carbon cycle feedback”

This is my second review of the manuscript.

In my first review, I raised a problem with the central part of the manuscript with its focus on time-dependence of the climate-carbon cycle feedback. The study ignores how the carbon-cycle feedback parameters are defined and the inherent nonlinearity in their framework, which is clearly set out in Schwinger et al. (2014). In this study, the carbon-cycle feedback parameters, β and γ , are based on a Taylor expansion relative to the pre-industrial state; see equation (2) in Schwinger et al. (2014).

I am very pleased that the authors have dealt with that major concern in explicitly now evaluating the nonlinear feedbacks as part of their study. They find that nonlinear effect is small, which is in contrast to other prior studies. So some further explanation is needed, which reconciles their finding and the prior studies that the nonlinear effect is large (e.g. Gregory et al., 2009; Schwinger et al., 2014). I suspect that each approach is making their assessment by comparing against a different measure.

I do have a further major concern that links back to the original Taylor expansion used to define the carbon-cycle feedback parameters. As repeated in their Methods section (L569 to 572), β and γ are correctly defined by the differential of the ocean-land carbon inventory with respect to atmospheric carbon and air temperature, all evaluated at the same initial time $t=0$. The crucial assumption is that the same reference time $t=0$ needs to be used. Thus, if we are evaluating β and γ for the present day period, the β and γ are all evaluated for that same pre-industrial time. The authors need to be much more precise and consistent then in defining their analysis in terms of their chosen time period (see below detailed points).

For an analysis of β and γ during any time period, I think the authors need to always define the reference time $t=0$. I think it is fine to compare β and γ evaluated for different time periods (eg. Year 1000 to year 1850 versus year 1850 to 2100), but it is incorrect to report changes in β and γ within those time periods (as that is inconsistent with the original definition for β and γ using the Taylor expansion). I think that main results of the study are drawing upon the former approach, but there is text reporting on changes in carbon-cycle feedback parameters during the same time period.

In summary, I commend the authors for taking on board the critical points raised and in improving the manuscript. I do still have the crucial concern as explicitly set out by Schwinger et al. that the carbon-cycle feedback framework ignores time dependence, which instead the authors attempt to incorporate. I agree that the analysis can be repeated for different time periods, but I disagree that carbon-cycle

feedbacks can be reported within the same time period. While there might be a lot of interest in the reporting of how carbon-cycle feedbacks vary in time, it is crucial that the analysis remains well posed and consistent with the underlying mathematical assumptions used to derive the framework.

Detailed points:

L48-50. This point of different gain for historical analyses and the Earth system model integrations needs further explanation in the main text. Why might there be a different representation? Clearly longer timescales are present in the historical data, which by design are omitted in the Earth system model studies. Unclear why that omission of longer timescales should lead though to a larger warming from carbon-cycle feedback for the Earth system models.

L68-71: You need to define that beta and gamma are defined by the rate of change relative to a fixed reference time.

L103. This connection has been previously shown, see Gregory et al. (2009), J. Climate and Jones and Friedlingstein (2020) ERL.

L110. Equation (1) is a more difficult starting point than equation (4) in the methods.

L115. Given that beta and gamma are evaluated relative to the same reference time, the statement that "two feedbacks vary across different timescales" is either incorrect or needs qualifying. In practice, you are repeating your analysis over different time periods to get different estimates of beta and gamma.

L125 How do you reconcile your result that nonlinear errors are relatively small with the prior published studies (e.g. Gregory et al., 2009; Schwinger et al., 2014), making clear that there are significant errors linked to the method of inference and the nonlinearity.

L135,201,202, Preferable to avoid abbreviations like PILM.

L136. I think it is fine to compare beta and gamma for different periods, but incorrect to report on timescale dependence within a single time period (as that is inconsistent with the original Taylor expansion and definitions).

L222-224. I do not see how these deductions are robust (as that is inconsistent with the original Taylor expansion and definitions).

L376-378. Again I do not see how these deductions are robust for temporal changes in gamma (as that is inconsistent with the original Taylor expansion and definitions).

L407 to 410. I do not see how these deductions are robust (as that is inconsistent with the original Taylor expansion and definitions).

L620. This expansion is correct for the time dependence of the emissions, but explicitly assumes that beta and gamma do not vary in time. In contrast, the authors elsewhere report on time-dependence of beta and gamma in the text.

L886 Paper has appeared.

Response to Reviewer #1:

[Comment A1] Review of “A small climate-amplifying effect of climate-carbon cycle feedback”

As I have described in my review of the previous version of the manuscript, in my opinion the study is interesting and has the potential to influence and revise the allowable carbon emissions before reaching warming targets. The revised version nicely demonstrates how the carbon cycle feedbacks behave on different time scales. The study provides new estimates of the carbon cycle feedbacks based on observations and discuss how comparable or not are these observation-based estimates with those from earth system models, on different time periods and under different CO₂ increase trajectory. The authors have clearly taken my recommendations and comments into consideration and included new analysis along with clarifications, as well as rewritten parts of the manuscript to highlight how meaningful is to compare estimates of beta and gamma from different eras and under different emissions rate. Most of my previous concerns have been addressed in the new manuscript and I appreciate the authors’ effort. However I still have some concerns/reservations about the statement for the amplification effect from the carbon cycle feedbacks based on the gain factor estimated from the observations and the C4MIP models, given that these estimates are based on a different method. Hence, I recommend some minor revisions (or at least some clarifications) as I explain below for the manuscript to be accepted for publication.

[Response A1]: We thank the reviewer again for highlighting the novelty and importance of our work. Based on your comments, we made further revisions and additional analyses, especially the estimates of Fourier analysis-based β and γ^* for the C4MIP models during 1880-2017, and the feedback gain. See our responses to Comments A2 and A3.

Specific comments:

[Comment A2] 1. Estimates of beta and gamma during 1880-2017 and their comparison: In my understanding beta and gamma* from the observations are estimated based on equation 2, and so essentially by the slope and intercept of alpha_k vs TCRE_k in figure 2. However, beta and gamma during 1880-2017 for the C4MIP models (as reported in lines 303 and 322) are estimated using the FEA approach (with their timescale dependence estimated using supplementary equations S5 and S6). Hence, I wonder how much of the difference between observations and C4MIP is actually due to the method used. In my opinion a more direct comparison between the observations-based and*

the model based beta and gamma* will be to estimate them using the same method, such that for the C4MIP you estimate beta and gamma as a slope and an intercept using the alpha_k and TCRE_k from the C4MIP models. Hence, I suggest that for the 1880-2017 time period only, for the C4MIP together with the estimate of beta and gamma* from FEA you report the estimates of beta and gamma based on your approach (equation 2).

[Response A2]: Thanks very much for this constructive suggestion. We conducted additional analysis and compared the two methods. The results are now stated in the main text (see L594-601) and the supplementary file (see supplementary Fig. 11).

Following the reviewer's suggestion, we compared the estimates of β and γ^* over the historical period (1880-2017) for the C4MIP models using our Fourier analysis-based method (equation 2) with those using the FEA approach. Our results for the C4MIP ensemble showed that the estimated value of β using the Fourier analysis-based method was 2.997 ± 0.556 Gt C ppm⁻¹, which was close to the value of 3.064 ± 0.680 Gt C ppm⁻¹ using the FEA approach. The estimated value of γ^* using the Fourier analysis-based method was -30.66 ± 18.72 Gt C K⁻¹, which was about 10% greater than that estimated using the FEA approach (-27.52 ± 11.92 Gt C K⁻¹). Although the estimated values of β and γ^* are quite similar for the C4MIP ensemble using those two different methods, but can be quite different for some individual models (see Figure A1 below). R² for the estimates of β using the two methods is 0.52 across 11 C4MIP models. For γ^* , the slope of a linear regression between the two estimates is 0.62 with quite low R². The estimated values of γ^* using the two methods are reasonably close for 5 of 11 models (UMD, CLIMBER, HadCM3LC, FRCGC, IPSL-CM2C) (Figure A1b). The large differences in the estimates values of β or γ^* for some C4MIP models between the two methods may be associated to bias in CO₂ exchange between the atmosphere and the biosphere due to the poor representation of geophysical and biogeochemical processes in terrestrial ecosystems and oceanic circulations, e.g., no vegetation dynamics and nutrient (nitrogen or phosphorus) cycles coupled with carbon cycle, and their feedbacks to physical climate (e.g., global surface air temperature), and the use of emission as forcings to the C4MIP models. *Schwinger et al. (2014)* also showed that the indirect warming effects from CO₂ increase differed significantly among different C4MIP models forced with CO₂ emission rather than CO₂ concentration, which influenced the simulated ocean carbon uptake in the BGC or COU experiments and contributed to

the differences in the estimated β and γ^* using the two methods.

Figure A1. Comparison of the estimated β and γ^* during 1880-2017 for the C4MIP models using the Fourier analysis-based method with those using the FEA-based approach.

[Comment A3] 2. Estimates of feedback gain, lines 343-368 and associated methods and supplementary. I am somewhat perplexed as why is the airborne fraction used to estimate the gain factor rather directly γ^* , α and β as in equation 9. Then the relative contribution from γ^* and β to the gain is more straightforward. I appreciate that the argument here is that AF is relative constant but in my understanding this is associated/reflected largely by β being relative constant. Nevertheless, my main issue is that, in my understanding, the gain factor based on observations is estimated using the β estimate of 1.52 GtC/GtC based on equation 2. However, the gain factor based on C4MIP is estimated using the β estimate from the FEA approach. In my opinion, a β and γ^* estimate for C4MIP derived from the same method as the observation-based estimate (equation 2) should be used (this relates to my above comment 1) to estimate the gain for C4MIP during the 1880-2017. Further, in my opinion equation 9 should be used directly to estimate the gain rather than the airborne fraction.

[Response A3]: Thanks for this constructive comment. We included the comparison of the estimated g using two different methods in the revised manuscript (see L594-601) and Supplementary Information (see Supplementary Fig. 11).

We agree that the gain factor can be estimated from γ^* , α and β as in equation (9) (i.e., $g = \frac{-\gamma^* \alpha}{(1+\beta)}$). Mathematically, the gain factor estimated using equation (10) (i.e., $g = 1 - \frac{1}{AF(1+\beta)}$) is

identical to that estimated using equation (9), as equation (10) is deduced from equation (7) (i.e., $\frac{1}{TCRE} = (1 + \beta) \frac{1}{\alpha} + \gamma^*$) and equation (9). The observation-based g as estimated using equation (10) is 0.014 ± 0.053 , which falls within the range of the estimated g of ~ 0.017 using equation (9). Difference between these two estimates of g largely results from the uncertainties in air temperature. Calculation of g using equation (9) requires the estimates of γ^* and α , both of which are strongly dependent on global mean annual temperature variations (we used four temperature datasets to account for their uncertainties), therefore estimate of g using equation (9) would vary, depending on which temperature dataset is used. On the other hand, estimate of g using equation (10) is quite stable, because both atmospheric CO₂ concentration and cumulative CO₂ emissions increased nearly exponentially over the period of 1880-2017 (Figure 1a and Figure 2a in manuscript), therefore AF is nearly constant over timescale (i.e., 1/frequency) as is shown in the revised manuscript (see L462-465). Hence, we prefer using the equation (10) (as a new finding in this study) to estimate g . In calculating AF , we firstly estimated AF_k over different timescales from amplitudes of atmospheric CO₂ concentration and cumulative CO₂ emissions using Fourier analysis (Figure 2a in manuscript), then calculated the average and standard deviation of AF from AF_k over timescales (Supplementary Fig. 7). We used the same method as for the observation-based AF (Supplementary Fig. 7) to estimate AF for C4MIP models in this study. As shown in our response to A2, we find that the value of β estimated using the Fourier analysis-based method (2.997 ± 0.556 GtC ppm⁻¹) is close to the value of β estimated using FEA approach (3.064 ± 0.680 GtC ppm⁻¹) for the C4MIP ensemble over 1880-2017. The estimated g using the Fourier analysis-based approach and equation (10) is 0.09 ± 0.05 , which is very close to the estimate of g using the FEA approach (0.09 ± 0.04) for the C4MIP models.

Minor suggestions for clarity/typos:

[Comment A4] 3. Line 80 and 84: Use of “idealized model experiment”. I understand that the authors mean idealised experiments with Earth system models as I am familiar with the referenced studies. However, I think for clarity and to avoid any confusion with use of idealised models rather than Earth system models, I suggest to rephrase to something along the lines of “.. Earth system models under idealised experiments”.

[Response A4]: Agreed and revised. See L90 and L103 in the revised manuscript.

[Comment A5] 4. Lines 172-177. In Supplementary Figure 2, there is a somewhat larger divergence of the predicted and observed temperature after 1980's which you mention at lines 700-701. Is it correct to presume that this divergence after the 1980's is linked with the gamma* not actually being constant during 1880-2017. If yes, maybe the authors could mention in this paragraph (172-177) something along the lines that the larger predicted trend for temperature relative to the observed after 1980's is associated with gamma* treated/assumed as constant in these estimates.

[Response A5]: Agreed and revised. See L229-232.

[Comment A6] 5. Line 213-214 typo: there is an extra that before “ ... the constant value”

[Response A6]: Agreed and revised. See L278.

Response to Reviewer #3:

[Comment B1]: Review of Zhang et al. “A small climate-amplifying effect of climate-carbon cycle feedback” This is my second review of the manuscript.

In my first review, I raised a problem with the central part of the manuscript with its focus on time-dependence of the climate-carbon cycle feedback. The study ignores how the carbon-cycle feedback parameters are defined and the inherent nonlinearity in their framework, which is clearly set out in Schwinger et al. (2014). In this study, the carbon cycle feedback parameters, beta and gamma, are based on a Taylor expansion relative to the pre-industrial state; see equation (2) in Schwinger et al. (2014).

I am very pleased that the authors have dealt with that major concern in explicitly now evaluating the nonlinear feedbacks as part of their study. They find that nonlinear effect is small, which is in contrast to other prior studies. So some further explanation is needed, which reconciles their finding and the prior studies that the nonlinear effect is large (e.g. Gregory et al., 2009; Schwinger et al., 2014). I suspect that each approach is making their assessment by comparing against a different measure.

[Response B1]: Thank very much for your constructive and helpful comments. Firstly, we have proposed a new non-linear carbon-cycle feedback framework by incorporating the nonlinear

feedback term as a function of beta and gamma parameters, i.e., $f(\beta, \gamma)$ into the original linear carbon-cycle feedback framework developed by *Friedlingstein et al. (2006)*. This new framework has been used to estimate observation-based β and γ^* by applying the Fourier analysis approach to historical global mean near-surface temperature (T_A), atmospheric CO₂ concentration (C_A), and CO₂ cumulative emission (C_E) over 1880-2017 (see Figure 2 in revised manuscript). The estimated β , γ^* and other feedback parameters using Fourier analysis approach does not depend on the values of T_A and CO₂ at the reference time, whereas the FEA approach does. Fourier analysis approach based on Fast Fourier Transform extracts amplitudes of T_A , C_A , and C_E at different frequencies (1/timescales) from the three time series over the same periods (e.g., 1880-2017). We also used this new framework (including the nonlinear term) to estimate model-based β and γ^* (i.e., β^{BGC} and $\gamma^{COU-BGC}$) using the FEA approach (*Friedlingstein et al., 2006*) from the coupled/uncoupled (e.g., COU, BGC, RAD) simulations of C4MIP and CMIP5 models. The FEA approach has been widely used for estimating β^{BGC} and $\gamma^{COU-BGC}$ (e.g., *Friedlingstein et al., 2006; Gregory et al., 2009; Schwinger et al. 2014; Arora et al., 2013, 2020*), which is based on a Taylor expansion relative to the pre-industrial state and the reference time (*Schwinger et al. 2014*). The FEA approach cannot be used for estimating β and γ^* only from observations, because FEA approach requires responses of global climate (or mean annual surface temperature) or atmospheric CO₂ to increasing CO₂ emission as obtained in the model simulations (BGC or RAD), which are not available from observations. This study mainly focused on the estimates of observation-based β , γ^* and gain factor using the Fourier analysis approach and comparisons of the estimated β and γ^* and gain factor using the Fourier analysis approach with those using FEA approach using simulations of C4MIP and CMIP5 models.

Secondly, using the new framework including the nonlinear feedback term ($\gamma^* = \gamma + f(\beta, \gamma)\Delta C_A$, where γ is the linear feedback, and γ^* is the sum of linear and nonlinear feedback), we show that the nonlinear feedback ($f(\beta, \gamma)\Delta C_A$) only has a very small impact ($3 \pm 3\%$) on the estimated carbon-concentration feedback parameter (β of land + ocean), and that the contribution of nonlinear feedback to the climate-carbon feedback parameter (γ^* or $\gamma^{COU-BGC}$ of land + ocean) is about $15 \pm 23\%$ for CMIP5 models (see Table B1 or Supplementary Table 4). Note that in this study, we define the nonlinear feedback term as: $f(\beta, \gamma) = [\Delta C_B^{COU} - (\Delta C_B^{BGC} + \Delta C_B^{RAD})] / \Delta C_A^{COU} \Delta T_A^{COU}$ and estimate its contributions to the feedback parameters β and γ^* (in units of GtC

ppm⁻¹ or GtC K⁻¹), whereas previous studies (i.e., *Gregory et al., 2009; Schwinger et al., 2014*) only estimated the contribution of nonlinear feedback to the overall carbon uptakes by land and ocean or by ocean only (in units of Gt C) based on model simulations. For example, *Schwinger et al. (2014)* estimated that the contribution of nonlinear feedback to the estimated ocean carbon uptake under a fully coupled simulation was relatively small (3.6% to 10.6%).

Here, we estimated the contribution of nonlinear feedback, $f(\beta, \gamma)$ using the published results from *Gregory et al. (2009)* or *Schwinger et al. (2014)*. Using the modeling result in *Schwinger et al. (2014)*, we estimated that the non-linear contribution to the ocean γ -feedback γ_O^{nl} is -9.9 GtC K⁻¹, which is about 60% of the total ocean γ parameter, $\gamma_O^{COU-BGC}$ (-16.6 GtC K⁻¹). The non-linear contribution to the ocean β parameter, β_O^{nl} is -0.053 GtC ppm⁻¹, which is only 6.6% of the $\beta_O^{COU-BGC}$ (0.801 GtC ppm⁻¹).

Using the result of the HadCM3LC model under the 1% yr⁻¹ CO₂ increase scenario from Fig.3 in *Gregory et al. (2009)*, we estimated that the non-linear contribution to the land+ocean γ -feedback is -53.84 GtC K⁻¹, which is 45% of the total land+ocean γ -feedback $\gamma_B^{COU-BGC}$ (-119 GtC K⁻¹), and the non-linear contribution to the land+ocean β -feedback is -0.33 GtC ppm⁻¹, which is 20% of the land+ocean β -feedback $\beta_B^{COU-BGC}$ (1.65 GtC ppm⁻¹).

These estimates based on *Gregory et al. (2009)* are noticeably greater than our estimates using the simulations of the nine CMIP5 models (Table B1). Contributions of nonlinear feedback among the CMIP5 models are greatest for the HadGEM2 with the contribution to γ being -28.71 Gt C K⁻¹ (45% of the total γ -feedback) and the contribution to β being -0.185 Gt C ppm⁻¹ (9.6% of the total β -feedback). Across the CMIP5 models, the contribution of non-linear feedback varies from 0.2 to 9.6% for β and from 0.8 to 45% for γ (see Table B1). Also see [Response B8] for more detailed explanation about our calculations. We have included these comparisons in the revised Supplementary Information (see Supplementary Text 3).

Table B1. Estimates of β , γ and $f(\beta, \gamma)$ for the nine CMIP5 models from the 1pctCO₂ climate-carbon cycle feedback experiments.

Model	β^{BGC} (GtC ppm ⁻¹)	$f(\beta, \gamma)\Delta T_A$ (GtC ppm ⁻¹)	$\gamma^{COU-BGC}$ (GtC K ⁻¹)	γ^{RAD} (GtC K ⁻¹)	$f(\beta, \gamma)\Delta C_A$ (GtC K ⁻¹)	$f(\beta, \gamma)$ (GtC ppm ⁻¹ K ⁻¹)
BCC-CSM1	2.06	-0.045	-89.95	-86.22	-9.71	-11.35×10 ⁻³
CanESM2	1.66	0.004	-75.75	-78.82	0.64	0.75×10 ⁻³
CESM1-BGC	0.96	0.022	-17.27	-23.77	5.04	5.89×10 ⁻³

HadGEM2	1.92	-0.185	-63.34	-44.04	-28.71	-33.57×10 ⁻³
IPSL-CM5A-LR	2.05	-0.036	-63.06	-65.13	-6.08	-7.15×10 ⁻³
MIROC-ESM	1.55	-0.041	-104.15	-100.14	-6.3	-7.41×10 ⁻³
MPI-ESM-LR	2.30	-0.111	-102.38	-88.09	-18.98	-22.19×10 ⁻³
NorESM-ME	1.11	-0.042	-21.69	-14.08	-9.76	-11.41×10 ⁻³
UVic ESCM2.9	1.74	-0.058	-93.67	-85.44	-12.59	-14.60×10 ⁻³
Ensemble Mean	1.71	-0.055	-70.14	-65.08	-9.6	-11.22×10 ⁻³
+/-S.D.	+/-0.44	0.062	+/-32.43	+/-30.74	+/-10.03	+/-11.72×10 ⁻³

[Comment B2] I do have a further major concern that links back to the original Taylor expansion used to define the carbon-cycle feedback parameters. As repeated in their Methods section (L569 to 572), beta and gamma are correctly defined by the differential of the ocean land carbon inventory with respect to atmospheric carbon and air temperature, all evaluated at the same initial time $t=0$. The crucial assumption is that the same reference time $t=0$ needs to be used. Thus, if we are evaluating beta and gamma for the present day period, the beta and gamma are all evaluated for that same pre-industrial time. The authors need to be much more precise and consistent then in defining their analysis in terms of their chosen time period (see below detailed points).

For an analysis of beta and gamma during any time period, I think the authors need to always define the reference time $t=0$. I think it is fine to compare beta and gamma evaluated for different time periods (eg. Year 1000 to year 1850 versus year 1850 to 2100), but it is incorrect to report changes in beta and gamma within those time periods (as that is inconsistent with the original definition for beta and gamma using the Taylor expansion). I think that main results of the study are drawing upon the former approach, but there is text reporting on changes in carbon-cycle feedback parameters during the same time period.

In summary, I commend the authors for taking on board the critical points raised and in improving the manuscript. I do still have the crucial concern as explicitly set out by Schwinger et al. that the carbon-cycle feedback framework ignores time dependence, which instead the authors attempt to incorporate. I agree that the analysis can be repeated for different time periods, but I disagree that carbon-cycle feedbacks can be reported within the same time period. While there might be a lot of interest in the reporting of how carbon-cycle feedbacks vary in time, it is crucial that the analysis remains well posed and consistent with the underlying mathematical assumptions used to derive the framework.

[Response B2]: We agreed that the estimated values of β and γ critically depends on the reference

time if the FEA approach (Taylor expansion) is used. However, this is not a significant problem if the Fourier approach is used, as we used the amplitudes of three time series ($C_A(t)$, $C_E(t)$ and $T_A(t)$) and a linear regression (see Figures 1 and 2, and Supplementary Fig.1) to estimate those two parameters. Both Fourier series (from Fourier transform) and Taylor series (from Taylor expansion) are decompositions of a function e.g., $f(x)$, which is represented as a linear combination of a set of functions. But the Fourier series consists of orthonormal base functions $\{1, \sin(\omega x), \cos(\omega x), \sin(2\omega x), \cos(2\omega x)\dots\}$, implying that the coefficients (amplitudes) depends on a global property of the function (and does not rely on the reference time). While Taylor series does not use an orthonormal basis, e.g., $\{1, x, x^2, x^3\dots\}$, in that the coefficient depends only in local properties of the function, i.e., the variable state at the reference time. For the model-based β and γ estimates using the FEA approach, we have set the same reference time to 1880 for historical period (1880-2017) and the future scenario (1880-2100), and updated model-based estimates of β and γ for these two periods in the revised manuscript (L408-409 and L426-427, see also Table B2). Furthermore, the estimates for 2018-2100 are not presented in the revised manuscript, because the reference time would be different from the historical or preindustrial periods, then the comparisons of the estimated β and γ across different periods would be problematic if the FEA approach is used, as pointed out by the reviewer.

Table B2. Estimates of β and γ for eleven C4MIP models for the periods of 1880-2017 and 1880-2100 and 2018-2100 using the FEA approach.

Model	β^{BGC} (GtC ppm ⁻¹)			$\gamma^{COU-BGC}$ (GtC K ⁻¹)		
	1880-2017	1880-2100	2018-2100	1880-2017	1880-2100	2018-2100
BERN-CC	3.52	2.94	2.78	-38.97	-61.23	-70.80
CCSM-1	2.46	1.97	1.87	-12.97	-21.77	-24.05
CLIMBER	2.47	1.99	1.87	-17.99	-41.21	-50.23
FRCGC	3.18	2.36	2.16	-36.29	-72.19	-91.24
HadCM3LC	2.95	2.14	1.94	-41.19	-113.29	-132.63
IPSL-CM2C	3.49	3.25	3.19	-13.22	-48.79	-72.53
IPSL-CM4-LOOP	3.17	2.34	2.11	-10.20	-19.05	-22.21
LLNL	4.27	3.70	3.55	-25.89	-32.71	-37.04
MPI	3.27	2.50	2.30	-36.83	-44.11	-46.74

UMD	1.67	1.71	1.73	-30.64	-52.22	-58.49
UVic-2.7	3.26	2.29	2.05	-38.54	-67.40	-79.01
Ensemble Mean	3.07	2.47	2.32	-27.52	-52.18	-62.27
+/-S.D.	+/-0.68	+/-0.60	+/-0.59	+/-11.93	+/-26.54	+/-32.17

Detailed points:

[Comment B3] L48-50. *This point of different gain for historical analyses and the Earth system model integrations needs further explanation in the main text. Why might there be a different representation? Clearly longer timescales are present in the historical data, which by design are omitted in the Earth system model studies. Unclear why that omission of longer timescales should lead though to a larger warming from carbon-cycle feedback for the Earth system models.*

[Response B3]: Thank you very much for this important comment. We agree that historical data have included information of all time scales (from annual-to-centennial and longer timescales) affecting the feedback gain, while the Earth system models (from the C4MIP used in this study) by design have omitted the longer timescales. This study finds that the gain factor of carbon-climate feedback as estimated based on model simulation is much greater than value of the observation-based gain factor. Reasons for this difference may include:

(1) the poor representations/descriptions of soil carbon pools (no vertical resolution, as one-layer is used in most ESMs for modeling soil carbon), vegetation dynamics, and soil respiration/microbial processes in C4MIP models. For example, most models applied the same reference respiration rate (with the parameter temperature sensitivity of soil respiration Q_{10}) to soil carbon pools in all regions and organic carbon at different soil depths, which may overestimate the sensitivity of soil carbon to warming, therefore land climate-carbon feedback. Studies found that Q_{10} for soil carbon varied significantly across different regions and decreased with soil depth (*Ren et al., 2020; Meyer et al., 2018; Zhou et al., 2009* for example);

(2) the omission of land use change in C4MIP models may lead to biases in the simulated land surface air temperature over the historical period (both biophysical and biogeochemical effect of land use change on surface temperature) (see *Pongratz et al., 2010*).

(3) *Willeit et al. (2014)* found that carbon cycle feedback was sensitive to initial climate state and initial values of carbon pools. The C4MIP model simulations used the equilibrium climate state and

carbon pools obtained from spinup as the initial conditions for the climate in 1860s. However, in real world, the climate and all carbon pools in 1860 are unlikely to be at steady state. Therefore, model simulations would have biases in the simulated carbon pools, water storages, and energy balance in land and oceans over the historical period, therefore errors in the estimated carbon-climate feedback parameters as estimated based on those model simulations. We have added these points in the revised manuscript (see L587-594).

[Comment B4] L68-71: You need to define that beta and gamma are defined by the rate of change relative to a fixed reference time.

[Response B4]: Agreed. We have rephrased this sentence to “The β (Gt C ppm⁻¹) and γ (Gt C K⁻¹) are also defined as the rates of change in land and ocean carbon storages relative to a fixed reference time to atmospheric CO₂ concentration increase and to global climate change that is often quantified by the global mean surface temperature change, respectively.” in the revised version (see L79-80). We also included the reference time for the estimated values of β and γ when FEA approach was used (e.g., see L871-873).

[Comment B5] L103. This connection has been previously shown, see Gregory et al. (2009), J. Climate and Jones and Friedlingstein (2020) ERL.

[Response B5]: Thanks for this comment. This connection between β , γ , α and $TCRE$ was not clearly shown in Gregory et al. (2009), J. Climate, but is shown in Jones and Friedlingstein (2020) ERL. We have added this citation (Jones and Friedlingstein, 2020, ERL) in the revised manuscript.

[Comment B6] L110. Equation (1) is a more difficult starting point than equation (4) in the methods.

[Response B6]: Thank you for your suggestion. To some extent, equation (4) was stated implicitly in words in the first paragraph. We felt that it may be too repetitive to start with equation (4) here, so no change is made here.

Equation (1) is a new climate-carbon cycle feedback framework including a non-linear feedback term, which is developed based on the well-known linear climate-carbon cycle feedback framework developed by Friedlingstein et al., 2006 (i.e., $\Delta C_E = \Delta C_A + \beta \Delta C_A + \gamma \Delta T_A$). We have included details in the Methods to help readers understand equation (1) (see L734-766).

[Comment B7] L115. Given that beta and gamma are evaluated relative to the same reference time, the statement that “two feedbacks vary across different timescales” is either incorrect or needs qualifying. In practice, you are repeating your analysis over different time periods to get different estimates of beta and gamma.

[Response B7]: Thanks for this comment. We did not repeat our analysis over different time periods to estimate the timescale-dependence of β and γ , but decomposed three observational time series (C_E , C_A and T_A) using the Fourier transform to extract information about the variations of C_E , C_A and T_A at different frequencies (or 1/timescale), then estimates β and γ at different timescales. This Fourier approach was used to estimate the timescale-dependence of β and γ for three different time periods (e.g., 1880-2017, 1880-2100, and 1000-1850). We modified the statement to “...the two feedback parameters vary over different periods of time, or across different timescales (1/frequencies) over the same time period.” (see L147-148).

[Comment B8] L125 How do you reconcile your result that nonlinear errors are relatively small with the prior published studies (e.g. Gregory et al., 2009; Schwinger et al., 2014), making clear that there are significant errors linked to the method of inference and the nonlinearity.

[Response B8]: Our calculation using CMIP5 simulations showed that the contribution of nonlinear feedback to the global (land+ocean) β (carbon-concentration feedback) is negligible ($\sim 3 \pm 3\%$), and the contribution to the global (land+ocean) γ (carbon-climate feedback) is significant ($\sim 15 \pm 23\%$) (see Table B1). However, the estimated contribution of nonlinear feedback varied greatly among the different CMIP5 models (0.2 to 9.6% for β and 0.8 to 45% for γ , see Supplementary Table 4). This suggests that uncertainties are quite large in the estimated nonlinear carbon-climate feedback based the simulations by Earth system models. We have included these comparisons in the revised manuscript (L486-496) and the revised supplementary information (see Supplementary Text 3). Key information is also reproduced below.

Gregory et al. (2009) found that the fully coupled (COU) and uncoupled (biogeochemically coupled (BGC)) simulations of C4MIP models (see Friedlingstein et al. 2006) only were inadequate for quantifying the nonlinear contribution to the modelled carbon cycle-climate feedback, they then carried out three experiments (i.e., COU, BGC, and radiatively coupled (RAD)) using the

HadCM3LC model forced with an CO₂ increase at 1% yr⁻¹ for 140 years. Based on these three simulations, *Gregory et al.* found that: (1) carbon uptake by land and ocean (ΔC_B) in COU experiment was less than the sum of the carbon uptake by land and ocean from the BGC and RAD experiments ($\Delta C_B^{BGC} + \Delta C_B^{RAD}$), and that difference increased with time; (2) the COU carbon uptake (ΔC_B^{COU}) was about two-thirds that of $\Delta C_B^{BGC} + \Delta C_B^{RAD}$ by the year 140 (see Fig.3 of *Gregory et al. (2009)*). But *Gregory et al. (2009)* did not further estimate the nonlinear contribution to the estimated β and γ , or the $f(\beta, \gamma)$ term as defined in our study. Using the results as shown Fig.3 of *Gregory et al. (2009)*, we estimated that $f(\beta, \gamma) = \Delta C_B^{nl} / (\Delta C_A^{COU} \Delta T_A^{COU}) = -280 \text{ Gt C} / (850 \text{ ppm} * 5.2 \text{ K}) = -0.063 \text{ Gt C ppm}^{-1} \text{ K}^{-1}$, or the contribution of nonlinear term to the land+ocean γ -feedback is $-53.84 \text{ Gt C K}^{-1}$ which is 45% of the total land+ocean γ -feedback $\gamma_B^{COU-BGC}$ (-119 Gt C K^{-1}), and the contribution of nonlinear term to the land+ocean β -feedback is $-0.33 \text{ Gt C ppm}^{-1}$ which is 20% of the land+ocean β -feedback $\beta_B^{COU-BGC}$ ($1.65 \text{ Gt C ppm}^{-1}$).

Schwinger et al. (2014) quantified the nonlinear effects on ocean carbon-climate feedback using the COU, BGC and RAD simulations of 7 CMIP5 models, and showed that ocean carbon uptakes (ΔC_O) from the BGC and RAD simulations did not add up linearly to the ocean carbon uptake as simulated in the COU simulation, ie $\Delta C_O^{COU} \neq \Delta C_O^{BGC} + \Delta C_O^{RAD}$ because of the nonlinear effect. By the end of year 140, the nonlinear contribution to ocean carbon uptake, $\Delta C_O^{nl} (= \Delta C_O^{COU} - (\Delta C_O^{BGC} + \Delta C_O^{RAD}))$ ranges from -19 to -58 Gt C in the CMIP5 models, which is 3.6% to 10.6% of the ΔC_O^{COU} . Note that their calculations were for ocean carbon uptake only. *Schwinger et al. (2014)* also estimated γ feedback parameter by using the differences of COU-BGC ($\gamma_O^{COU-BGC}$) or RAD (γ_O^{RAD}), and found that the average value of $\gamma_O^{COU-BGC}$ varied from -11.2 to -21.9 GtC K^{-1} with an average value of -16.6 GtC K^{-1} for 7 CMIP5 models, while γ_O^{RAD} varied from -1.9 to -10.3 GtC K^{-1} with an average value of -6.7 GtC K^{-1} (see the Table 2 in *Schwinger et al.'s*). *Schwinger et al. (2014)* did not explicitly define the contribution of the nonlinear γ term to the estimated ocean γ feedback (that is the $f(\beta, \gamma) \Delta C_A^{COU}$ term defined in our study), but we can diagnose that the difference between $\gamma_O^{COU-BGC}$ and γ_O^{RAD} was largely contributed by the nonlinear feedback, eg $\gamma_O^{nl} = \gamma_O^{COU-BGC} - \gamma_O^{RAD} = -9.9 \text{ GtC K}^{-1}$ on average of 7 CMIP5 models, which is about 59.6% of the $\gamma_O^{COU-BGC}$. *Schwinger et al. (2014)* also showed the difference between two methods (using COU-BGC or RAD) for estimating ocean β ($\beta_O^{COU-RAD} - \beta_O^{COU-BGC}$) was $0.748 - 0.801 = -0.053 \text{ Gt C ppm}^{-1}$, which was only 6.6% of the $\beta_O^{COU-BGC}$ (see the Table 2 in *Schwinger et al.'s*).

In this study, we firstly defined the nonlinear feedback term $f(\beta, \gamma) = [\Delta C_B^{COU} - (\Delta C_B^{BGC} + \Delta C_B^{RAD})] / \Delta C_A^{COU} \Delta T_A^{COU}$, and quantified its contributions to the β feedback ($f(\beta, \gamma) \Delta T_A^{COU}$) and to the γ feedback ($f(\beta, \gamma) \Delta C_A^{COU}$). We estimated that the nonlinear feedback term, $f(\beta, \gamma)$ for both land and ocean is $-11.22 \pm 11.72 \times 10^{-3}$ Gt C ppm⁻¹ K⁻¹, and its contribution to the land+ocean γ -feedback is -9.6 ± 10.03 Gt C K⁻¹ on average, ranging from -28.71 to 5.04 Gt C K⁻¹ among the nine CMIP5 models, which means that the γ_B^{RAD} (-65.08 ± 30.74 Gt C K⁻¹) was about $15 \pm 23\%$ smaller in magnitude than the $\gamma_B^{COU-BGC}$ (-70.14 ± 32.43 Gt C K⁻¹), and the contribution of the nonlinear term to the land+ocean β -feedback was -0.055 ± 0.062 Gt C ppm⁻¹, or only $3 \pm 3\%$ of the $\beta_B^{COU-BGC}$ (1.71 ± 0.44 Gt C ppm⁻¹) (see Table B1).

[Comment B9] L135,201,202, *Preferable to avoid abbreviations like PILM.*

[Response B9]: Agreed. All “the PILM” have been replaced by “the 1000-1850 period”.

[Comment B10] L136. *I think it is fine to compare beta and gamma for different periods, but incorrect to report on timescale dependence within a single time period (as that is inconsistent with the original Taylor expansion and definitions).*

[Response B10]: Thanks for this detailed comment. As stated in our response to Comment B2, we applied the Fourier analysis (Fast Fourier Transform) approach to estimate timescale dependence of observation-based β and γ for three time periods. We did not use Taylor expansion to estimate observation-based β and γ . Taylor expansion was applied to estimate model-based β and γ from C4MIP and CMIP5 only for different periods. In the revised manuscript, we have used $\beta_k, \gamma_k, \eta_k$ or γ_{100yr} to represent the estimates at different timescales (or frequencies) using Fourier analysis-based approach (see L304-313 for example).

[Comment B11] L222-224. *I do not see how these deductions are robust (as that is inconsistent with the original Taylor expansion and definitions).*

[Response B11]: See our responses to Comments B2 and B10.

[Comment B12] L376-378. *Again I do not see how these deductions are robust for temporal changes*

in gamma (as that is inconsistent with the original Taylor expansion and definitions).

[Response B12]: See our responses to Comments B2 and B10.

[Comment B13] L407 to 410. I do not see how these deductions are robust (as that is inconsistent with the original Taylor expansion and definitions).

[Response B13]: See our responses to Comments B2 and B10.

[Comment B14] L620. This expansion is correct for the time dependence of the emissions, but explicitly assumes that beta and gamma do not vary in time. In contrast, the authors elsewhere report on time-dependence of beta and gamma in the text.

[Response B14]: We only reported the time-scale dependence of β and γ for three different periods, not the time-dependence of β and γ . Timescale is the inverse of frequency, and variations at different frequencies were extracted using the Fourier transform.

[Comment B15] L886 Paper has appeared.

[Response B15]: Thanks. The citation information of Arora et al. (2020) has been updated in the revised manuscript.

References:

1. Arora, V. K., et al. Carbon–concentration and carbon–climate feedbacks in CMIP6 models and their comparison to CMIP5 models. *Biogeosciences* 17, 4173-4222 (2020).
2. Arora, V.K. et al. Carbon–Concentration and Carbon–Climate Feedbacks in CMIP5 Earth System Models. *J. Clim.*, 26(15): 5289-5314 (2013).
3. Friedlingstein, P., et al. Climate–Carbon Cycle Feedback Analysis: Results from the C4MIP Model Intercomparison. *J. Clim.* 19, 3337-3353 (2006).
4. Gregory, J. M., Jones, C. D., Cadule, P. & Friedlingstein, P. Quantifying Carbon Cycle Feedbacks. *J. Clim.* 22, 5232-5250 (2009).
5. Jones, C. D. & Friedlingstein, P. Quantifying process-level uncertainty contributions to TCRE and carbon budgets for meeting Paris Agreement climate targets. *Environmental Research Letters* 15, (2020).

6. Meyer N, Welp G, Amelung W. The Temperature Sensitivity (Q10) of Soil Respiration: Controlling Factors and Spatial Prediction at Regional Scale Based on Environmental Soil Classes. *GLOBAL BIOGEOCHEMICAL CYCLES*, 32, 306-323 (2018).
7. Pongratz J, Reick CH, Raddatz T, Claussen M. Biogeophysical versus biogeochemical climate response to historical anthropogenic land cover change. *Geophysical Research Letters*, 37 (2010).
8. Ren S, Ding J, Yan Z et al. Higher Temperature Sensitivity of Soil C Release to Atmosphere From Northern Permafrost Soils as Indicated by a Meta - Analysis. *GLOBAL BIOGEOCHEMICAL CYCLES*, 34, (2020).
9. Schwinger, J., et al. Nonlinearity of Ocean Carbon Cycle Feedbacks in CMIP5 Earth System Models. *J. Clim.* 27, 3869-3888 (2014).
10. Willeit M, Ganopolski A, Dalmonech D, Foley AM, Feulner G. Time-scale and state dependence of the carbon-cycle feedback to climate. *Climate Dynamics*, 42, 1699-1713 (2014).
11. Zhou T, Shi P, Hui D, Luo Y. Global pattern of temperature sensitivity of soil heterotrophic respiration (Q10) and its implications for carbon-climate feedback. *Journal of Geophysical Research*, 114 (2009).

REVIEWERS' COMMENTS

Reviewer #1 (Remarks to the Author):

The authors have clearly addressed all my previous concerns/comments in the new version of the manuscript. This study has implications for how the carbon cycle feedbacks operate on different time-scales and the potential to revise allowable carbon emissions before reaching warming targets based on observations. Hence, it is of interest to the wider scientific community and fits well the scope of nature communications. I recommend that this revised version of the manuscript is accepted for publication as is.

Reviewer #3 (Remarks to the Author):

3rd review

The manuscript has significantly improved over the past 2 rounds. My primary concerns had been the omission of the non-linear terms and addressing the time dependence when the usual Taylor approach assumes a constant reference time. Both these concerns have been fully dealt with and I thank the authors for their work here.

I have one remaining scientific concern and one editorial concern.

The scientific concern is that the main result is a reduced climate amplification based on observational data compared with model data. However, it is unclear as to why there is less amplification. Usually the observations contain more variability than the models and the observations contain longer timescale feedbacks involving deglaciation, so the outcome that the models over exaggerate the climate amplification is surprising. Does this involve competing and partly compensating contributions in the observational record and possibly linked to land use changes? Unlikely to be able to unravel the reasons for this difference, but including some speculations would be useful.

My editorial concern is that the text is very hard to follow due to the detailed writing and the widespread use of abbreviations. You might know what you refer to, but the text is very difficult to follow for a non-expert reader due to the number of abbreviations and changing choices into which parameters are used. The text usually only refers to either the mathematical symbol for the variable or an abbreviation, so makes the material very challenging to follow.

In summary, the manuscript provides a new analysis of global carbon uptake and feedback over different time periods, and there is a new and distinct outcome of a relative small climate-amplifying factor. Thus, there is an important outcome and message. However, the manuscript would benefit from being made more accessible.

Detailed points:

L103 and 104. Jones and Friedlingstein (2020) have previously shown how a single equation connects TCRE, beta, gamma and alpha. So this derivation should not be claimed as new here. In fact, Gregory et al. (2009) also previously has shown a similar relationship for the TCRE and beta and gamma. The authors claim that bot to be the case, but see P5247 in Gregory et al. (2009).

L121 to 126. Most studies focus on the TCRE rather than the reciprocal of the TCRE.

L125 You start by using $1/\alpha$ here, but then later switch to η in L199. Would be better to reduce the number of variables to make the text more accessible.

L132 Better to expand more fully the links in the TCRE (and alpha) to the ratios of the variability in C_A (or C_E) to the variability in T_A . This concise construction is less clear and actually misleading as TCRE is defined by temperature change T_A divided by cumulative carbon emissions (C_E).

L138 and later. Define FEA, and try to avoid abbreviations if not needed.

L138-140. This is an important point that is now mentioned.

L144 This construction of "15 or 1563 combinations" is difficult to understand. Is that 15 combinations

of CO₂ and 1563 combinations of air temperature?

L206. Usually the radiative forcing from CO₂ is related to a logarithmic dependence on the change in atmospheric CO₂. You can then assess whether the discrepancy in η between the pre-industrial and industrial forcing is due to the dependence of radiative forcing on atmospheric CO₂ or instead due to the lack of a steady state.

L236 to 238. You report that γ increases with timescale. Here or later can you speculate as to why that might be the case?

L242. EnOBS is a little cryptic.

L240 to 252. The details of the outcomes of the calculations are reported, but without providing much insight or context here. The text is becoming difficult to follow.

L256 & 261. Avoid the use of abbreviations when possible, MCA and LIA, to make the text easier to read.

L254 to 265. This analysis might well be potentially interesting, but the details are obscuring the wider implications.

L299 & L355 Explain what FEA refers to

L395 This is a useful extra step.

L408 Need to make clear that this relationship has been previously stated by Jones and Friedlingstein (2020), and Gregory et al. (2009) has a similar relationship.

L417 This outcome is important of reduced climate amplification based on observational data compared with model data. However, it is unclear as to why there is less amplification. Usually the observations contain more variability than the models and the observations contain longer timescale feedbacks involving deglaciation, so the outcome that the models over exaggerate the climate amplification is surprising. Does this involve competing and partly compensating contributions in the observational record? Is the inclusion of land use change in the observational record a crucial difference to the Earth system model diagnostics? Unlikely to be able to unravel the reasons for this difference, but including some speculations would be useful.

L429 LinExp, explain what is meant.

L445 γ^* and γ are used here, if the distinction is important, then good to be clear as to what is meant.

L686 Is the correct equation referenced? (19) has not yet been reached.

L714 FEA not defined

L1060 Define what the global-mean temperature is, i.e surface air temperature.

Responses to the comments from the two reviewers.

Comments from the two reviewers are in black and our responses in blue.

To Reviewer #1

[Comment]: *The authors have clearly addressed all my previous concerns comments in the new version of the manuscript. This study has implications for how the carbon cycle feedbacks operate on different time-scales and the potential to revise allowable carbon emissions before reaching warming targets based on observations. Hence, it is of interest to the wider scientific community and fits well the scope of nature communications. I recommend that this revised version of the manuscript is accepted for publication as is.*

[Response]: We thank the reviewer again for highlighting the novelty and importance of our work and for the recommendation for publication.

To Reviewer #3

3rd review

[Comment B1]: *The manuscript has significantly improved over the past 2 rounds. My primary concerns had been the omission of the non-linear terms and addressing the time dependence when the usual Taylor approach assumes a constant reference time. Both these concerns have been fully dealt with and I thank the authors for their work here. I have one remaining scientific concern and one editorial concern.*

[Response B1]: We thank the reviewer again for providing constructive comments that has helped considerably improve our manuscript.

[Comment B2]: *The scientific concern is that the main result is a reduced climate amplification based on observational data compared with model data. However, it is unclear as to why there is less amplification. Usually the observations contain more variability than the models and the observations contain longer timescale feedbacks involving deglaciation, so the outcome that the models over exaggerate the climate amplification is surprising. Does this involve competing and partly compensating contributions in the observational record and possibly linked to land use changes? Unlikely to be able to unravel the reasons for this difference, but including some speculations would be useful.*

[Response B2]: Thanks very much for this important comment. We also realized that it is difficult to address why the observation-based amplification effect is smaller than the model-based. We have discussed some potential reasons for this difference (L467-472 in revised version). On one hand, the real terrestrial and oceanic ecosystems and climate system are more complex than the state-of-the-art models. Many processes of ecosystem processes such as non-linear soil microbial respiration, ecosystem resilience and stability (Huang & Xia, 2019), ecosystem acclimation (Wang *et al.*, 2020), vegetation phenology change and nutrient limitation effect have not been included or only partially included by current Earth system models. For instance, land surface modules in most of these models are developed on canopy-level “big-leaf”-based carbon cycle conceptual models, but ignores the competition mechanism within individuals of an ecosystem. On the other hand, land use changes from deforestation and afforestation over different historical periods were not included in C⁴MIP

experiments. In reality, land use change could alter both biochemical (e.g., carbon cycle) and biogeophysical (e.g., surface albedo, evapotranspiration, radiative transfer, and surface temperature etc.). This could be a reason for C⁴MIP model overestimation. We suggest future C⁴MIP or feedback sensitivity modeling experiments should consider the impact of land use change.

[Comment B3]: *My editorial concern is that the text is very hard to follow due to the detailed writing and the widespread use of abbreviations. You might know what you refer to, but the text is very difficult to follow for a non-expert reader due to the number of abbreviations and changing choices into which parameters are used. The text usually only refers to either the mathematical symbol for the variable or an abbreviation, so makes the material very challenging to follow.*

[Response B3]: Thanks very much for this comment and the following detailed points. We have carefully revised the manuscript according to your detailed comments.

[Comment B4]: *In summary, the manuscript provides a new analysis of global carbon uptake and feedback over different time periods, and there is a new and distinct outcome of a relative small climate-amplifying factor. Thus, there is an important outcome and message. However, the manuscript would benefit from being made more accessible.*

[Response B4]: We thank the reviewer again for highlighting the novelty and importance of the finding of a relative small climate-amplifying effect based on observational analysis and compared to modeled values. According to your comments, we further modified the manuscript to enhance its readability.

Detailed points

[Comment B5]: *L103 and 104. Jones and Friedlingstein (2020) have previously shown how a single equation connects TCRE, beta, gamma and alpha. So this derivation should not be claimed as new here. In fact, Gregory et al. (2009) also previously has shown a similar relationship for the TCRE and beta and gamma. The authors claim that bot to be the case, but see P5247 in Gregory et al. (2009).*

[Response B5]: Agreed. We have removed the statement and added citation of both *Gregory et al. (2009)* and *Jones and Friedlingstein (2020)*. Please see L99-101.

[Comment B6]: *L121 to126. Most studies focus on the TCRE rather than the reciprocal of the TCRE.*

[Response B6]: Thanks for this comment. As in this study, the linear relationship to estimate $(1 + \beta_k)$ can only be reflected by the reciprocal of the TCRE and the reciprocal of the α as shown in Equation (2) and Fig. 2 in manuscript, therefore we keep the reciprocal of the TCRE.

[Comment B7]: *L125 You start by using $1/\alpha$ here, but then later switch to η in L199. Would be better to reduce the number of variables to make the text more accessible.*

[Response B7]: Thanks for this comment. We took the use of η based on two reasons: (1) the η defined as $\Delta C_A / \Delta T_A$ in a unit of ppm K⁻¹, has been frequently used as a metric for centennial to millennial scales that represents the strength of carbon cycle-climate feedback without the forcing from anthropogenic CO₂ emissions (Cox &

Jones, 2008, Frank *et al.*, 2010, Scheffer *et al.*, 2006, Willeit *et al.*, 2014). (2) the α defined as $\Delta T_A / \Delta C_A$ in K GtC^{-1} , is usually used for the anthropogenic CO_2 emissions-driven time periods or scenarios (Friedlingstein *et al.*, 2006, Gregory *et al.*, 2009) which describes the response of carbon cycle-climate feedback to anthropogenic CO_2 emissions. The direct use of the α for the 1000-1850 period could be physically meaningless. Therefore, we use α and η respectively for the 1850-2017 and the 1000-1850 periods.

[Comment B8]: *L132 Better to expand more fully the links in the TCRE (and alpha) to the ratios of the variability in C_A (or C_E) to the variability in T_A . This concise construction is less clear and actually misleading as TCRE is defined by temperature change T_A divided by cumulative carbon emissions (C_E).*

[Response B8]: Agreed and changed. Please see L127-129 in the revised manuscript.

[Comment B9]: *L138 and later. Define FEA, and try to avoid abbreviations if not needed.*

[Response B9]: The FEA has been defined in the revised version (L133-134).

[Comment B10]: *L138-140. This is an important point that is now mentioned.*

[Response B10]: Thanks.

[Comment B11]: *L144 This construction of “15 or 1563 combinations” is difficult to understand. Is that 15 combinations of CO_2 and 1563 combinations of air temperature*

[Response B12]: Thanks for this detailed comment. We have rephrased this point into “large ensembles based on combinations of ice-core atmospheric CO_2 records (C_A) and reconstructed surface air temperature (T_A) datasets (Methods)” (L141-143).

[Comment B13]: L206. Usually the radiative forcing from CO₂ is related to a logarithmic dependence on the change in atmospheric CO₂. You can then assess whether the discrepancy in η between the pre-industrial and industrial forcing is due to the dependence of radiative forcing on atmospheric CO₂ or instead due to the lack of a steady state.

[Response B13]: We agree with this comment. We have calculated industrial temperature change in response to the historical growth of atmospheric CO₂ using the box-model as provided by equations 29-30 in the manuscript. Detailed results are provided in Supplementary Fig. 2. Result shows that box-model simulated temperature has a similar increase rate to the observed temperature, indicating that the discrepancy in η between the pre-industrial and industrial forcings is partly due to the dependence of radiative forcing on atmospheric CO₂. We speculated that it may be largely due to the temperature change in response to the increase in atmospheric CO₂ during the industrial era has not reached steady state, as large part of atmospheric CO₂ increase during industrial period was driven by emissions not due to warming-induced CO₂ release from land and ocean reservoirs (L205-207).

[Comment B14]: L236 to 238. You report that γ increases with timescale. Here or later can you speculate as to why that might be the case

[Response B14]: We inferred that the γ increases with timescales over the 1000-1850 period might be caused by the dependence of climate-carbon feedback on climate state, e.g., γ on a near equilibrium climate state (average on longer timescale during the pre-industrial period) would be larger than γ on a transient climate

state (L435-441). This speculation may be supported by the fact that values of equilibrium climate sensitivity derived from Earth system models are about 1.5~2 times larger than the transient climate sensitivity (Dai *et al.*, 2020, He *et al.*, 2017)

[Comment B15]: L242. *EnOBS is a little cryptic.*

[Response B15]: The EnOBS is defined as a dataset of ensemble estimates of >1500 combinations of 521 reconstructed temperature records from 1000 to 1850 with 3 ice-core CO₂ records (L235-237).

[Comment B16]: L240 to 252. *The details of the outcomes of the calculations are reported, but without providing much insight or context here. The text is becoming difficult to follow.*

[Response B16]: Thanks for this comment. Our result here may support the finding of Frank et al. (Frank *et al.*, 2010). We have included an inference in the revised manuscript (L247-251). *“These results suggest that the timescale or temporal dependence of η over the 1000-1850 (Frank et al., 2010) is largely driven by the positive feedback of terrestrial and oceanic carbon pools to climate (i.e., the γ feedback), implying that on longer timescales, warming of the climate would cause more release of CO₂ into the atmosphere and in return, amplify warming.”*

[Comment B17]: L256 & 261. *Avoid the use of abbreviations when possible, MCA and LIA, to make the text easier to read.*

[Response B17]: Agreed and changed. Please see L251-265 through the revised manuscript.

[Comment B18]: *L254 to 265. This analysis might well be potentially interesting, but the details are obscuring the wider implications.*

[Response B18]: We agree with this comment. Comparison between the estimates of carbon-climate feedback parameter (γ) over the warmer period of 1000-1300 and the cooler period of 1400-1700 may improve understanding of climate-carbon feedback system on longer time scales and help constrain future projections (Tierney *et al.*, 2020). In future studies, more analysis is required for the 1000-1850 period and longer time periods.

[Comment B19]: *L299 & L355 Explain what FEA refers to*

[Response B19]: The FEA has been defined in the revised version (L133-134).

[Comment B20]: *L395 This is a useful extra step.*

[Response B20]: Thanks.

[Comment B21]: *L408 Need to make clear that this relationship has been previously stated by Jones and Friedlingstein (2020), and Gregory *et al.* (2009) has a similar relationship.*

[Response B21]: Agreed and these two citations have been added. Please see L400-401.

[Comment B22]: *L417 This outcome is important of reduced climate amplification based on observational data compared with model data. However, it is unclear as to why there is less amplification. Usually the observations contain more variability than the models and the observations contain longer timescale feedbacks involving deglaciation, so the outcome that the models over exaggerate the climate amplification is surprising. Does this involve competing and partly compensating contributions in the*

observational record? Is the inclusion of land use change in the observational record a crucial difference to the Earth system model diagnostics? Unlikely to be able to unravel the reasons for this difference, but including some speculations would be useful.

[Response B22]: Please see our response to **Comment 2**.

[Comment B23]: *L429 LinExp, explain what is meant.*

[Response B23]: Agreed. We have revised the “LinExp theory” to “the linear system in response to exponential increase of forcing (LinExp) theory” in the revised version.

Please see L422-423.

[Comment B24]: *L445 gamma star and gamma are used here, if the distinction is important, then good to be clear as to what is meant.*

[Response B24]: The $\gamma^* = \gamma + f(\beta, \gamma)\Delta C_A$ has been clearly defined in L117-119.

[Comment B25]: *L686 Is the correct equation referenced?(19) has not yet been reached.*

[Response B25]: Thanks for this detailed comment. We have re-checked the formula deduction and revised the text (L673-688).

[Comment B26]: *L714 FEA not defined*

[Response B26]: The FEA has been defined in the revised version (L133-134).

[Comment B27]: *L1060 Define what the global-mean temperature is, i.e surface air temperature.*

[Response B27]: Agreed and changed (L1040-1041). The global-mean temperature was area-weighted averaged from land surface air temperature and sea surface temperature (also see Methods, L554-559).

References:

- Cox P, Jones C (2008) Climate change. Illuminating the modern dance of climate and CO₂. *Science*, **321**, 1642-1644.
- Dai A, Huang D, Rose BEJ, Zhu J, Tian X (2020) Improved methods for estimating equilibrium climate sensitivity from transient warming simulations. *Climate Dynamics*, **54**, 4515-4543.
- Frank DC, Esper J, Raible CC, Buntgen U, Trouet V, Stocker B, Joos F (2010) Ensemble reconstruction constraints on the global carbon cycle sensitivity to climate. *Nature*, **463**, 527-530.
- Friedlingstein P, Cox P, Betts R *et al.* (2006) Climate–Carbon Cycle Feedback Analysis: Results from the C4MIP Model Intercomparison. *Journal of Climate*, **19**, 3337-3353.
- Gregory JM, Jones CD, Cadule P, Friedlingstein P (2009) Quantifying Carbon Cycle Feedbacks. *Journal of Climate*, **22**, 5232-5250.
- He J, Winton M, Vecchi G, Jia L, Rugenstein M (2017) Transient Climate Sensitivity Depends on Base Climate Ocean Circulation. *Journal of Climate*, **30**, 1493-1504.
- Huang K, Xia J (2019) High ecosystem stability of evergreen broadleaf forests under severe droughts. *Glob Chang Biol*, **25**, 3494-3503.
- Scheffer M, Brovkin V, Cox PM (2006) Positive feedback between global warming and atmospheric CO₂ concentration inferred from past climate change. *Geophysical Research Letters*, **33**, L10702.
- Tierney JE, Poulsen CJ, Montañez IP *et al.* (2020) Past climates inform our future. *Science*, **370**.
- Wang H, Atkin OK, Keenan TF *et al.* (2020) Acclimation of leaf respiration consistent with optimal photosynthetic capacity. *Glob Chang Biol*.
- Willeit M, Ganopolski A, Dalmonech D, Foley AM, Feulner G (2014) Time-scale and state dependence of the carbon-cycle feedback to climate. *Climate Dynamics*, **42**, 1699-1713.